# PRICE OF QUALITY: SUFFICIENT CONDITIONS FOR SPARSE RECOVERY USING MIXED-QUALITY DATA

**Youssef Chaabouni**
Operations Research Center
Massachusetts Institute of Technology
Cambridge, MA 02139, USA
youss404@mit.edu

**David Gamarnik**
Operations Research Center
Massachusetts Institute of Technology
Cambridge, MA 02139, USA
gamarnik@mit.edu

## ABSTRACT

We study sparse recovery when observations come from mixed-quality sources: a small collection of high-quality measurements with small noise variance and a larger collection of lower-quality measurements with higher variance. For this heterogeneous-noise setting, we establish sample-size conditions for information-theoretic and algorithmic recovery. On the information-theoretic side, we show that it is sufficient for $(n_1, n_2)$ to satisfy a linear trade-off defining the *Price of Quality*: the number of low-quality samples needed to replace one high-quality sample. In the agnostic setting, where the decoder is completely agnostic to the quality of the data, it is uniformly bounded, and in particular one high-quality sample is never worth more than two low-quality samples for this sufficient condition to hold. In the informed setting, where the decoder is informed of per-sample variances, the price of quality can grow arbitrarily large. On the algorithmic side, we analyze the LASSO in the agnostic setting and show that the recovery threshold matches the homogeneous-noise case and only depends on the average noise level, revealing a striking robustness of computational recovery to data heterogeneity. Together, these results give the first conditions for sparse recovery with mixed-quality data and expose a fundamental difference between how the information-theoretic and algorithmic thresholds adapt to changes in data quality.

## 1 INTRODUCTION

### 1.1 OVERVIEW AND PREVIOUS WORK

#### 1.1.1 SPARSE RECOVERY

Sparse recovery is a central problem in high-dimensional statistics and machine learning. Its applications include compressive sensing (Foucart et al., 2013; Candès et al., 2006; Donoho, 2006), signal denoising (Chen et al., 2001), sparse regression (Miller, 2002), data-stream algorithms (Cormode & Hadjieleftheriou, 2009; Indyk, 2007; Muthukrishnan et al., 2005), and combinatorial group testing (Du & Hwang, 1999). Other applications range from medical imaging to communications and compression (Foucart et al., 2013, Chap. 1).

We formulate the problem as follows. A high-dimensional *signal* $\beta^\star \in \mathbb{R}^p$ (also called *model* or *ground truth*), unknown but a-priori $s$-sparse, is transmitted through a noisy channel that projects it onto a collection of $n$ random vectors $\{x_i\}_{i \in [n]}$ in $\mathbb{R}^p$. This is expressed as:

$$Y := X\beta^\star + Z, \tag{1}$$

where $X = (x_1, \ldots, x_n)^T$ is called *measurements*, *design* or *features*; $Y$ *observations*, *annotations* or *labels*; and $Z$ *noise*. Specifically, we consider the setting of additive Gaussian noise, which is standard in the compressive sensing and sparse linear regression literatures. On the other end of the channel, a decoder who observes $(X, Y)$ is interested in recovering the support of the original signal $\beta^\star$, i.e. the subset $S^\star := \{i \in [p] : \beta_i^\star \neq 0\} \subseteq [p]$, known a-priori to be of cardinality $s$. How many observations $n$ (as a function of $p$ and $s$) does the decoder need to recover the support of the signal as the dimension of the problem grows to infinity?

Previous works have shown that the sparse recovery problem exhibits two phase transitions at two thresholds, one *information-theoretic* and one *algorithmic*:

$$n_{\text{INF}} = \frac{2s \log (p/s)}{\log s} \quad \text{and} \quad n_{\text{ALG}} = 2s \log (p - s) + s + 1, \tag{2}$$

leading to three regimes:

- $n < n_{\text{INF}}$: Signal support recovery is information-theoretically impossible. (Reeves et al., 2019).

- $n_{\text{INF}} < n < n_{\text{ALG}}$: The maximum likelihood estimator (MLE) recovers $S^\star$. However, it is believed that no algorithm can do it in polynomial time since the problem exhibits an Overlap Gap Property (OGP) (Gamarnik & Zadik, 2022), a structural property of the solution space known to often imply the failure of tractable algorithms to find optimal solutions.

- $n > n_{\text{ALG}}$: The $\ell_1$-regularized least-squares estimator (also known as the LASSO (Tibshirani, 1996)) recovers $S^\star$ (Wainwright, 2009).

Of particular interest is the signal-to-noise ratio (SNR), known to be an important quantity for characterizing the difficulty of sparse recovery problems (Wang et al., 2010; Reeves et al., 2019; Chaabouni & Gamarnik). It's defined as follows:

$$\text{SNR} := \frac{\mathbb{E}\|X\beta^\star\|_2^2}{\mathbb{E}\|Z\|_2^2}. \tag{3}$$

### 1.1.2 MIXED QUALITY DATA

A recent body of work has explored how low-quality data, e.g. labeled by an LLM or weak annotator (Ratner et al., 2017; Frénay & Verleysen, 2013), should be combined with fewer but higher-quality data, e.g. labeled by humans or experts, for prediction and inference tasks (Gligorić et al., 2024; Li et al., 2023; Zhang et al., 2023; Egami et al., 2023).

In this paper, we formalize the mixed-quality data setting for sparse signal recovery: the decoder has access to $n_1$ noisy projections of the signal $\beta^\star$ with a small noise level $\sigma_1^2 > 0$ that we denote $\{(y_i, x_i)\}_{i=1}^{n_1}$ and call *high-quality* data. In addition, the decoder also observes a larger set of $n_2 > n_1$ noisy projections of the same signal $\beta^\star$, but with a higher noise level $\sigma_2^2 > \sigma_1^2$, that we denote $\{(y_i, x_i)\}_{i=n_1+1}^{n_2}$ and call *low-quality* data. We distinguish two settings:

- **Agnostic setting**: The decoder lacks access to observation-level noise variances and treats all measurements as if drawn from a single homogeneous model. This occurs when heterogeneous data sources lose provenance: for example in web-scale text corpora (Ratner et al., 2017; Frénay & Verleysen, 2013) or citizen-science campaigns lacking sensor calibration (Silvertown, 2009). The decoder simply applies standard sparse-recovery methods without noise estimation or reweighting.

- **Informed setting**: where the decoder has access to the per-sample noise variance of the data. This regime captures situations where provenance information accompanies each observation, so the decoder knows which measurements are high- or low-quality. Examples include multi-site clinical trials or sensor networks that log calibration statistics (Loh & Wainwright, 2011; Delaigle et al., 2008), and medical-imaging datasets with per-rater confidence scores (Rajpurkar et al., 2018).

### 1.2 OUR WORK

In this paper, we consider the sparse recovery problem described above (1). Specifically, we study the setting where the measurements are drawn i.i.d. from a standard normal Gaussian distribution, and the noise is unbiased and drawn independently from Gaussian distributions of variance $\sigma_1^2$ for the high-quality samples and $\sigma_2^2 > \sigma_1^2$ for the low-quality ones:

$$\{X_{ij}\}_{i\in[n],j\in[p]} \overset{\text{i.i.d.}}{\sim} \mathcal{N}(0,1) \text{ and } Z = \Sigma W; \text{ where } \Sigma = \begin{pmatrix} \sigma_1 I_{n_1} & 0 \\ 0 & \sigma_2 I_{n_2} \end{pmatrix}, W \sim \mathcal{N}(0, I_n). \tag{4}$$

Although much of the literature on sparse recovery in the homogeneous noise setting assumes constant noise level $\sigma^2$, we don't assume in this work that $\sigma_1^2$ and $\sigma_2^2$ are constant. In fact, the reason

previous work can assume constant noise variance without loss of generality is that the model (1) could be scaled down by $\sigma$ when the noise is homogeneous with variance $\sigma^2$ to make it constant. However, it is not the case anymore when the noise is heterogeneous. While our results naturally extend to sub-Gaussian errors, the sufficient conditions derived herein are not universal for general additive noise distributions.

The assumptions we impose (Gaussian design, exact sparsity, and additive noise) are standard in the sparse recovery literature (e.g. Wainwright (2009); Reeves et al. (2019); Gamarnik & Zadik (2022)), and are adopted here to isolate the effect of heterogeneous noise while retaining the canonical structure of the recovery problem.

Our analysis allows for arbitrary scalings of $\sigma_1^2$ and $\sigma_2^2$ with respect to $p$ and $s$. Since data come from two different sources with different scalings, we define in addition to (3) two signal-to-noise ratios: $\text{SNR}_1$ for high-quality observations and $\text{SNR}_2$ corresponding to low-quality observations.

We are interested in the two following questions:

- **Sampling complexity of sparse recovery:** How large do the sample sizes $(n_1, n_2)$ need to be for the decoder to be able, information-theoretically, to recover the support of the signal?

- **Algorithmic recovery:** How large do the sample sizes $(n_1, n_2)$ need to be for the decoder to be able to recover the support of the signal using a polynomial-time algorithm?

We summarize below our findings on each of these questions in the agnostic and informed settings.

### 1.2.1 SAMPLING COMPLEXITY OF SPARSE RECOVERY

In the first part of our work (section 3), we focus on the question of sampling complexity. For simplicity, we assume the signal is binary, i.e. $\beta^\star \in \{0, 1\}^p$. Note that in this case, recovering the support is equivalent to recovering the signal. This assumption is very common in the literature (Aeron et al., 2010; Reeves et al., 2019; Gamarnik & Zadik, 2022; Chaabouni & Gamarnik). Intuitively, detecting a component of size 1 is at least as hard as detecting a stronger component, so the resulting thresholds are representative of signals with non-zero entries bounded away from zero. We discuss this assumption in more detail in Remark 3.1.

Our main results, Theorem 1 for the agnostic setting and Theorem 2 for the informed one, each provide a sufficient condition (9, 16) on the sample sizes $(n_1, n_2)$ for support recovery. In both results, the condition has the form $\alpha_1 n_1 + \alpha_2 n_2 > n^\star$, for some coefficients $\alpha_1, \alpha_2 > 0$ depending on $\sigma_1^2, \sigma_2^2$ and $s$, and having different expressions in the agnostic and informed settings. In particular, we note that if $(n_1, n_2)$ verify this condition (i.e. are together large enough), then so do $(n_1 - 1, n_2 + \alpha_1/\alpha_2)$. In this sense, we say that one unit of high-quality data is worth:

$$\gamma\left(s, \sigma_1^2, \sigma_2^2\right) \coloneqq \frac{\alpha_1}{\alpha_2} \tag{5}$$

units of low-quality data for the sufficient condition to hold. We label $\gamma$ the *Price of Quality* and study its behavior in the agnostic and informed settings and for different regimes of $\text{SNR}_1$ and $\text{SNR}_2$. In the agnostic setting, it is uniformly bounded. In particular, under our sufficient condition, one high-quality sample is never worth more than *two* low-quality samples (13, 14) for the sufficient condition. In the informed setting, where the decoder is informed of per-sample variances, the price of quality goes to infinity in the low $\text{SNR}_2$ & high $\text{SNR}_1$ regime (20), and can be arbitrarily large in both low and high SNR regimes (19, 21).

### 1.2.2 ALGORITHMIC RECOVERY

In the second part of our work (section 4), we focus on the question of algorithmic recovery. Unlike for sampling complexity, we don't assume that the signal is binary, but still require non-zero components to be bounded away from zero, i.e. there exists $\rho > 0$ such that $\min_{i \in S^\star} |\beta_i^\star| \geq \rho$. This is standard in the literature (Aeron et al., 2010; Ndaoud & Tsybakov, 2020; Wang et al., 2010) since we can't hope to detect non-zero signal components if they can have arbitrarily small amplitude.

Specifically, we study the question of *signed support* recovery, that is, recovering not only the indices of the non-zero components of the signal but also their sign ($+$ or $-$). This is usual in the algorithmic

sparse recovery literature (Wainwright, 2009; Wang et al., 2010; Omidiran & Wainwright, 2008), as it follows naturally from the standard proof techniques.

Our main result, Theorem 3, provides necessary and sufficient conditions for the $\ell_1$-regularized least-squares estimator (known as the LASSO) to recover the signed support of $\beta^\star$ in the agnostic setting. Our result reveals that the problem behaves like the homogeneous-noise setting (Wainwright, 2009) with a homogeneous noise level equal to the average noise level of $Z$:

$$\sigma_{\text{avg}}^2 := \frac{n_1\sigma_1^2 + n_2\sigma_2^2}{n}. \tag{6}$$

In particular, the sample size conditions (26, 27) do not depend on the noise levels $\sigma_1^2, \sigma_2^2$. The condition on the LASSO regularization parameter (28) only depends on $\sigma_1^2$ and $\sigma_2^2$ through $\sigma_{\text{avg}}^2$ and is the same as the one for homogeneous noise $\sigma_{\text{avg}}^2$ (see equation (28) in Wainwright (2009)). We further provide a necessary and sufficient condition on noise scaling (Proposition 4.1).

This shows that, unlike in sampling complexity, high-quality and low-quality data contribute equally to the sample size condition under which the LASSO recovers the support of the signal.

Although we don't address algorithmic recovery in the informed setting, we briefly discuss it in Remark 4.2, where we discuss why the proof of Theorem 3 cannot be easily extended to the informed case.

## 1.3 CONTRIBUTIONS, OUTLINE AND NOTATIONS

To the best of our knowledge, this paper is the first to:

1. Provide a sufficient condition for sparse recovery in the heterogeneous noise case, and quantify the trade-off between high-quality and low-quality data in the agnostic and informed settings.

2. Extend necessary and sufficient conditions for LASSO sparse recovery to the heterogeneous-noise, agnostic setting and show that high-quality and low-quality data contribute equally to reaching the algorithmic threshold.

We organize the rest of the paper as follows. Section 2 introduces the problem setup. Section 3 studies the sampling complexity of sparse recovery under heterogeneous noise. Section 4 investigates algorithmic recovery using the LASSO. Section 5 concludes and outlines directions for future work.

Throughout this document, we will use the following notations:

- We say that $f(x) \simeq g(x)$ as $x \to a \in \mathbb{R} \cup \{-\infty, +\infty\}$ if and only if $f(x) = g(x)(1 + o(1))$.

- We denote by $h(\cdot)$ the binary entropy: $h(x) = -x\log x - (1-x)\log(1-x)$, $x \in (0, 1)$.

- We call $\ell_0$-*norm* the number of non-zero coordinates of $x \in \mathbb{R}^d$, that is $\|x\|_0 := \sum_{i=1}^d \mathbb{1}(x_i \neq 0)$.

- We use uppercase letters (e.g. $X, Y, Z$) to indicate random quantities, and lowercase letters (e.g. $\beta$) to denote deterministic parameters.

## 2 PRELIMINARIES

The problem of sparse signal recovery is defined above (1). The decoder a-priori knows that $\beta^\star$ is $s$-sparse and belongs to a known set $\mathcal{A} \subseteq \mathbb{R}^p$. The design and noise are random with $(X_{ij})_{i\in[n], j\in[p]} \overset{\text{i.i.d.}}{\sim} \mathcal{N}(0, 1)$ and $Z := \Sigma W$ with $\Sigma$ and $W$ defined as in (4) and $n_1 + n_2 = n$. We define $Z^1 := (Z_1, \ldots, Z_{n_1})^T$ and $Z^2 := (Z_{n_1+1}, \ldots, Z_n)^T$, so that $Z = \begin{pmatrix} Z^1 \\ Z^2 \end{pmatrix}$. The signal-to-noise ratio (3) writes:

$$\text{SNR} := \frac{\mathbb{E}\|X\beta\|_2^2}{\mathbb{E}\|Z\|_2^2} = \frac{ns}{n_1\sigma_1^2 + n_2\sigma_2^2} = \frac{s}{\sigma_{\text{avg}}^2}, \tag{7}$$

where $\sigma_{\text{avg}}^2$ denotes the average noise level (6). In addition, we define the *high-quality SNR* and the *low-quality SNR* respectively by:

$$\text{SNR}_1 := \frac{\mathbb{E}\left\|\left[y_i - x_i^T\beta^\star\right]_{i=1}^{n_1}\right\|_2^2}{\mathbb{E}\|Z^1\|_2^2} = \frac{s}{\sigma_1^2} \ , \ \text{SNR}_2 := \frac{\mathbb{E}\left\|\left[y_i - x_i^T\beta^\star\right]_{i=n_1+1}^{n_2}\right\|_2^2}{\mathbb{E}\|Z^2\|_2^2} = \frac{s}{\sigma_2^2}.$$

In particular, we always have $\text{SNR}_2 < \text{SNR}_1$, which reveals three regimes of interest:

- High SNR: when $\text{SNR}_1, \text{SNR}_2 \to +\infty$, or equivalently $\sigma_2^2 = o(s)$.
- Low $\text{SNR}_2$, High $\text{SNR}_1$: $\text{SNR}_2 \to 0$, $\text{SNR}_1 \to +\infty$ or equivalently $\sigma_2^2 = \omega(s)$, $\sigma_1^2 = o(s)$.
- Low SNR: when $\text{SNR}_1, \text{SNR}_2 \to 0$, or equivalently $\sigma_1^2 = \omega(s)$.

## 3 SAMPLING COMPLEXITY OF SPARSE RECOVERY

In this section, we are interested in determining whether it is possible, information-theoretically, to recover the support of the signal, depending on the sample size $n$. We assume that $\beta^\star$ is binary and a priori $s$-sparse, that is: $\mathcal{A} := \mathcal{B}_{p,s} = \{\beta \in \{0,1\}^p : \|\beta\|_0 = s\}$.

**Remark 3.1** (Binary-signal assumption). Our results for sparse recovery can be viewed as applying to signals whose non-zero components are at least 1 in magnitude, i.e. $\beta^\star \in \mathcal{C}_{p,s}(1) := \left\{\beta \in \mathbb{R}^d : \min_{i \in \text{Supp}(\beta)} |\beta_i| \geq 1\right\}$. Assuming that the non-zero entries are exactly equal to 1 serves only to simplify computations. Intuitively, detecting a component of magnitude 1 is at least as hard as detecting a stronger component, so stronger signals can only make recovery easier. Conversely, detecting a signal in $\mathcal{C}_{p,s}(1)$ is at least as hard as detecting a binary signal, since $\{0,1\}^p \subseteq \mathcal{C}_{p,s}(1)$. More generally, recovering any signal whose non-zero entries are bounded below by some $\rho > 0$ can be reduced to the case of $\mathcal{C}_{p,s}(1)$ by rescaling the model (1) by $\rho$.

Let $A \triangle B := (A \cup B) \setminus (A \cap B)$ denote the symmetric difference between any two finite sets $A$ and $B$, and $\text{Supp}(\beta) := \{i \in [p] : \beta_i \neq 0\}$ denote the support of any vector $\beta \in \mathbb{R}^p$. Let $\delta \in (0,1)$. We say that $\hat{\beta} \in \mathcal{B}_{p,s}$ *recovers the support of $\beta^\star$ up to error $\delta$* if $\left|\text{Supp}\left(\hat{\beta}\right) \triangle \text{Supp}(\beta^\star)\right| < 2\delta s$.

### 3.1 AGNOSTIC SETTING

In the agnostic setting where the decoder ignores the quality of each observation, the sample sizes $(n_1, n_2)$ and the noise levels $(\sigma_1^2, \sigma_2^2)$. Motivated by the maximum likelihood estimator in the homoscedastic setting (see Gamarnik & Zadik (2022)), we define an estimator such that:

$$\hat{\beta} \in \arg\min_{\beta \in \mathcal{B}_{p,s}} \|Y - X\beta\|_2^2. \tag{8}$$

**Theorem 1** (Sufficient condition for support recovery in the agnostic setting).

*1. Assume $s = o(p)$ and $s \to +\infty$ as $p \to +\infty$. Then let $n^\star := 2s \log(p/s)$.*

*2. Assume $s = \alpha p$ for some constant $\alpha \in (0,1)$. Then let $n^\star := 2h(\alpha)p$.*

*In both settings described above, if there exists $\varepsilon > 0$ such that:*

$$n_1 \log\left(1 + \frac{\delta\left(2\sigma_2^2 - \sigma_1^2\right)s}{2\sigma_2^4}\right) + n_2 \log\left(1 + \frac{\delta s}{2\sigma_2^2}\right) \geq (1+\varepsilon)n^\star, \tag{9}$$

*then $\hat{\beta}$ recovers the support of $\beta^\star$ up to error $\delta$ w.h.p.:*

$$\mathbb{P}\left(\left|\text{Supp}(\beta^\star) \triangle \text{Supp}\left(\hat{\beta}\right)\right| < 2\delta s\right) \geq 1 - \exp\left\{-(\varepsilon + o(1))n^\star/2\right\} \overset{p \to +\infty}{\longrightarrow} 1. \tag{10}$$

*Proof Sketch.* The proof of Theorem 1 is in appendix A and uses standard techniques. We control the probability that a high-error support attains a lower objective value in (8) than the ground truth and then take a union bound over such supports. For any $\beta$, we have:

$$\|Y - X\beta\|_2^2 - \|Y - X\beta^\star\|_2^2 = \sum_{i=1}^{n}\left\{\langle X_i, \beta^\star - \beta\rangle^2 + 2Z_i\langle X_i, \beta^\star - \beta\rangle\right\}. \tag{11}$$

Applying a Chernoff bound to the RHS above and analyzing the MGF of the summands yields an exponent that factorizes across two blocks (see Proposition A.1). We conclude using a union bound over supports $S$ with $|S \triangle S^\star| \geq 2\delta s$ (there are at most $\binom{p}{s}$ of them). $\qquad \square$

We interpret Theorem 1 as follows.

- **Sample Complexity.** In our setup, the decoder knows that $\beta^\star$ is exactly $s$-sparse. When $s = 0$ or $s = p$, the support is fully determined and there is no ambiguity, making recovery trivial and requiring no samples. For intermediate values of $s$, the decoder must distinguish among many candidate supports, whose cardinality is $\binom{p}{s}$. This combinatorial ambiguity renders the recovery problem non-trivial and leads to the sample complexity characterized by $n^\star$ in Theorem 1.

- **Price of Quality.** The sufficient condition for recovery (9) is equivalent to a linear combination of the sample size $n_1$ and $n_2$ being larger than the threshold $n^\star$. The coefficients of the sample sizes reveal that one unit of high-quality data is worth:

$$\gamma := \frac{\log\left(1 + \delta\left(2\sigma_2^2 - \sigma_1^2\right)s / \left(2\sigma_2^4\right)\right)}{\log\left(1 + \delta s / \left(2\sigma_2^2\right)\right)} > 1 \tag{12}$$

  units of low-quality data for the sufficient condition to hold. We call $\gamma$ the *Price of Quality*. In fact, one unit of high-quality data can be replaced by $\gamma$ units of low-quality data: that is, if $(n_1, n_2)$ are large enough for the sufficient condition to hold (and are hence sufficient for recovering $\beta^\star$), then so are $(n_1 - 1, n_2 + \gamma)$.

- **High SNR$_2$ regime.** Assume $s = \omega\left(\sigma_2^2\right)$. The price of quality (12) writes:

$$\gamma \simeq \frac{\log\left(\delta s / \left(2\sigma_2^2\right)\right) + \log\left(2 - \sigma_1^2/\sigma_2^2\right)}{\log\left(\delta s / \left(2\sigma_2^2\right)\right)} \simeq 1, \tag{13}$$

  which means that when $\sigma_1^2, \sigma_2^2 = o(s)$, the high-quality and low-quality data contribute equally to the recovery condition (9).

- **Low SNR$_2$ regime.** Assume $s = o\left(\sigma_2^2\right)$. The price of quality (12) writes:

$$\gamma \simeq \frac{\delta\left(2\sigma_2^2 - \sigma_1^2\right)s / \left(2\sigma_2^4\right)}{\delta s / \left(2\sigma_2^2\right)} \simeq 2 - \frac{\sigma_1^2}{\sigma_2^2}. \tag{14}$$

  Note that $\gamma < 2$ for any $\sigma_1^2, \sigma_2^2$. We conclude, in the low SNR regime, that under our sufficient condition, one high-quality sample can be replaced by up to two low-quality samples.

**Remark 3.2** (Limitations).

- The condition in Theorem 1 is sufficient and is not expected to be information-theoretically sharp. The potential looseness arises from a relaxation in the Chernoff bound used to control the probability of support misidentification. In the heterogeneous-noise setting, optimizing the Chernoff exponent leads to a cubic equation (see (37)), whose exact solution yields a tighter but less tractable condition. To retain a closed-form and interpretable sufficient condition, we rely on a relaxation of this equation. In the homogeneous-noise case, solving the analogous equation is known to recover the sharp threshold, and we expect that optimizing (37) would similarly lead to a tighter characterization here, though we do not pursue this direction in the present work.

- Even under the assumption that the decoder is agnostic to the quality of the data, the estimator $\hat{\beta}$ (8), might not constitute the best approach to recover the support of $\beta^\star$. For instance, especially in the low SNR regime, the decoder might re-weight the loss of each observation by the magnitude of its observed label, i.e.:

$$\arg\min_{\beta \in \mathcal{B}_{p,s}} \sum_{i=1}^{n} \frac{1}{Y_i^2}\left(Y_i - \langle x_i, \beta \rangle\right)^2,$$

  as an attempt to rescale each row of data by its noise level. In fact, in the low SNR regime we have $\mathbb{E}Y_i^2 \simeq \sigma_i^2$ where $\sigma_i^2$ denotes the noise level corresponding to the $i^{\text{th}}$ observation, which motivates the use of $Y_i^2$ as a proxy for $\sigma_i^2$ when the noise levels are unknown.

- Classical approaches to heteroscedastic regression either assume known noise levels or explicitly acknowledge heteroscedasticity as part of the statistical modeling assumptions (see, e.g. Buja et al. (2019)). Extending such considerations to sparse support recovery in the mixed-quality setting introduces an additional layer of difficulty, since one must control both the accuracy of variance-related modeling and its impact on support identification. While variance-aware procedures may improve performance when the noise levels differ significantly, a rigorous analysis of such methods is beyond the scope of this work and constitutes an interesting direction for future research.

## 3.2 INFORMED SETTING

In this section, we assume that the decoder knows the distribution of each noise entry: $\mathcal{N}\left(0, \sigma_1^2\right)$ or $\mathcal{N}\left(0, \sigma_2^2\right)$. Recall the distributions of $Z$ and $W$ from (4). The MLE is define by (see appendix B for a proof):

$$\hat{\beta}_{\mathrm{MLE}} \in \underset{\beta \in \mathcal{B}_{p,s}}{\arg \min} \left\| \Sigma^{-1} \left(Y - X\beta\right) \right\|_2^2. \tag{15}$$

**Theorem 2** (Sufficient condition for support recovery in the informed setting)**.**

*1. Assume $s = o\left(p\right)$ and $s \to +\infty$ as $p \to +\infty$. Then let $n^\star := 2s \log\left(p/s\right)$.*

*2. Assume $s = \alpha p$ for some constant $\alpha \in (0, 1)$. Then let $n^\star := 2h\left(\alpha\right) p$.*

*In both settings described above, if there exists $\varepsilon > 0$ such that:*

$$n_1 \log\left(1 + \frac{\delta s}{2\sigma_1^2}\right) + n_2 \log\left(1 + \frac{\delta s}{2\sigma_2^2}\right) \geq (1 + \varepsilon)\, n^\star, \tag{16}$$

*then $\hat{\beta}_{MLE}$ recovers the support of $\beta^\star$ up to error $\delta$ w.h.p.:*

$$\mathbb{P}\left(\left| Supp\left(\beta^\star\right) \triangle Supp\left(\hat{\beta}_{MLE}\right)\right| < 2\delta s\right) \geq 1 - \exp\left\{-\left(\varepsilon + o\left(1\right)\right) n^\star/2\right\} \overset{p \to +\infty}{\longrightarrow} 1. \tag{17}$$

*Proof Sketch.* The proof of Theorem 2 is given in appendix C and follows a similar argument as Theorem 1. Here, the rescaled loss in (15) leads to a Chernoff bound that can be optimized in closed-form, yielding a sharp convergence rate. □

We interpret Theorem 2 as follows.

- **Price of Quality.** In the informed setting, the expression of the price of quality is different from the one in the agnostic case (12). It writes:

$$\gamma = \log\left(1 + \frac{\delta s}{2\sigma_1^2}\right) \Big/ \log\left(1 + \frac{\delta s}{2\sigma_2^2}\right). \tag{18}$$

- **Low SNR regime.** Assume $\sigma_1^2 = \omega\left(s\right)$. Then:

$$\gamma \simeq \sigma_2^2/\sigma_1^2. \tag{19}$$

- **Low SNR$_2$, High SNR$_1$ regime.** Assume $\sigma_2^2 = \omega\left(s\right)$ and $\sigma_1^2 = o\left(s\right)$. Then:

$$\gamma = \Theta\left(\frac{\log\left(s/\sigma_1^2\right)}{s/\sigma_2^2}\right) = \Theta\left(\frac{\log \mathrm{SNR}_1}{\mathrm{SNR}_2}\right) \overset{p \to +\infty}{\longrightarrow} +\infty. \tag{20}$$

- **High SNR regime.** Assume $\sigma_2^2 = o\left(s\right)$. Then:

$$\gamma \simeq \log\left(s/\sigma_1^2\right) \Big/ \log\left(s/\sigma_2^2\right) = \log \mathrm{SNR}_1 / \log \mathrm{SNR}_2. \tag{21}$$

**Remark 3.3.**

- Compared to the agnostic setting (Theorem 1), the appropriate rescaling of the loss in the MLE (15) constitutes a better use of the high-quality data, in the sense that it leads to a higher price of quality $\gamma$. In particular, $\gamma$ is infinite in the low SNR$_2$ & high SNR$_1$ setting (20) and can be arbitrarily large in both low and high SNR regimes (19, 21).

- The sufficient condition (16) is obtained by optimizing the Chernoff exponent exactly (see (39) and (42)). In homogeneous-noise settings, analogous optimizations are known to yield necessary and sufficient thresholds (Gamarnik & Zadik, 2022; Wang et al., 2010; Chaabouni & Gamarnik). Establishing full necessity in the heterogeneous setting remains an interesting direction for future work.

**Remark 3.4** (Generalizations of Theorem 1 and Theorem 2)**.**

- **Generalization to *signed* support recovery.** The large-deviation bound in the proofs of Theorem 1 and Theorem 2 suggests a potential extension of the sufficient conditions, respectively (9) and (16), to the setting where the non-zero components of the signal $\beta^\star$ are not necessarily $+1$ but rather in $\{-1, +1\}$, and the decoder is interested in the *signed* support recovery, where they recover not only the indices of the non-zero components of $\beta^\star$, but also their sign. In this setting, the error measure expressed by the symmetric difference of supports in (10) and (17) extends to the number of 'wrong' components in $\hat{\beta}$, given by $\|\hat{\beta} - \beta^\star\|_0$. The threshold $n^\star \simeq \log \binom{p}{s}$ would increase by an additive factor of $s \log 2$ to account for the increase in the size of the search space $\left(\text{since } \beta^\star \in \{\beta \in \{-1, 0, 1\}^p : \|\beta\|_0 = s\}, \text{ which has cardinality } 2^s \binom{p}{s}\right)$. Asymptotically, this does not change the leading-order scaling in the sub-linear regime when $s = o(p)$, and adds an extra $\alpha \log 2$ term to the $h(\alpha)$ factor in the linear regime when $s = \alpha p$.

- **Generalization to *arbitrary noise* structures.** Theorem 1 and Theorem 2 are stated in the simple setting where the data comes from two sources, one *good* and one *bad*, motivated by the high- and low- quality data problem. The proof strategy suggests that these results extend to non-singular noise. In fact, if we only assume that $\Sigma$ is invertible, but not necessarily having the form in (4), then the sufficient condition (9) in Theorem 1 extends to:

$$\sum_{i=1}^n \log \left(1 + \frac{\delta \left(2\sigma_{\max}(\Sigma)^2 - \sigma_i(\Sigma)^2\right) s}{2\sigma_{\max}(\Sigma)^4}\right) \geq (1 + \varepsilon) n^\star, \tag{22}$$

where $\{\sigma_i(\Sigma)\}_{i=1}^{i=n}$ denote the $\sigma$-values of $\Sigma$, $\sigma_{\max}(\Sigma) := \max_{i=1,\ldots,n} \sigma_i(\Sigma)$ and $\sigma_{\min}(\Sigma) := \min_{i=1,\ldots,n} \sigma_i(\Sigma)$. Similarly, the sufficient condition (16) in Theorem 2 extends to:

$$\sum_{i=1}^n \log \left(1 + \frac{\delta s}{2\sigma_i(\Sigma)^2}\right) \geq (1 + \varepsilon) n^\star. \tag{23}$$

## 4 Algorithmic recovery

In this section, we are interested in the existence of a tractable algorithm to recover the support of the underlying signal. We assume that the components of the signal $\beta^\star$ take real values and are bounded away from zero: that is, $\mathcal{A} := \mathcal{C}_{p,s}(\rho) = \{\beta \in \mathbb{R}^p : \|\beta\|_0 = s, \min_{i \in \text{Supp}(\beta)} |\beta_i| \geq \rho\}$, for some $\rho \in \mathbb{R}_+$. We say that $\hat{\beta} \in \mathbb{R}^p$ recovers the signed support of $\beta^\star$ if $\text{sign}(\hat{\beta}) = \text{sign}(\beta^\star)$, where the $\text{sign} \colon \mathbb{R} \longrightarrow \{-1, 0, 1\}$ function is defined by $\text{sign}(0) = 0$ and $\text{sign}(x) = x/|x|$ for all $x \neq 0$, and is applied coordinate-wise. A common approach to recovering the signed support of the signal is using the solution to the following $\ell_1$-constrained quadratic program, also known as the LASSO:

$$\mathcal{B}_{\text{LASSO}} := \arg\min_{\beta \in \mathbb{R}^p} \left\{\frac{1}{2n} \|Y - X\beta\|_2^2 + \lambda_p \|\beta\|_1\right\}, \tag{24}$$

where $\lambda_p \geq 0$ denotes a sequence of regularization parameters converging to 0 as $p \to +\infty$. We are interested in characterizing the regime where the LASSO recovers the signed support of the true signal. Specifically, we call "recovery" the event:

$$\mathcal{R}(X, \beta^\star, Z, \lambda_p) := \left\{\exists \hat{\beta} \in \mathcal{B}_{\text{LASSO}} \colon \text{sign}\left(\hat{\beta}\right) = \text{sign}(\beta^\star)\right\}. \tag{25}$$

In the homogeneous noise setting, Wainwright (2009) showed that the performance of the LASSO in estimating the signed support of $\beta^\star$ exhibits a phase transition with respect to the sample size. In fact, there exists a threshold $n_{\text{ALG}}$ such that:

- If $n > n_{\text{ALG}}$: then the LASSO correctly recovers the signed support of $\beta^\star$.

- If $n < n_{\mathrm{ALG}}$: then the LASSO fails to recover the signed support of $\beta^\star$.

In addition, it is widely believed that no algorithm can recover the support of $\beta^\star$ in polynomial time when $n < n_{\mathrm{ALG}}$. Indeed, Gamarnik & Zadik (2022) showed that the problem exhibits an OGP. This motivates the use of (24) to estimate $\beta^\star$ in the agnostic setting where the decoder treats the data impartially. Our main result of this section, Theorem 3, extends the result mentioned above on the LASSO threshold (by Wainwright (2009)) to the heterogeneous, agnostic noise setting.

**Theorem 3** (Lasso recovery phase transition). *Assume that, as $p \to +\infty$; $s$ goes to infinity, $s = o(p)$ and $n_1, n_2 = \omega(s)$. Let $n_{ALG} := 2s \log(p - s) + s + 1$.*

*i. If there exists $\varepsilon > 0$ such that:*
$$n < (1 - \varepsilon) n_{ALG}, \tag{26}$$

*then, for any sequence $\lambda_p > 0$ such that $\frac{n_1 \sigma_1^2 + n_2 \sigma_2^2}{\lambda_p^2 n^2}$ has a limit in $\mathbb{R}_{\geq 0} \cup \{+\infty\}$, we have*
$$\mathbb{P}_{X,Z}\Big(\mathcal{R}\left(X, \beta^\star, Z, \lambda_p\right)\Big) \to 0.$$

*ii. If there exists $\varepsilon > 0$ such that:*
$$n > (1 + \varepsilon) n_{ALG}, \tag{27}$$

*and $(\lambda_p)_{p \geq 1} \to 0$ is chosen such that:*
$$\frac{n \lambda_p^2}{\sigma_{avg}^2 \log(p - s)} \to +\infty, \quad and \quad \frac{1}{\rho}\left[\lambda_p \sqrt{s} + \sqrt{\frac{\sigma_{avg}^2 \log s}{n}}\right] \to 0, \tag{28}$$

*then $\mathbb{P}_{X,Z}\Big(\mathcal{R}\left(X, \beta^\star, Z, \lambda_p\right)\Big) \to 1$.*

The full proof of Theorem 3 is given in Appendix D and follows the core LASSO threshold argument of Wainwright (2009). We use the same argument but generalize it to the heterogeneous-noise setting, where the presence of the matrix $\Sigma$, no longer a scalar multiple of the identity, causes key steps of the classical proof to fail. We overcome this by applying a Gram–Schmidt (QR) decomposition of $X_S$ (49) and analyzing the resulting orthogonal matrix using properties of the Haar measure on the orthogonal group (e.g. see Lemma D.6). The monograph of Meckes (2019) on Haar-distributed matrices was particularly valuable in understanding this component from random-matrix theory.

*Proof Sketch of Theorem 3.* We express the recovery property (25) via the first-order optimality conditions of the LASSO (24):

$$\mathcal{R}\left(X, \beta^\star, \Sigma w, \lambda_p\right) \iff \begin{cases} \left| \left(\frac{1}{n} X_S^T X_S\right)^{-1} \left(\frac{1}{n} X_S^T \Sigma w - \lambda_p \operatorname{sign}\left(\beta_S^\star\right)\right) \right| < |\beta_S^\star| \\ \left| X_{S^c}^T X_S \left(X_S^T X_S\right)^{-1} \left(\frac{1}{n} X_S^T \Sigma w - \lambda_p \operatorname{sign}\left(\beta_S^\star\right)\right) - \frac{1}{n} X_{S^c}^T \Sigma w \right| \leq \lambda_p \end{cases} \tag{29}$$

where absolute values and inequalities are taken component-wise. This well-known result (Wainwright, 2009; Fuchs, 2004; Meinshausen & Bühlmann, 2006; Tropp, 2006; Zhao & Yu, 2006) is stated in Proposition $D.1$. When (27) and (28) hold, the random variables inside the absolute values on the RHS of (29) concentrate below their respective upper bounds, establishing sufficiency. When (26) holds, the second absolute value in (29) concentrates above $\lambda_p$, showing necessity. $\square$

Although Theorem 3 does not explicitly state any condition on the scaling on the noise, the existence of $\lambda_p \to 0$ such that (28) holds requires that the noise does not scale arbitrarily large. The next result explicitly states this condition.

**Proposition 4.1** (Necessary and sufficient condition on noise scaling). *If there exists $(\lambda_p)_{p \geq 1} \to 0$ such that (28) holds, then:*
$$\sigma_{avg}^2 = o\left(\frac{n}{\left(1 + s/\rho^2\right) \log(p - s)}\right). \tag{30}$$

*Conversely, if (30) holds, let:*
$$\lambda_p := \left(\frac{\sigma_{avg}^2 \log(p - s)}{\left(1 + s/\rho^2\right) n}\right)^{1/4}. \tag{31}$$

*Then $\lambda_p \to 0$ and (28) holds.*

*Proof.* See appendix E. □

**Remark 4.1** (Correlated features). Theorem 3 is stated for independent features (i.e. $x_i \sim \mathcal{N}(0, I_p)$ for all $i \in [n]$). In the homogeneous-noise setting, analogous results for correlated designs under suitable regularity conditions on the covariance matrix were established by Wainwright (2009). Extending the heterogeneous-noise analysis to correlated designs requires additional tools and is left for future work. In this paper, we therefore focus on the independent-feature case.

**Remark 4.2** (Informed setting). Although we establish the phase transition for the LASSO only in the agnostic setting, a natural extension in the informed setting is the rescaled estimator defined by minimizing $\left\|\Sigma^{-1}(Y - X\beta)\right\|_2^2$ instead of $\|Y - X\beta\|_2^2$ in (24). Extending the proof of Theorem 3 and Wainwright (2009) to this setting is nontrivial, as the presence of $\Sigma^{-1}$ factors alongside the design matrix in (29) destroys the Wishart structure $X_S^T X_S \sim \mathcal{W}(I_s, n)$ used to control the moments of its inverse via classical inverse-Wishart arguments (Anderson et al., 1958; Siskind, 1972). An analysis would therefore require controlling the moments of $(X_S^T \Sigma^{-2} X_S)^{-1}$, which remains an interesting direction for future work.

## 5 Conclusion and Future Work

We study the problem of sparse recovery when observations come from mixed-quality sources. We establish sufficient conditions on the sample sizes $(n_1, n_2)$ for both information-theoretic and algorithmic recovery purposes and in two settings, one when the decoder is completely agnostic to noise and one where they are informed of the per-sample noise variance.

At the level of the information-theoretic threshold, we study the trade-off between high-quality and low-quality samples, and label the number of low-quality samples required to replace one high-quality sample when our sufficient condition holds *the Price of Quality*. In the agnostic setting, we reveal that this entity is quite low: in particular, under our sufficient condition, one high-quality sample is never worth more than two low-quality samples. However, in the informed setting, the price of quality can grow arbitrarily large depending on the noise variances and the signal-to-noise regime. This highlights a key practical implication of our results: whenever possible, quantify uncertainty in the annotations and rescale the loss accordingly.

At the algorithmic threshold, we show in the agnostic setting that the classical LASSO recovery results from the homogeneous setting remain valid in the heterogeneous case and depend only on the total sample size $n_1 + n_2$. First, the threshold itself is independent of the individual noise levels. Second, the sufficient condition on the penalization coefficient involves the noise only through its average, exactly as if all observations had that average noise. Consequently, high-quality and low-quality samples contribute *equally* to the sample-size requirement for LASSO recovery. This reveals an unexpected difference in the effect of data heterogeneity on the information-theoretic and algorithmic thresholds for recovery.

Within the Gaussian design framework considered here, the informed information-theoretic threshold and the LASSO threshold are sharp, whereas the agnostic information-theoretic condition is sufficient but not proven tight.

In a broader discussion on how the information-theoretic and algorithmic thresholds interact across different problem settings, our result further emphasizes that the algorithmic threshold seems to be more "robust" to changes in the traditional problem settings (Gamarnik & Zadik, 2022; Wainwright, 2009). In fact, Wang et al. (2010) and Chaabouni & Gamarnik observed that when the noise is homogeneous but the design is sparse (i.e. $X_{ij}$ set to 0 uniformly at random) the information-theoretic threshold increases, while Omidiran & Wainwright (2008) showed that the algorithmic threshold remains the same and is unaffected by changes in the sparsity level of the data (although this was shown only for the sufficient condition, with no corresponding result on necessity).

Although we do not study LASSO recovery in the informed setting, this remains a promising direction for future work. It would be interesting to study the price of quality there, and compare it to LASSO recovery in the agnostic setting on one hand, and to the price of quality of information-theoretic recovery on the other.

ACKNOWLEDGMENTS

This work was supported by the National Science Foundation (NSF) under grant CISE-2233897. Youssef Chaabouni thanks Mehdi Makni, Marouane Nejjar, Panos Tsimpos, Malo Lahogue, and Alexandre Misrahi for insightful discussions and valuable feedback.

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

# A PROOF OF THEOREM 1

*Proof of Theorem 1.* We denote by $S^\star := \text{Supp}(\beta^\star)$. Let $\mathcal{S}_{p,s} := \{S \subset [p] : |S| = s\}$. We define the function:

$$L : \mathcal{S}_{p,s} \longrightarrow \mathbb{R}_{\geq 0}$$
$$S \longmapsto \|Y - X\mathbb{1}_S\|_2^2,$$

where $\mathbb{1}_S$ denotes the vector in $\{0,1\}^p$ such that $[\mathbb{1}_S]_j = \mathbb{1}(j \in S)$ for all $j \in [p]$. In particular, note from (8) that:

$$\hat{\beta} = \mathbb{1}_{\hat{S}}, \quad \text{where} \quad \hat{S} \in \underset{S \in \mathcal{S}_{p,s}}{\arg\min} L(S).$$

For every $S \in \mathcal{S}_{p,s}$, we define: $M(S) := |S \triangle S^\star|/2$, and let $U(S) := S^\star \setminus S$, $V(S) := S \setminus S^\star$. Note that, since $|S| = |S^\star| = s$, we have $|U(S)| = |V(S)| = M(S)$. We also define:

$$\Delta : \mathcal{S}_{p,s} \longrightarrow \mathbb{R}$$
$$S \longmapsto L(S) - L(S^\star).$$

**Proposition A.1.** *For any $S \in \mathcal{S}_{p,s}$: if $M(S) \geq \delta s$, then:*

$$\mathbb{P}(\Delta(S) \leq 0) \leq \left(1 + \frac{\delta(2\sigma_2^2 - \sigma_1^2)s}{2\sigma_2^2}\right)^{-n_1/2}\left(1 + \frac{\delta s}{2\sigma_2^2}\right)^{-n_2/2}.$$

*Proof.* See section A.1. $\qquad\square$

Hence we have, for any support $S \in \mathcal{S}_{p,s}$ such that $|S \triangle S^\star| \geq 2\delta s$:

$$\mathbb{P}\left(\|Y - X\mathbb{1}_S\|_2^2 \leq \|Y - X\mathbb{1}_{S^\star}\|_2^2\right) \leq \left(1 + \frac{\delta(2\sigma_2^2 - \sigma_1^2)s}{2\sigma_2^2}\right)^{-n_1/2}\left(1 + \frac{\delta s}{2\sigma_2^2}\right)^{-n_2/2} \quad (32)$$

Using (32) and a union bound over $\{S \in \mathcal{S}_{p,s} : |S \triangle S^\star| \geq 2\delta s\}$ we have:

$$\mathbb{P}_{X,Z}\left(\left|\text{Supp}(\hat{\beta}) \triangle \text{Supp}(\beta^\star)\right| < 2\delta s\right)$$
$$\geq \mathbb{P}_{X,Z}\left(\|Y - X\mathbb{1}_S\|_2^2 > \|Y - X\mathbb{1}_{S^\star}\|_2^2, \forall S \in \mathcal{S}_{p,s} : |S \triangle S^\star| \geq 2\delta s\right)$$
$$= 1 - \mathbb{P}_{X,Z}\left(\exists S \in \mathcal{S}_{p,s} : |S \triangle S^\star| \geq 2\delta s, \|Y - X\mathbb{1}_S\|_2^2 \leq \|Y - X\mathbb{1}_{S^\star}\|_2^2\right)$$
$$\overset{\text{U.B.}}{\geq} 1 - \sum_{S \in \mathcal{S}_{p,s} : |S \triangle S^\star| \geq 2\delta s} \mathbb{P}_{X,Z}\left(\|Y - X\mathbb{1}_S\|_2^2 \leq \|Y - X\mathbb{1}_{S^\star}\|_2^2\right)$$
$$\overset{(32)}{\geq} 1 - \sum_{S \in \mathcal{S}_{p,s} : |S \triangle S^\star| \geq 2\delta s} \left(1 + \frac{\delta(2\sigma_2^2 - \sigma_1^2)s}{2\sigma_2^2}\right)^{-n_1/2}\left(1 + \frac{\delta s}{2\sigma_2^2}\right)^{-n_2/2}$$
$$\geq 1 - |\mathcal{S}_{p,s}|\left(1 + \frac{\delta(2\sigma_2^2 - \sigma_1^2)s}{2\sigma_2^2}\right)^{-n_1/2}\left(1 + \frac{\delta s}{2\sigma_2^2}\right)^{-n_2/2}$$
$$= 1 - \binom{p}{s}\left(1 + \frac{\delta(2\sigma_2^2 - \sigma_1^2)s}{2\sigma_2^2}\right)^{-n_1/2}\left(1 + \frac{\delta s}{2\sigma_2^2}\right)^{-n_2/2}.$$

**Case 1: Assume $s = o(p)$.** We use the corollary of Stirling:

$$\binom{p}{s} = \exp\left(s\log(p/s)(1 + o(1))\right),$$

which yields:

$$\mathbb{P}_{X,Z}\left(\left|\text{Supp}\left(\hat{\beta}\right) \triangle \text{Supp}\left(\beta^\star\right)\right| < 2\delta s\right)$$

$$\geq 1 - \exp\left\{s \log\left(p/s\right)\left(1 + o\left(1\right)\right) - \frac{n_1}{2}\log\left(1 + \frac{\delta\left(2\sigma_2^2 - \sigma_1^2\right)s}{2\sigma_2^2}\right) - \frac{n_2}{2}\log\left(1 + \frac{\delta s}{2\sigma_2^2}\right)\right\}.$$

Now using (9) in above, we have:

$$\mathbb{P}_{X,Z}\left(\left|\text{Supp}\left(\hat{\beta}\right) \triangle \text{Supp}\left(\beta^\star\right)\right| < 2\delta s\right)$$
$$\geq 1 - \exp\left\{s \log\left(p/s\right)\left(1 + o\left(1\right)\right) - \left(1 + \varepsilon\right)s \log\left(p/s\right)\right\}$$
$$\geq 1 - \exp\left\{-\varepsilon s \log\left(p/s\right) - o\left(s \log\left(p/s\right)\right)\right\}.$$

Finally we conclude:

$$\mathbb{P}_{X,Z}\left(\left|\text{Supp}\left(\hat{\beta}\right) \triangle \text{Supp}\left(\beta^\star\right)\right| < 2\delta s\right) \geq 1 - \exp\left\{-\left(\varepsilon + o\left(1\right)\right)n^\star/2\right\} \overset{p \to +\infty}{\longrightarrow} 1,$$

where $n^\star = 2s \log\left(p/s\right)$.

**Case 2: Assume $s = \alpha\left(p\right)$ for some constant $\alpha \in \left(0, 1\right)$.** We use the corollary of Stirling:

$$\binom{p}{s} = \exp\left(h\left(\alpha\right)p\left(1 + o\left(1\right)\right)\right),$$

which yields:

$$\mathbb{P}_{X,Z}\left(\left|\text{Supp}\left(\hat{\beta}\right) \triangle \text{Supp}\left(\beta^\star\right)\right| < 2\delta s\right)$$

$$\geq 1 - \exp\left\{h\left(\alpha\right)p\left(1 + o\left(1\right)\right) - \frac{n_1}{2}\log\left(1 + \frac{\delta\left(2\sigma_2^2 - \sigma_1^2\right)s}{2\sigma_2^2}\right) - \frac{n_2}{2}\log\left(1 + \frac{\delta s}{2\sigma_2^2}\right)\right\}.$$

Now using (9) in above, we have:

$$\mathbb{P}_{X,Z}\left(\left|\text{Supp}\left(\hat{\beta}\right) \triangle \text{Supp}\left(\beta^\star\right)\right| < 2\delta s\right)$$
$$\geq 1 - \exp\left\{h\left(\alpha\right)p\left(1 + o\left(1\right)\right) - \left(1 + \varepsilon\right)h\left(\alpha\right)p\right\}$$
$$\geq 1 - \exp\left\{-\varepsilon h\left(\alpha\right)p - o\left(p\right)\right\}.$$

Finally we conclude:

$$\mathbb{P}_{X,Z}\left(\left|\text{Supp}\left(\hat{\beta}\right) \triangle \text{Supp}\left(\beta^\star\right)\right| < 2\delta s\right) \geq 1 - \exp\left\{-\left(\varepsilon + o\left(1\right)\right)n^\star/2\right\} \overset{p \to +\infty}{\longrightarrow} 1,$$

where $n^\star = 2h\left(\alpha\right)p$. $\qquad\square$

### A.1 PROOF OF PROPOSITION A.1

*Proof of Proposition A.1.* Fix $S \in \mathcal{S}_{p,s}$ such that $M\left(S\right) \geq \delta s$. We have:

$$\Delta\left(S\right) = L\left(S\right) - L\left(S^\star\right)$$
$$= \|Y - X\mathbb{1}_S\|_2^2 - \|Y - X\mathbb{1}_{S^\star}\|_2^2$$
$$= \|X\beta^\star + Z - X\mathbb{1}_S\|_2^2 - \|X\beta^\star + Z - X\mathbb{1}_{S^\star}\|_2^2$$
$$= \|X\left(\mathbb{1}_{S^\star} - \mathbb{1}_S\right)\|_2^2 + 2\langle Z, X\left(\mathbb{1}_{S^\star} - \mathbb{1}_S\right)\rangle$$
$$= \sum_{i=1}^{n}\langle X_i, \mathbb{1}_{S^\star} - \mathbb{1}_S\rangle^2 + 2\sum_{i=1}^{n}Z_i\langle X_i, \mathbb{1}_{S^\star} - \mathbb{1}_S\rangle.$$

Let $X_1 \in \mathbb{R}^{n_1 \times p}$, $X_2 \in \mathbb{R}^{n_2 \times p}$ such that:

$$X = \begin{pmatrix} X_1 \\ X_2 \end{pmatrix}.$$

Then the above expression of $\Delta(S)$ writes:

$$\Delta(s) = \sum_{i=1}^{n_1} \left\{ \langle X_i^1, \mathbb{1}_{S^\star} - \mathbb{1}_S \rangle^2 + 2Z_i^1 \langle X_i^1, \mathbb{1}_{S^\star} - \mathbb{1}_S \rangle \right\}$$
$$+ \sum_{i=1}^{n_2} \left\{ \langle X_i^2, \mathbb{1}_{S^\star} - \mathbb{1}_S \rangle^2 + 2Z_i^2 \langle X_i^2, \mathbb{1}_{S^\star} - \mathbb{1}_S \rangle \right\}.$$

We denote by $\left(\Delta_i^1\right)_{i\in[n_1]}$ and $\left(\Delta_i^2\right)_{i\in[n_2]}$ the terms of the sums above, that is:

$$\begin{cases} \Delta_i^1 := \langle X_i^1, \mathbb{1}_{S^\star} - \mathbb{1}_S \rangle^2 + 2Z_i^1 \langle X_i^1, \mathbb{1}_{S^\star} - \mathbb{1}_S \rangle \\ \Delta_i^2 := \langle X_i^2, \mathbb{1}_{S^\star} - \mathbb{1}_S \rangle^2 + 2Z_i^2 \langle X_i^2, \mathbb{1}_{S^\star} - \mathbb{1}_S \rangle \end{cases}.$$

Note that each of the elements of each of $\left\{\Delta_i^1 : i \in [n_1]\right\}$ and $\left\{\Delta_i^2 : i \in [n_2]\right\}$ are i.i.d. and:

$$\Delta(S) = \sum_{i=1}^{n_1} \Delta_i^1 + \sum_{i=1}^{n_2} \Delta_i^2.$$

Recalling the Chernoff bound:

$$\mathbb{P}(\Delta(S) \leq 0) = \mathbb{P}(-\Delta(S) \geq 0)$$
$$= \inf_{\theta \geq 0} \mathbb{P}\left(e^{-\theta\Delta(S)} \geq 1\right)$$
$$\text{(Markov's inequality)} \quad \leq \inf_{\theta \geq 0} \mathbb{E}\left[e^{-\theta\Delta(S)}\right]$$
$$= \inf_{\theta \geq 0} \mathbb{E}\left[e^{-\sum_{i=1}^{n_1}\theta\Delta_i^1 + \sum_{i=1}^{n_2}\theta\Delta_i^2}\right]$$
$$\stackrel{\text{ind.}}{=} \inf_{\theta \geq 0} \prod_{i=1}^{n_1} \mathbb{E}\left[e^{-\theta\Delta_i^1}\right] \prod_{i=1}^{n_2} \mathbb{E}\left[e^{-\theta\Delta_i^2}\right]$$
$$= \inf_{\theta \geq 0} \prod_{i=1}^{n_1} M_{-\Delta_i^1}(\theta) \prod_{i=1}^{n_2} M_{-\Delta_i^2}(\theta)$$
$$\stackrel{\text{i.d.}}{=} \inf_{\theta \geq 0} \left\{M_{-\Delta_i^1}(\theta)\right\}^{n_1} \left\{M_{-\Delta_i^2}(\theta)\right\}^{n_2}.$$

Therefore:
$$\mathbb{P}(\Delta(S) \leq 0) \leq \inf_{\theta \geq 0} \left\{M_{-\Delta_i^1}(\theta)\right\}^{n_1} \left\{M_{-\Delta_i^2}(\theta)\right\}^{n_2}. \tag{33}$$

Now we have, for any $\theta \geq 0$:

$$M_{-\Delta_i^1}(\theta) = \mathbb{E}_{X_i^1, Z_i^1}\left[e^{-\theta\left[\langle X_i^1, \beta^\star - \beta(S)\rangle^2 + 2Z_i^1\langle X_i^1, \beta^\star - \beta(S)\rangle\right]}\right]$$
$$= \mathbb{E}_{X_i^1}\left[e^{-\theta\langle X_i^1, \beta^\star - \beta(S)\rangle^2} \mathbb{E}_{Z_i^1}\left[e^{-2\theta Z_i^1\langle X_i^1, \beta^\star - \beta(S)\rangle}\big|X_i^1\right]\right]$$
$$= \mathbb{E}_{X_i^1}\left[e^{-\theta\langle X_i^1, \beta^\star - \beta(S)\rangle^2} M_{Z_i^1|X_i^1}\left(-2\theta\langle X_i^1, \beta^\star - \beta(S)\rangle\right)\right]$$
$$= \mathbb{E}_{X_i^1}\left[e^{-\theta\langle X_i^1, \beta^\star - \beta(S)\rangle^2} e^{\frac{1}{2}\left(-2\theta\langle X_i^1, \beta^\star - \beta(S)\rangle\right)^2\sigma_1^2}\right]$$
$$= \mathbb{E}_{X_i}\left[e^{\left(-\theta + 2\theta^2\sigma_1^2\right)\langle X_i^1, \beta^\star - \beta(S)\rangle^2}\right]$$
$$= \mathbb{E}_{X_i}\left[e^{\left(-\theta + 2\theta^2\sigma_1^2\right)\left(\sum_{j\in U} X_{ij}^1 - \sum_{j\in V} X_{ij}^1\right)^2}\right],$$

where we write $U$ and $V$ for $U(S)$ and $V(S)$, respectively, for simplicity. We know that:

$$\sum_{j\in U} X_{ij}^1 - \sum_{j\in V} X_{ij}^1 \stackrel{d}{=} \sum_{j\in U\cup V} X_{ij}^1 \sim \mathcal{N}(0, |U \cup V|).$$

Hence:

$$M_{-\Delta_i^1}(\theta) = \mathbb{E}_{X_i}\left[e^{\left(-\theta + 2\theta^2\sigma_1^2\right)\left(\sum_{j\in U} X_{ij}^1 - \sum_{j\in V} X_{ij}^1\right)^2}\right]$$

$$= \mathbb{E}_{X_i}\left[e^{|U\cup V|\left(-\theta + 2\theta^2\sigma_1^2\right)\Gamma}\right],$$

where:

$$\Gamma = \left(\frac{\sum_{j\in U} X_{ij}^1 - \sum_{j\in V} X_{ij}^1}{\sqrt{|U\cup V|}}\right)^2 \sim \chi^2(1).$$

Therefore:

$$M_{-\Delta_i^1}(\theta) = \begin{cases} \frac{1}{\sqrt{1-2|U\cup V|\left(-\theta+2\theta^2\sigma_1^2\right)}} & \text{if } |U\cup V|\left(-\theta+2\theta^2\sigma_1^2\right) < 1/2 \\ +\infty & \text{else.} \end{cases} \tag{34}$$

Similarly to (34), we obtain the following expression for $M_{-\Delta_i^2}(\theta)$:

$$M_{-\Delta_i^2}(\theta) = \begin{cases} \frac{1}{\sqrt{1-2|U\cup V|\left(-\theta+2\theta^2\sigma_2^2\right)}} & \text{if } |U\cup V|\left(-\theta+2\theta^2\sigma_2^2\right) < 1/2 \\ +\infty & \text{else.} \end{cases} \tag{35}$$

Therefore, for any $\theta \geq 0$:

$$M_{-\Delta_i^1}(\theta)^{n_1} M_{-\Delta_i^2}(\theta)^{n_2} = \begin{cases} \left(1 - 2|U\cup V|\left(-\theta+2\theta^2\sigma_1^2\right)\right)^{-n_1/2} \\ \qquad \times \left(1 - 2|U\cup V|\left(-\theta+2\theta^2\sigma_2^2\right)\right)^{-n_2/2} \\ \qquad\qquad \text{if } \begin{cases} |U\cup V|\left(-\theta+2\theta^2\sigma_1^2\right) < 1/2 \\ |U\cup V|\left(-\theta+2\theta^2\sigma_2^2\right) < 1/2 \end{cases} \\ +\infty \qquad \text{else.} \end{cases}$$

**Remark A.1** (Best Chernoff bound). To find the best Chernoff bound (33), we need to solve the optimization problem in (33), defined by:

$$\inf_{\theta\geq 0}\left\{M_{-\Delta_i^1}(\theta)\right\}^{n_1}\left\{M_{-\Delta_i^2}(\theta)\right\}^{n_2}. \tag{36}$$

Using the First Order Optimality Condition, (36) reduces to finding the roots of $\xi'(\theta) = 0$ is closed form, where $\xi(\cdot)$ is defined by:

$$\xi(\theta) := M_{-\Delta_i^1}(\theta)^{n_1} M_{-\Delta_i^2}(\theta)^{n_2}.$$

Using the closed-form solution of $\xi(\cdot)$ obtained above, we conclude that solving 36 reduces to finding the positive roots of the following third-degree polynomial in $\theta$:

$$n_1\left(4\sigma_1^2\theta - 1\right)\left(1 - 2|U\cup V|\left(-\theta+2\theta^2\sigma_2^2\right)\right) + n_2\left(4\sigma_2^2\theta - 1\right)\left(1 - 2|U\cup V|\left(-\theta+2\theta^2\sigma_1^2\right)\right). \tag{37}$$

To the best of our knowledge, this doesn't lead to any "reasonable" closed-form expression for the minimizer $\theta_{\min}^\star$. Instead, we note that:

$$\left\{\theta\colon |U\cup V|\left(-\theta+2\theta^2\sigma_2^2\right) < 1/2\right\} \subseteq \left\{\theta\colon |U\cup V|\left(-\theta+2\theta^2\sigma_1^2\right) < 1/2\right\},$$

and choose $\theta$ to the middle of the LHS interval.

In particular, setting $\theta^\star := \frac{1}{4\sigma_2^2}$, we have:

$$|U\cup V|\left(-\theta^\star + 2\theta^{\star 2}\sigma_2^2\right) = -\theta^\star|U\cup V|\left(-1 + 2\theta^\star\sigma_2^2\right) = -\theta^\star|U\cup V|/2 < 0 < 1/2,$$

and:

$$|U\cup V|\left(-\theta^\star + 2\theta^{\star 2}\sigma_1^2\right) = -\theta^\star|U\cup V|\left(-1 + \frac{\sigma_1^2}{2\sigma_2^2}\right) < 0 < 1/2 \quad (\text{since } \sigma_1^2 < \sigma_2^2).$$

Therefore:

$$M_{-\Delta_i^1}(\theta^\star)^{n_1} M_{-\Delta_i^2}(\theta^\star)^{n_2}$$

$$= \left(1 - 2|U \cup V|\left(-\theta^\star + 2\theta^{\star 2}\sigma_1^2\right)\right)^{-n_1/2} \left(1 - 2|U \cup V|\left(-\theta^\star + 2\theta^{\star 2}\sigma_2^2\right)\right)^{-n_2/2}$$

$$= \left(1 - 2|U \cup V|\left(-\frac{1}{4\sigma_2^2} + 2\left(\frac{1}{4\sigma_2^2}\right)^2 \sigma_1^2\right)\right)^{-n_1/2}$$

$$\times \left(1 - 2|U \cup V|\left(-\frac{1}{4\sigma_2^2} + 2\left(\frac{1}{4\sigma_2^2}\right)^2 \sigma_2^2\right)\right)^{-n_2/2}$$

$$= \left(1 + |U \cup V|\left(\frac{2\sigma_2^2 - \sigma_1^2}{4\sigma_2^4}\right)\right)^{-n_1/2} \left(1 + \frac{|U \cup V|}{4\sigma_2^2}\right)^{-n_2/2}$$

$$\leq \left(1 + \frac{\delta\left(2\sigma_2^2 - \sigma_1^2\right)s}{2\sigma_2^4}\right)^{-n_1/2} \left(1 + \frac{\delta s}{2\sigma_2^2}\right)^{-n_2/2},$$

where the last inequality holds because $|U \cup V| = 2M(S) \geq 2\delta s$. Finally, using this in (33) we conclude:

$$\mathbb{P}\left(\Delta(S) \leq 0\right) \leq \left(1 + \frac{\delta\left(2\sigma_2^2 - \sigma_1^2\right)s}{2\sigma_2^4}\right)^{-n_1/2} \left(1 + \frac{\delta s}{2\sigma_2^2}\right)^{-n_2/2}.$$

$\square$

# B  PROOF OF (15)

*Proof of (15).* Let $\beta \in \mathcal{B}_{p,s}$. We know from (1) that:

$$Y \mid X, \beta \sim \mathcal{N}\left(X\beta, \Sigma^2\right).$$

Its pdf writes:

$$f_{Y \mid X, \beta}(y) = (2\pi)^{-n/2} \det(\Sigma)^{-1} \exp\left\{-(y - X\beta)^T \Sigma^{-2}(y - X\beta)\right\}.$$

The MLE is defined as:

$$\hat{\beta}_{\text{MLE}} = \arg\max_{\beta \in \mathcal{B}_{p,s}} f_{Y \mid X, \beta}(Y)$$

$$= \arg\min_{\beta \in \mathcal{B}_{p,s}} (Y - X\beta)^T \Sigma^{-2}(Y - X\beta)$$

$$= \arg\min_{\beta \in \mathcal{B}_{p,s}} (Y - X\beta)^T \left(\Sigma^{-1}\right)^T \Sigma^{-1}(Y - X\beta)$$

$$= \arg\min_{\beta \in \mathcal{B}_{p,s}} \left\|\Sigma^{-1}(Y - X\beta)\right\|_2^2.$$

$\square$

# C  PROOF OF THEOREM 2

*Proof of Theorem 2.* We denote by $S^\star := \text{Supp}(\beta^\star)$. Let $\mathcal{S}_{p,s} := \{S \subset [p]: |S| = s\}$. We define the $\Sigma$-rescaled loss:

$$L_\Sigma : \mathcal{S}_{p,s} \longrightarrow \mathbb{R}_{\geq 0}$$

$$S \longmapsto \left\|\Sigma^{-1}(Y - X\mathbb{1}_S)\right\|_2^2,$$

where $\mathbb{1}_S$ denote the vector in $\{0, 1\}^p$ such that $[\mathbb{1}_S]_j = \mathbb{1}(j \in S)$ for all $j \in [p]$. In particular, note from (15) that:

$$\hat{\beta}_{\text{MLE}} = \mathbb{1}_{\hat{S}_{\text{MLE}}}, \quad \text{where} \quad \hat{S}_{\text{MLE}} := \arg\min_{S \in \mathcal{S}_{p,s}} L_\Sigma(S).$$

For every $S \in \mathcal{S}_{p,s}$, we define: $M(S) := |S \triangle S^\star|/2$, and let $U(S) := S^\star \setminus S$, $V(S) := S \setminus S^\star$. Note that, since $|S| = |S^\star| = s$, we have $|U(S)| = |V(S)| = M(S)$. We also define:

$$\Delta:\ \mathcal{S}_{p,s} \longrightarrow \mathbb{R}$$
$$S \longmapsto L_\Sigma(S) - L_\Sigma(S^\star).$$

**Proposition C.1.** *For any $S \in \mathcal{S}_{p,s}$: if $M(S) \geq \delta s$, then:*

$$\mathbb{P}(\Delta(S) \leq 0) \leq \left(1 + \frac{\delta s}{2\sigma_1^2}\right)^{-n_1/2} \left(1 + \frac{\delta s}{2\sigma_2^2}\right)^{-n_2/2}.$$

*Proof.* See section C.1. □

Hence we have, for any support $S \in \mathcal{S}_{p,s}$ such that $|S \triangle S^\star| \geq 2\delta s$:

$$\mathbb{P}\left(\left\|\Sigma^{-1}(Y - X\mathbb{1}_S)\right\|_2^2 \leq \left\|\Sigma^{-1}(Y - X\mathbb{1}_{S^\star})\right\|_2^2\right) \leq \left(1 + \frac{\delta s}{2\sigma_1^2}\right)^{-n_1/2} \left(1 + \frac{\delta s}{2\sigma_2^2}\right)^{-n_2/2}. \quad (38)$$

Using (38) and a union bound over $\{S \in \mathcal{S}_{p,s}:\ |S \triangle S^\star| \geq 2\delta s\}$ we have:

$$\mathbb{P}_{X,Z}\left(\left|\text{Supp}(\hat{\beta}) \triangle \text{Supp}(\beta^\star)\right| < 2\delta s\right)$$
$$\geq \mathbb{P}_{X,Z}\left(\left\|\Sigma^{-1}(Y - X\mathbb{1}_S)\right\|_2^2 > \left\|\Sigma^{-1}(Y - X\mathbb{1}_{S^\star})\right\|_2^2,\ \forall S \in \mathcal{S}_{p,s}:\ |S \triangle S^\star| \geq 2\delta s\right)$$
$$= 1 - \mathbb{P}_{X,Z}\left(\exists S \in \mathcal{S}_{p,s}:\ |S \triangle S^\star| \geq 2\delta s,\ \left\|\Sigma^{-1}(Y - X\mathbb{1}_S)\right\|_2^2 \leq \left\|\Sigma^{-1}(Y - X\mathbb{1}_{S^\star})\right\|_2^2\right)$$
$$\overset{\text{U.B.}}{\geq} 1 - \sum_{S \in \mathcal{S}_{p,s}:\ |S \triangle S^\star| \geq 2\delta s} \mathbb{P}_{X,Z}\left(\left\|\Sigma^{-1}(Y - X\mathbb{1}_S)\right\|_2^2 \leq \left\|\Sigma^{-1}(Y - X\mathbb{1}_{S^\star})\right\|_2^2\right)$$
$$\overset{(38)}{\geq} 1 - \sum_{S \in \mathcal{S}_{p,s}:\ |S \triangle S^\star| \geq 2\delta s} \left(1 + \frac{\delta s}{2\sigma_1^2}\right)^{-n_1/2} \left(1 + \frac{\delta s}{2\sigma_2^2}\right)^{-n_2/2}$$
$$\geq 1 - |\mathcal{S}_{p,s}| \left(1 + \frac{\delta s}{2\sigma_1^2}\right)^{-n_1/2} \left(1 + \frac{\delta s}{2\sigma_2^2}\right)^{-n_2/2}$$
$$= 1 - \binom{p}{s} \left(1 + \frac{\delta s}{2\sigma_1^2}\right)^{-n_1/2} \left(1 + \frac{\delta s}{2\sigma_2^2}\right)^{-n_2/2}.$$

**Case 1: Assume $s = o(p)$.** We use the corollary of Stirling:

$$\binom{p}{s} = \exp\left(s \log(p/s)(1 + o(1))\right),$$

which yields:

$$\mathbb{P}_{X,Z}\left(\left|\text{Supp}(\hat{\beta}) \triangle \text{Supp}(\beta^\star)\right| < 2\delta s\right)$$
$$\geq 1 - \exp\left\{s \log(p/s)(1 + o(1)) - \frac{n_1}{2}\log\left(1 + \frac{\delta s}{2\sigma_1^2}\right) - \frac{n_2}{2}\log\left(1 + \frac{\delta s}{2\sigma_2^2}\right)\right\}.$$

Now using (16) in above, we have:

$$\mathbb{P}_{X,Z}\left(\left|\text{Supp}(\hat{\beta}) \triangle \text{Supp}(\beta^\star)\right| < 2\delta s\right)$$
$$\geq 1 - \exp\left\{s \log(p/s)(1 + o(1)) - (1 + \varepsilon)s \log(p/s)\right\}$$
$$\geq 1 - \exp\left\{-\varepsilon s \log(p/s) - o(s \log(p/s))\right\}.$$

Finally we conclude:

$$\mathbb{P}_{X,Z}\left(\left|\text{Supp}(\hat{\beta}) \triangle \text{Supp}(\beta^\star)\right| < 2\delta s\right) \geq 1 - \exp\left\{-(\varepsilon + o(1))n^\star/2\right\} \overset{p \to +\infty}{\longrightarrow} 1,$$

where $n^\star = 2s \log(p/s)$.

**Case 2: Assume $s = h(\alpha) p$, for some constant $\alpha \in (0, 1)$.** We use the corollary of Stirling:

$$\binom{p}{s} = \exp\left(h(\alpha) p(1 + o(1))\right),$$

which yields:

$$\mathbb{P}_{X,Z}\left(\left|\text{Supp}\left(\hat{\beta}\right) \triangle \text{Supp}\left(\beta^\star\right)\right| < 2\delta s\right)$$
$$\geq 1 - \exp\left\{h(\alpha) p(1 + o(1)) - \frac{n_1}{2} \log\left(1 + \frac{\delta s}{2\sigma_1^2}\right) - \frac{n_2}{2} \log\left(1 + \frac{\delta s}{2\sigma_2^2}\right)\right\}.$$

Now using (16) in above, we have:

$$\mathbb{P}_{X,Z}\left(\left|\text{Supp}\left(\hat{\beta}\right) \triangle \text{Supp}\left(\beta^\star\right)\right| < 2\delta s\right)$$
$$\geq 1 - \exp\left\{h(\alpha) p(1 + o(1)) - (1 + \varepsilon) h(\alpha) p\right\}$$
$$\geq 1 - \exp\left\{-\varepsilon h(\alpha) p - o(p)\right\}.$$

Finally we conclude:

$$\mathbb{P}_{X,Z}\left(\left|\text{Supp}\left(\hat{\beta}\right) \triangle \text{Supp}\left(\beta^\star\right)\right| < 2\delta s\right) \geq 1 - \exp\left\{-(\varepsilon + o(1)) n^\star/2\right\} \xrightarrow{p \to +\infty} 1,$$

where $n^\star = 2h(\alpha) p$. $\qquad\square$

### C.1 PROOF OF PROPOSITION C.1

*Proof of Proposition C.1.* Fix $S \in \mathcal{S}_{p,s}$ such that $M(S) \geq \delta s$. We have:

$$\Delta(S) = L_\Sigma(S) - L_\Sigma(S^\star)$$
$$= \left\|\Sigma^{-1}(Y - X\mathbb{1}_S)\right\|_2^2 - \left\|\Sigma^{-1}(Y - X\mathbb{1}_{S^\star})\right\|_2^2$$
$$= \left\|\Sigma^{-1}(X\beta^\star + Z - X\mathbb{1}_S)\right\|_2^2 - \left\|\Sigma^{-1}(X\beta^\star + Z - X\mathbb{1}_{S^\star})\right\|_2^2$$
$$= \left\|\Sigma^{-1}X(\mathbb{1}_{S^\star} - \mathbb{1}_S)\right\|_2^2 + 2\left\langle\Sigma^{-1}Z, \Sigma^{-1}X(\mathbb{1}_{S^\star} - \mathbb{1}_S)\right\rangle$$
$$= \sum_{i=1}^{n_1} \frac{1}{\sigma_1^2}\langle X_i, \mathbb{1}_{S^\star} - \mathbb{1}_S\rangle^2 + \sum_{i=n_1+1}^{n_2} \frac{1}{\sigma_2^2}\langle X_i, \mathbb{1}_{S^\star} - \mathbb{1}_S\rangle^2$$
$$+ 2\sum_{i=1}^{n_1} \frac{1}{\sigma_1^2}Z_i\langle X_i, \mathbb{1}_{S^\star} - \mathbb{1}_S\rangle + 2\sum_{i=n_1+1}^{n_2} \frac{1}{\sigma_2^2}Z_i\langle X_i, \mathbb{1}_{S^\star} - \mathbb{1}_S\rangle.$$

Let $X_1 \in \mathbb{R}^{n_1 \times p}$, $X_2 \in \mathbb{R}^{n_2 \times p}$ such that:

$$X = \begin{pmatrix} X_1 \\ X_2 \end{pmatrix}.$$

Then the above expression of $\Delta(S)$ writes:

$$\Delta(s) = \frac{1}{\sigma_1^2}\sum_{i=1}^{n_1}\left\{\langle X_i^1, \mathbb{1}_{S^\star} - \mathbb{1}_S\rangle^2 + 2Z_i^1\langle X_i^1, \mathbb{1}_{S^\star} - \mathbb{1}_S\rangle\right\}$$
$$+ \frac{1}{\sigma_2^2}\sum_{i=1}^{n_2}\left\{\langle X_i^2, \mathbb{1}_{S^\star} - \mathbb{1}_S\rangle^2 + 2Z_i^2\langle X_i^2, \mathbb{1}_{S^\star} - \mathbb{1}_S\rangle\right\}.$$

We denote by $\left(\Delta_i^1\right)_{i \in [n_1]}$ and $\left(\Delta_i^2\right)_{i \in [n_2]}$ the terms of the sums above, that is:

$$\begin{cases} \Delta_i^1 := \langle X_i^1, \mathbb{1}_{S^\star} - \mathbb{1}_S\rangle^2 + 2Z_i^1\langle X_i^1, \mathbb{1}_{S^\star} - \mathbb{1}_S\rangle \\ \Delta_i^2 := \langle X_i^2, \mathbb{1}_{S^\star} - \mathbb{1}_S\rangle^2 + 2Z_i^2\langle X_i^2, \mathbb{1}_{S^\star} - \mathbb{1}_S\rangle \end{cases}.$$

Note that each of the elements of each of $\{\Delta_i^1 : i \in [n_1]\}$ and $\{\Delta_i^2 : i \in [n_2]\}$ are i.i.d. and:

$$\Delta\left(S\right) = \frac{1}{\sigma_1^2}\sum_{i=1}^{n_1}\Delta_i^1 + \frac{1}{\sigma_2^2}\sum_{i=1}^{n_2}\Delta_i^2.$$

Using the Chernoff bound, we have:

$$\begin{aligned}
\mathbb{P}\left(\Delta\left(S\right) \le 0\right) &= \mathbb{P}\left(-\Delta\left(S\right) \ge 0\right) \\
&= \inf_{\theta \ge 0}\mathbb{P}\left(e^{-\theta\Delta(S)} \ge 1\right) \\
&\le \inf_{\theta \ge 0}\mathbb{E}\left[e^{-\theta\Delta(S)}\right] \\
&= \inf_{\theta \ge 0}\mathbb{E}\left[e^{-\frac{1}{\sigma_1^2}\sum_{i=1}^{n_1}\theta\Delta_i^1 + \frac{1}{\sigma_2^2}\sum_{i=1}^{n_2}\theta\Delta_i^2}\right] \\
&\stackrel{\text{ind.}}{=} \inf_{\theta \ge 0}\prod_{i=1}^{n_1}\mathbb{E}\left[e^{-\theta\Delta_i^1/\sigma_1^2}\right]\prod_{i=1}^{n_2}\mathbb{E}\left[e^{-\theta\Delta_i^2/\sigma_2^2}\right] \\
&= \inf_{\theta \ge 0}\prod_{i=1}^{n_1}M_{-\Delta_i^1}\left(\frac{\theta}{\sigma_1^2}\right)\prod_{i=1}^{n_2}M_{-\Delta_i^2}\left(\frac{\theta}{\sigma_2^2}\right) \\
&\stackrel{\text{i.d.}}{=} \inf_{\theta \ge 0}\left\{M_{-\Delta_i^1}\left(\frac{\theta}{\sigma_1^2}\right)\right\}^{n_1}\left\{M_{-\Delta_i^2}\left(\frac{\theta}{\sigma_1^2}\right)\right\}^{n_2}.
\end{aligned}$$

Therefore:

$$\mathbb{P}\left(\Delta\left(S\right) \le 0\right) \le \inf_{\theta \ge 0}\left\{M_{-\Delta_i^1}\left(\frac{\theta}{\sigma_1^2}\right)\right\}^{n_1}\left\{M_{-\Delta_i^2}\left(\frac{\theta}{\sigma_2^2}\right)\right\}^{n_2}. \tag{39}$$

Now we have, for any $\theta \ge 0$:

$$\begin{aligned}
M_{-\Delta_i^1}(\theta/\sigma_1^2) &= \mathbb{E}_{X_i^1, Z_i^1}\left[e^{-\theta\left[\langle X_i^1, \beta^\star - \beta(S)\rangle^2 + 2Z_i^1\langle X_i^1, \beta^\star - \beta(S)\rangle\right]/\sigma_1^2}\right] \\
&= \mathbb{E}_{X_i^1}\left[e^{-\theta\langle X_i^1, \beta^\star - \beta(S)\rangle^2/\sigma_1^2}\mathbb{E}_{Z_i^1}\left[e^{-2\theta Z_i^1\langle X_i^1, \beta^\star - \beta(S)\rangle/\sigma_1^2}\Big|X_i^1\right]\right] \\
&= \mathbb{E}_{X_i^1}\left[e^{-\theta\langle X_i^1, \beta^\star - \beta(S)\rangle^2/\sigma_1^2}M_{Z_i^1|X_i^1}\left(-2\theta\langle X_i^1, \beta^\star - \beta(S)\rangle/\sigma_1^2\right)\right] \\
&= \mathbb{E}_{X_i^1}\left[e^{-\theta\langle X_i^1, \beta^\star - \beta(S)\rangle^2/\sigma_1^2}e^{\frac{1}{2}\left(-2\theta\langle X_i^1, \beta^\star - \beta(S)\rangle/\sigma_1^2\right)^2\sigma_1^2}\right] \\
&= \mathbb{E}_{X_i}\left[e^{\left(-\theta + 2\sigma_1^2\right)\langle X_i^1, \beta^\star - \beta(S)\rangle^2/\sigma_1^2}\right] \\
&= \mathbb{E}_{X_i}\left[e^{\left(-\theta + 2\theta^2\right)\left(\sum_{j\in U}X_{ij}^1 - \sum_{j\in V}X_{ij}^1\right)^2/\sigma_1^2}\right],
\end{aligned}$$

where we write $U$ and $V$ for $U\left(S\right)$ and $V\left(S\right)$, respectively, for simplicity. We know that:

$$\sum_{j\in U}X_{ij}^1 - \sum_{j\in V}X_{ij}^1 \stackrel{d}{=} \sum_{j\in U\cup V}X_{ij}^1 \sim \mathcal{N}\left(0, |U\cup V|\right).$$

Hence:

$$\begin{aligned}
M_{-\Delta_i^1}(\theta) &= \mathbb{E}_{X_i}\left[e^{\left(-\theta + 2\theta^2\right)\left(\sum_{j\in U}X_{ij}^1 - \sum_{j\in V}X_{ij}^1\right)^2/\sigma_1^2}\right] \\
&= \mathbb{E}_{X_i}\left[e^{|U\cup V|\left(-\theta + 2\theta^2\right)\Gamma/\sigma_1^2}\right],
\end{aligned}$$

where:

$$\Gamma = \left(\frac{\sum_{j\in U}X_{ij}^1 - \sum_{j\in V}X_{ij}^1}{\sqrt{|U\cup V|}}\right)^2 \sim \chi^2\left(1\right).$$

Therefore:

$$M_{-\Delta_i^1}(\theta/\sigma_1^2) = \begin{cases} \frac{1}{\sqrt{1 - 2|U\cup V|\left(-\theta + 2\theta^2\right)/\sigma_1^2}} & \text{if } |U\cup V|\left(-\theta + 2\theta^2\right)/\sigma_1^2 < 1/2 \\ +\infty & \text{else.} \end{cases} \tag{40}$$

Similarly to (40), we obtain the following expression for $M_{-\Delta_i^2}(\theta/\sigma_2^2)$:

$$M_{-\Delta_i^2}(\theta/\sigma_2^2) = \begin{cases} \frac{1}{\sqrt{1-2|U\cup V|(-\theta+2\theta^2)/\sigma_2^2}} & \text{if } |U \cup V|\left(-\theta + 2\theta^2\right)/\sigma_2^2 < 1/2 \\ +\infty & \text{else.} \end{cases} \qquad (41)$$

From above, we clearly have:

$$\underset{\theta \geq 0}{\arg\min}\, M_{-\Delta_i^1}(\theta/\sigma_1^2) = \underset{\theta \geq 0}{\arg\min}\, M_{-\Delta_i^2}(\theta/\sigma_2^2) = \underset{\theta \geq 0}{\arg\min}\, \left\{-\theta + 2\theta^2\right\} = \frac{1}{4},$$

and, taking $\theta^\star := 1/4$ we have:

$$M_{-\Delta_i^1}(\theta/\sigma_1^2) = \frac{1}{\sqrt{1 + \frac{|U \cup V|}{4\sigma_1^2}}} \quad \text{and} \quad M_{-\Delta_i^2}(\theta/\sigma_2^2) = \frac{1}{\sqrt{1 + \frac{|U \cup V|}{4\sigma_2^2}}}.$$

Therefore:

$$\inf_{\theta \geq 0} \left\{M_{-\Delta_i^1}\left(\frac{\theta}{\sigma_1^2}\right)\right\}^{n_1} \left\{M_{-\Delta_i^2}\left(\frac{\theta}{\sigma_2^2}\right)\right\}^{n_2} = \left\{M_{-\Delta_i^1}\left(\frac{\theta^\star}{\sigma_1^2}\right)\right\}^{n_1} \left\{M_{-\Delta_i^2}\left(\frac{\theta^\star}{\sigma_2^2}\right)\right\}^{n_2}$$

$$= \left\{\frac{1}{\sqrt{1 + \frac{|U\cup V|}{4\sigma_1^2}}}\right\}^{n_1} \left\{\frac{1}{\sqrt{1 + \frac{|U\cup V|}{4\sigma_2^2}}}\right\}^{n_2}$$

Therefore:

$$\inf_{\theta \geq 0} \left\{M_{-\Delta_i^1}\left(\frac{\theta}{\sigma_1^2}\right)\right\}^{n_1} \left\{M_{-\Delta_i^2}\left(\frac{\theta}{\sigma_2^2}\right)\right\}^{n_2} = \left(1 + \frac{|U \cup V|}{4\sigma_1^2}\right)^{-n_1/2} \left(1 + \frac{|U \cup V|}{4\sigma_2^2}\right)^{-n_2/2}. \qquad (42)$$

Since $|U \cup V| = 2M(S) \geq 2\delta s$, the above yields:

$$\inf_{\theta \geq 0} \left\{M_{-\Delta_i^1}\left(\frac{\theta}{\sigma_1^2}\right)\right\}^{n_1} \left\{M_{-\Delta_i^2}\left(\frac{\theta}{\sigma_2^2}\right)\right\}^{n_2} \leq \left(1 + \frac{\delta s}{2\sigma_1^2}\right)^{-n_1/2} \left(1 + \frac{\delta s}{2\sigma_2^2}\right)^{-n_2/2}.$$

Finally, using this in (39) we conclude:

$$\mathbb{P}\left(\Delta(S) \leq 0\right) \leq \left(1 + \frac{\delta s}{2\sigma_1^2}\right)^{-n_1/2} \left(1 + \frac{\delta s}{2\sigma_2^2}\right)^{-n_2/2}.$$

$\square$

## D  PROOF OF THEOREM 3

*Proof of Theorem 3.* We define $\Sigma \in \mathbb{R}^{n \times n}$ and $W$ random vector in $\mathbb{R}^n$ such that:

$$Z = \Sigma W,$$

where:

$$\Sigma = \begin{pmatrix} \sigma_1 I_{n_1} & 0 \\ 0 & \sigma_2 I_{n_2} \end{pmatrix}, \quad W \sim \mathcal{N}(0, I_n).$$

Let $S := \mathrm{Supp}(\beta^\star)$ and $S^c := [p]\backslash S$. The following proposition characterizes the recovery property in a more tractable way that will help us in the proof.

**Proposition D.1.** *Assume that the matrix $X_S^T X_S$ is invertible. Then, for any given $\lambda_p > 0$ and noise $\Sigma w \in \mathbb{R}^n$ we have:*

$$\mathcal{R}(X, \beta^\star, \Sigma w, \lambda_p) \iff \begin{cases} \left|\left(\frac{1}{n}X_S^T X_S\right)^{-1}\left(\frac{1}{n}X_S^T \Sigma w - \lambda_p\, \mathrm{sign}(\beta_S^\star)\right)\right| < |\beta_S^\star| \\ \left|X_{S^c}^T X_S\left(X_S^T X_S\right)^{-1}\left(\frac{1}{n}X_S^T \Sigma w - \lambda_p \mathrm{sign}(\beta_S^\star)\right) - \frac{1}{n}X_{S^c}^T \Sigma w\right| \leq \lambda_p \end{cases}$$

*where the absolute values and inequalities are taken component-wise.*

*Proof.* The equivalence follows from the First Order Optimality Condition of the LASSO (24). It was used in the proof of the LASSO threshold (Wainwright, 2009) and previously by Fuchs (2004); Meinshausen & Bühlmann (2006); Tropp (2006); Zhao & Yu (2006). See appendix D.1 for the complete proof. ☐

Let $\vec{b} := \mathrm{sign}\left(\beta^\star\right)$. We define:

$$U_i := e_i^T \left(\frac{1}{n}X_S^T X_S\right)^{-1} \left[\frac{1}{n}X_S^T \Sigma W - \lambda_p \vec{b}\right], \tag{43}$$

and

$$V_j := X_j^T \left\{X_S \left(X_S^T X_S\right)^{-1} \lambda_p \vec{b} - \left[X_S \left(X_S^T X_S\right)^{-1} X_S^T - I_n\right]\frac{\Sigma W}{n}\right\}, \tag{44}$$

for all $i \in S$ and $j \in S^c$. Let $\rho := \min_{i \in S} |\beta_i^\star|$. Note that:

$$\max_{i \in S} |U_i| < \rho \implies \left|\left(\frac{1}{n}X_S^T X_S\right)^{-1}\left(\frac{1}{n}X_S^T \Sigma w - \lambda_p\, \mathrm{sign}\left(\beta_S^\star\right)\right)\right| < |\beta_S^\star|, \tag{45}$$

and

$$\max_{j \in S^c} |V_j| \le \lambda_p \iff \left|X_{S^c}^T X_S \left(X_S^T X_S\right)^{-1}\left(\frac{1}{n}X_S^T \Sigma w - \lambda_p\, \mathrm{sign}\left(\beta_S^\star\right)\right) - \frac{1}{n}X_{S^c}^T \Sigma w\right| \le \lambda_p. \tag{46}$$

In addition, note that when $s < n$, $X_S$ is full-rank a.s., and hence $X_S^T X_S$ is invertible a.s. Therefore, the equivalence in Proposition 3.1 holds a.s. The proof of Theorem 3 relies of the two following propositions:

**Proposition D.2.**

*i. Under the sample size condition (26) we have:*

$$\mathbb{P}\left(\max_{j \in S^c} |V_j| \le \lambda_p\right) \overset{p \to +\infty}{\longrightarrow} 0.$$

*ii. Under the sample size condition (27) and the regularization condition (28), we have:*

$$\mathbb{P}\left(\max_{j \in S^c} |V_j| \le \lambda_p\right) \overset{p \to +\infty}{\longrightarrow} 1.$$

*Proof.* See appendix D.2. ☐

**Proposition D.3.** *Under the sample size condition (27) and the regularization condition (28), we have:*

$$\mathbb{P}\left(\max_{i \in S} |U_i| < \rho\right) \overset{p \to +\infty}{\longrightarrow} 1.$$

*Proof.* See appendix D.3. ☐

Necessity: assume (26) holds. Then we conclude by Proposition D.2 [*i*] and (46).

Sufficiency: assume (27) and (28) hold. Then we conclude by Proposition D.2 [*ii*], Proposition D.3 and (45), (46). ☐

## D.1 PROOF OF PROPOSITION D.1

*Proof of Proposition D.1.* Let $\hat{\beta} \in \mathbb{R}^p$. By First Order Optimality Condition of the LASSO, $\hat{\beta}$ is optimal if and only if the exits $z \in \mathbb{R}^p$ such that:

$$\begin{cases} z \in \partial \ell_1 \left( \hat{\beta} \right) = \left\{ z \in \mathbb{R}^p \colon z_i = \text{sign} \left( \hat{\beta}_i \right) \ \text{ for } \hat{\beta}_i \neq 0, \ |z_i| \leq 1 \ \text{ otherwise} \right\}, \\ \frac{1}{n} X^T \left( X\hat{\beta} - Y \right) + \lambda_p z = 0. \end{cases}$$

The condition above can be equivalently written as:

$$\begin{cases} z_S = \text{sign} \left( \hat{\beta}_S \right), \\ |z_{S^c}| \leq 1, \\ \frac{1}{n} X^T X \hat{\beta} - \frac{1}{n} X^T Y + \lambda_p z = 0. \end{cases}$$

Substituting $Y = X\beta^\star + \Sigma w$, the condition writes:

$$\begin{cases} z_S = \text{sign} \left( \hat{\beta}_S \right), \\ |z_{S^c}| \leq 1, \\ \frac{1}{n} X^T X \left( \hat{\beta} - \beta^\star \right) - \frac{1}{n} X^T \Sigma w + \lambda_p z = 0. \end{cases}$$

Splitting on $S$ and $S^c$ we get:

$$\begin{cases} z_S = \text{sign} \left( \hat{\beta}_S \right), \\ |z_{S^c}| \leq 1, \\ \frac{1}{n} X_S^T X \left( \hat{\beta} - \beta^\star \right) - \frac{1}{n} X_S^T \Sigma w + \lambda_p \ \text{sign} \left( \hat{\beta}_S \right) = 0, \\ \frac{1}{n} X_{S^c}^T X \left( \hat{\beta} - \beta^\star \right) - \frac{1}{n} X_{S^c}^T \Sigma w + \lambda_p z_{S^c} = 0. \end{cases}$$

Now the LASSO recovers the support of $\beta^\star$ if and only if there exits $\hat{\beta} \in \mathbb{R}^p$ such that $\text{sign} \left( \hat{\beta} \right) = \text{sign} \left( \beta^\star \right)$ and:

$$\exists \, z \in \mathbb{R}^p \ \text{such that} \ \begin{cases} z_S = \text{sign} \left( \hat{\beta}_S \right), \\ |z_{S^c}| \leq 1, \\ \frac{1}{n} X_S^T X \left( \hat{\beta} - \beta^\star \right) - \frac{1}{n} X_S^T \Sigma w + \lambda_p \ \text{sign} \left( \hat{\beta}_S \right) = 0, \\ \frac{1}{n} X_{S^c}^T X \left( \hat{\beta} - \beta^\star \right) - \frac{1}{n} X_{S^c}^T \Sigma w + \lambda_p z_{S^c} = 0. \end{cases}$$

Which is equivalent to:

$$\exists \, z, \hat{\beta} \in \mathbb{R}^p \ \text{such that} \ \begin{cases} \text{sign} \left( \hat{\beta} \right) = \text{sign} \left( \beta^\star \right), \\ z_S = \text{sign} \left( \hat{\beta}_S \right), \\ |z_{S^c}| \leq 1, \\ \frac{1}{n} X_S^T X \left( \hat{\beta} - \beta^\star \right) - \frac{1}{n} X_S^T \Sigma w + \lambda_p \ \text{sign} \left( \hat{\beta}_S \right) = 0, \\ \frac{1}{n} X_{S^c}^T X \left( \hat{\beta} - \beta^\star \right) - \frac{1}{n} X_{S^c}^T \Sigma w + \lambda_p z_{S^c} = 0. \end{cases}$$

which is equivalent to

$$\exists \, z, \hat{\beta} \in \mathbb{R}^p \ \text{such that} \ \begin{cases} \text{sign} \left( \hat{\beta} \right) = \text{sign} \left( \beta^\star \right), \\ z_S = \text{sign} \left( \beta_S^\star \right), \\ |z_{S^c}| \leq 1, \\ \frac{1}{n} X_S^T X_S \left( \hat{\beta}_S - \beta_S^\star \right) - \frac{1}{n} X_S^T \Sigma w + \lambda_p \ \text{sign} \left( \beta_S^\star \right) = 0, \\ \frac{1}{n} X_{S^c}^T X_S \left( \hat{\beta}_S - \beta_S^\star \right) - \frac{1}{n} X_{S^c}^T \Sigma w + \lambda_p z_{S^c} = 0. \end{cases}$$

which is equivalent to

$$\exists\, z, \hat{\beta} \in \mathbb{R}^p \text{ such that } \begin{cases} \hat{\beta}_{S^c} = 0, \\ z_S = \text{sign}\left(\hat{\beta}_S\right) = \text{sign}\left(\beta_S^\star\right) \neq 0, \\ |z_{S^c}| \leq 1, \\ \hat{\beta}_S = \beta_S^\star + \left(\frac{1}{n}X_S^T X_S\right)^{-1}\left(\frac{1}{n}X_S^T \Sigma w - \lambda_p \, \text{sign}\left(\beta_S^\star\right)\right), \\ \left|\frac{1}{n}X_{S^c}^T X_S\left(\hat{\beta}_S - \beta_S^\star\right) - \frac{1}{n}X_{S^c}^T \Sigma w\right| \leq \lambda_p. \end{cases}$$

which is equivalent to

$$\begin{cases} \left|\left(\frac{1}{n}X_S^T X_S\right)^{-1}\left(\frac{1}{n}X_S^T \Sigma w - \lambda_p \, \text{sign}\left(\beta_S^\star\right)\right)\right| < |\beta_S^\star|, \\ \left|X_{S^c}^T X_S\left(X_S^T X_S\right)^{-1}\left(\frac{1}{n}X_S^T \Sigma w - \lambda_p \, \text{sign}\left(\beta_S^\star\right)\right) - \frac{1}{n}X_{S^c}^T \Sigma w\right| \leq \lambda_p. \end{cases}$$

$\square$

## D.2 PROOF OF PROPOSITION D.2

### D.2.1 PRELIMINARY RESULTS

**Lemma D.1** (Moments of $(V \mid X_S, W)$). *Conditionally on $X_S$ and $W$, $V$ is Gaussian vector. In addition, we have:*

$$\mathbb{E}\left[V \mid X_S, W\right] = 0,$$

*and:*

$$\text{Cov}\left[V \mid X_S, W\right] = M_p I_{|S^c|},$$

*where:*

$$M_p := \left\|X_S\left(X_S^T X_S\right)^{-1}\lambda_p \vec{b} + \left[I_n - X_S\left(X_S^T X_S\right)^{-1}X_S^T\right]\frac{\Sigma W}{n}\right\|_2^2,$$

$$= \lambda_p^2 \vec{b}^T\left(X_S^T X_S\right)^{-1}\vec{b} + \frac{1}{n^2}W^T \Sigma\left[I_n - X_S\left(X_S^T X_S\right)^{-1}X_S^T\right]\Sigma W.$$

*Proof.* See section D.2.4. $\square$

**Lemma D.2** (Bounding the second moment of $(V \mid X_S, W)$). *We have:*

$$\mathbb{E}\left[M_p\right] = \frac{\lambda_p^2}{n - s - 1}\|b\|_2^2 + \frac{(n - s)\left(n_1 \sigma_1^2 + n_2 \sigma_2^2\right)}{n^3}.$$

*In addition, for any constant $\delta > 0$, we have:*

$$\mathbb{P}\left(|M_p - \mathbb{E}\left[M_p\right]| \geq \delta \mathbb{E}\left[M_p\right]\right) \to 0,$$

*as $p \to +\infty$.*

*Proof.* See Section D.2.5. $\square$

### D.2.2 SHOWING THAT $\mathbb{P}\left(\max_{j \in S^c} |V_j| \leq \lambda_p\right) \longrightarrow 0$:

*Proof of Proposition D.2, part (i.)* . Let:

$$T(\delta) := \left\{|M_p - \mathbb{E}\left[M_p\right]| \geq \delta \mathbb{E}\left[M_p\right]\right\}.$$

We have, by total probability:

$$\mathbb{P}\left(\max_{j \in S^c} |V_j| \leq \lambda_p\right) \leq \mathbb{P}\left(\max_{j \in S^c} |V_j| \leq \lambda_p \,\big|\, T(\delta)^c\right) + \mathbb{P}\left(T(\delta)\right). \tag{47}$$

Conditioning on $X_S$ and $W$:

$$\mathbb{P}\left(\max_{j \in S^c} |V_j| \leq \lambda_p \,\big|\, T(\delta)^c\right) = \mathbb{E}\left[\mathbb{P}\left(\max_{j \in S^c} |V_j| \leq \lambda_p \,\big|\, X_S, W\right)\big|\, T(\delta)^c\right].$$

Note that, conditionally on $(X_S, W)$, we have $V_j \overset{\text{i.i.d.}}{\sim} \mathcal{N}(0, M_p)$, for $j \in S^c$. We know the bound on expectation of Gaussian maxima (see Theorem 5.3.1 in (De Haan & Ferreira, 2006)):

$$\mathbb{E}\left[\max_{j \in S^c} V_j \mid X_S, W\right] = \sqrt{2 \log(p-s) M_p} (1 + o(1)),$$

Conditionally on $T(\delta)^c$, we have:

$$M_p \geq (1 - \delta) \mathbb{E}[M_p].$$

Hence, conditionally on $T(\delta)^c$:

$$\frac{1}{\lambda_p} \mathbb{E}\left[\max_{j \in S^c} |V_j| \mid X_S, W\right]$$

$$\geq \frac{1}{\lambda_p} \sqrt{2 \log(p-s)(1-\delta) \mathbb{E}[M_p]} (1 + o(1))$$

$$\geq (1 + o(1)) \frac{\sqrt{1-\delta}}{\lambda_p} \sqrt{\frac{2\lambda_p^2 s \log(p-s)}{n-s-1} + \frac{2(n-s)(n_1\sigma_1^2 + n_2\sigma_2^2) \log(p-s)}{n^3}}$$

$$= (1 + o(1)) \sqrt{1-\delta} \sqrt{\frac{2s \log(p-s)}{n-s-1} + \frac{2(n-s)(n_1\sigma_1^2 + n_2\sigma_2^2) \log(p-s)}{\lambda_p^2 n^3}}.$$

We consider two cases, depending on the asymptotic behavior of $\frac{n_1\sigma_1^2 + n_2\sigma_2^2}{\lambda_p^2 n^2}$:

- Case 1: $\lim_{p \to +\infty} \frac{n_1\sigma_1^2 + n_2\sigma_2^2}{\lambda_p^2 n^2} = 0$.

- Case 2: $\lim_{p \to +\infty} \frac{n_1\sigma_1^2 + n_2\sigma_2^2}{\lambda_p^2 n^2} > 0$.

**Case 1:** Assume $\lim_{p \to +\infty} \frac{n_1\sigma_1^2 + n_2\sigma_2^2}{\lambda_p^2 n^2} = 0$. By the above inequality, we have:

$$\frac{1}{\lambda_p} \mathbb{E}\left[\max_{j \in S^c} |V_j| \mid X_S, W\right] \geq (1 + o(1)) \sqrt{1-\delta} \sqrt{\frac{2s \log(p-s)}{n-s-1}}.$$

Using condition (26), the above gives:

$$\frac{1}{\lambda_p} \mathbb{E}\left[\max_{j \in S^c} |V_j| \mid X_S, W\right] \geq (1 + o(1)) \sqrt{1-\delta} \sqrt{\frac{2s \log(p-s)}{n-s-1}}$$

$$\geq (1 + o(1)) \sqrt{1-\delta} \sqrt{\frac{2s \log(p-s)}{(1-\varepsilon)(2s \log(p-s) + s + 1) - s - 1}}$$

$$= (1 + o(1)) \sqrt{1-\delta} \sqrt{\frac{2s \log(p-s)}{2s(1-\varepsilon) \log(p-s) - \varepsilon(s+1)}}$$

$$\geq (1 + o(1)) \sqrt{1-\delta} \sqrt{\frac{2s \log(p-s)}{2s(1-\varepsilon) \log(p-s)}}$$

$$= (1 + o(1)) \sqrt{\frac{1-\delta}{1-\varepsilon}}.$$

Taking the liminf as $p \to +\infty$ we get, conditionally on $T(\delta)^c$:

$$\liminf_{n \to +\infty} \frac{1}{\lambda_p} \mathbb{E}\left[\max_{j \in S^c} |V_j| \mid X_S, W\right] \geq \sqrt{\frac{1-\delta}{1-\varepsilon}}.$$

Therefore, for $n$ large enough we have:

$$\frac{1}{\lambda_p} \mathbb{E}\left[\max_{j \in S^c} |V_j| \mid X_S, W\right] \geq \frac{1}{2}\left(1 + \sqrt{\frac{1-\delta}{1-\varepsilon}}\right) = \frac{1}{2} + \frac{1}{2}\sqrt{\frac{1-\delta}{1-\varepsilon}}.$$

Taking $\delta := \varepsilon/2$, we get:

$$\frac{1}{\lambda_p} \mathbb{E}\left[\max_{j \in S^c} |V_j| \;\middle|\; X_S, W\right] \geq \frac{1}{2} + \frac{1}{2}\sqrt{\frac{1 - \varepsilon/2}{1 - \varepsilon}} =: \kappa > 1.$$

Next, we use the following the result on concentration of Gaussian maxima:

**Lemma D.3** (Concentration of Gaussian maxima). *Let $k \in \mathbb{N}$ and $(N_i)_{i=1}^{i=k} \overset{i.i.d.}{\sim} \mathcal{N}\left(0, \tau^2\right)$. Then for any $\eta > 0$, we have:*

$$\begin{cases} \mathbb{P}\left(\max_{i \in [k]} N_i - \mathbb{E}\left[\max_{i \in [k]} N_i\right] > \eta\right) \leq \exp\left(-\frac{\eta^2}{2\tau^2}\right), \\ \mathbb{P}\left(\max_{i \in [k]} N_i - \mathbb{E}\left[\max_{i \in [k]} N_i\right] < -\eta\right) \leq \exp\left(-\frac{\eta^2}{2\tau^2}\right). \end{cases}$$

*Proof.* See appendix D.2.6. □

Using Lemma D.3 gives, for all $\eta > 0$:

$$\mathbb{P}\left(\max_{j \in S^c} |V_j| < \mathbb{E}\left[\max_{j \in S^c} |V_j|\right] - \eta\right) \leq \exp\left(-\frac{\eta^2}{2\left(1 + \delta\right)\mathbb{E}\left[M_p\right]}\right).$$

Setting $\eta := \left(\kappa - 1\right)\lambda_p/2$, we get:

$$\begin{aligned} \mathbb{P}\left(\max_{j \in S^c} |V_j| \leq \lambda_p \;\middle|\; T\left(\delta\right)^c\right) &\leq \mathbb{P}\left(\max_{j \in S^c} |V_j| < \left(\kappa + 1\right)\lambda_p/2 \;\middle|\; T\left(\delta\right)^c\right) \\ &\leq \mathbb{P}\left(\max_{j \in S^c} |V_j| < \mathbb{E}\left[\max_{j \in S^c} |V_j|\right] - \left(\kappa - 1\right)\lambda_p/2 \;\middle|\; T\left(\delta\right)^c\right) \\ &\leq \exp\left(-\frac{\left(\kappa - 1\right)^2 \lambda_p^2}{4\left(2 + \varepsilon\right)\mathbb{E}\left[M_p\right]}\right) \\ &= \exp\left(-\frac{\left(\kappa - 1\right)^2 \lambda_p^2}{4\left(2 + \varepsilon\right)\left(\frac{\lambda_p^2 s}{n - s - 1} + \frac{\left(n - s\right)\left(n_1 \sigma_1^2 + n_2 \sigma_2^2\right)}{n^3}\right)}\right) \\ &= \exp\left(-\frac{\left(\kappa - 1\right)^2}{4\left(2 + \varepsilon\right)\left(\frac{s}{n - s - 1} + \frac{n - s}{n}\frac{n_1 \sigma_1^2 + n_2 \sigma_2^2}{\lambda_p^2 n^2}\right)}\right). \end{aligned}$$

Now note that, because $\lim_{p \to +\infty} \frac{n_1 \sigma_1^2 + n_2 \sigma_2^2}{\lambda_p^2 n^2} = 0$, the above RHS goes to 0 as $p \to +\infty$. Hence, we get:

$$\mathbb{P}\left(\max_{j \in S^c} |V_j| \leq \lambda_p \;\middle|\; T\left(\delta\right)^c\right) \overset{p \to +\infty}{\longrightarrow} 0.$$

Taking the limit in (47) and using the fact that $\mathbb{P}\left(T\left(\delta\right)\right) \overset{p \to +\infty}{\longrightarrow} 0$, we conclude:

$$\lim_{p \to +\infty} \mathbb{P}\left(\max_{j \in S^c} |V_j| \leq \lambda_p\right) = 0.$$

**Case 2:** Assume $\lim_{p \to +\infty} \frac{n_1 \sigma_1^2 + n_2 \sigma_2^2}{\lambda_p^2 n^2} > 0$. Note that this could be $+\infty$. We have:

$$\frac{1}{\lambda_p} \mathbb{E}\left[\max_{j \in S^c} |V_j| \mid X_S, W\right]$$

$$\geq (1 + o(1)) \sqrt{1 - \delta} \sqrt{\frac{2s \log(p - s)}{n - s - 1} + \frac{2(n - s)(n_1 \sigma_1^2 + n_2 \sigma_2^2) \log(p - s)}{\lambda_p^2 n^3}}$$

$$\geq (1 + o(1)) \sqrt{1 - \delta} \sqrt{\frac{2(n - s)(n_1 \sigma_1^2 + n_2 \sigma_2^2) \log(p - s)}{\lambda_p^2 n^3}}$$

$$= (1 + o(1)) \sqrt{1 - \delta} \sqrt{2 \frac{n - s}{n} \frac{n_1 \sigma_1^2 + n_2 \sigma_2^2}{\lambda_p^2 n^2}} \sqrt{\log(p - s)}$$

$$\xrightarrow{p \to +\infty} +\infty.$$

Therefore, for $n$ large enough, we have:

$$\mathbb{E}\left[\max_{j \in S^c} |V_j| \mid X_S, W\right] \geq 4\lambda_p.$$

Now using Lemma D.3 on concentration of Gaussian maxima, we have for all $\eta > 0$:

$$\mathbb{P}\left(\max_{j \in S^c} |V_j| < \mathbb{E}\left[\max_{j \in S^c} |V_j|\right] - \eta\right) \leq \exp\left(-\frac{\eta^2}{2(1 + \delta) \mathbb{E}[M_p]}\right).$$

Fixing $\eta := \mathbb{E}[\max_{j \in S^c} |V_j|]/2$ and $\delta := \varepsilon/2$ we get, for $n$ large enough:

$$\mathbb{P}\left(\max_{j \in S^c} |V_j| \leq \lambda_p \,\Big|\, T(\delta)^c\right) \leq \mathbb{P}\left(\max_{j \in S^c} |V_j| < 2\lambda_p \,\Big|\, T(\delta)^c\right)$$

$$\leq \mathbb{P}\left(\max_{j \in S^c} |V_j| < \frac{1}{2} \mathbb{E}\left[\max_{j \in S^c} |V_j|\right] \,\Big|\, T(\delta)^c\right)$$

$$= \mathbb{P}\left(\max_{j \in S^c} |V_j| < \mathbb{E}\left[\max_{j \in S^c} |V_j|\right] - \frac{1}{2}\mathbb{E}\left[\max_{j \in S^c} |V_j|\right] \,\Big|\, T(\delta)^c\right)$$

$$\leq \exp\left(-\frac{\mathbb{E}[\max_{j \in S^c} |V_j|]^2}{4(2 + \varepsilon) \mathbb{E}[M_p]}\right)$$

$$= \exp\left(-\frac{\mathbb{E}[\max_{j \in S^c} |V_j|]^2 / \lambda_p^2}{4(2 + \varepsilon) \mathbb{E}[M_p] / \lambda_p^2}\right)$$

$$= \exp\left(-\frac{\mathbb{E}[\max_{j \in S^c} |V_j|]^2 / \lambda_p^2}{4(2 + \varepsilon)\left(\frac{\lambda_p^2 s}{n - s - 1} + \frac{(n - s)(n_1 \sigma_1^2 + n_2 \sigma_2^2)}{n^3}\right) / \lambda_p^2}\right)$$

$$= \exp\left(-\frac{\left(\mathbb{E}[\max_{j \in S^c} |V_j|] / \lambda_p\right)^2}{4(2 + \varepsilon)\left(\frac{s}{n - s - 1} + \frac{n - s}{n} \frac{n_1 \sigma_1^2 + n_2 \sigma_2^2}{\lambda_p^2 n^2}\right)}\right).$$

Since:

$$\frac{1}{\lambda_p} \mathbb{E}\left[\max_{j \in S^c} |V_j| \mid X_S, W\right] \geq (1 + o(1)) \sqrt{1 - \delta} \sqrt{2 \frac{n - s}{n} \frac{n_1 \sigma_1^2 + n_2 \sigma_2^2}{\lambda_p^2 n^2}} \sqrt{\log(p - s)},$$

the above yields:

$$\mathbb{P}\left(\max_{j \in S^c} |V_j| \leq \lambda_p \,\Big|\, T(\delta)^c\right) \leq \exp\left(-\frac{2(1+o(1))(1-\delta)\left(\frac{n-s}{n}\frac{n_1\sigma_1^2+n_2\sigma_2^2}{\lambda_p^2 n^2}\right)\log(p-s)}{4(2+\varepsilon)\left(\frac{s}{n-s-1}+\frac{n-s}{n}\frac{n_1\sigma_1^2+n_2\sigma_2^2}{\lambda_p^2 n^2}\right)}\right)$$

$$= \exp\left(-\frac{2(1+o(1))(1-\delta)\left(\frac{n-s}{n}\frac{n_1\sigma_1^2+n_2\sigma_2^2}{\lambda_p^2 n^2}\right)\log(p-s)}{4(2+\varepsilon)(1+o(1))\left(\frac{n-s}{n}\frac{n_1\sigma_1^2+n_2\sigma_2^2}{\lambda_p^2 n^2}\right)}\right)$$

$$= \exp\left(-\frac{(1-\delta)\log(p-s)}{2(2+\varepsilon)}(1+o(1))\right)$$

$$\overset{p\to+\infty}{\longrightarrow} 0.$$

Hence, we get:

$$\mathbb{P}\left(\max_{j \in S^c} |V_j| \leq \lambda_p \,\Big|\, T(\delta)^c\right) \overset{p\to+\infty}{\longrightarrow} 0.$$

Taking the limit in (47) and using the fact that $\mathbb{P}\big(T(\delta)\big) \overset{p\to+\infty}{\longrightarrow} 0$, we conclude:

$$\lim_{p\to+\infty} \mathbb{P}\left(\max_{j \in S^c} |V_j| \leq \lambda_p\right) = 0.$$

$\square$

### D.2.3 Showing that $\mathbb{P}\left(\max_{j \in S^c} |V_j| \leq \lambda_p\right) \longrightarrow 1$:

*Proof of Proposition D.2, part (ii.)* . Let:

$$T(\delta) := \{|M_p - \mathbb{E}[M_p]| \geq \delta \mathbb{E}[M_p]\}.$$

We have, by total probability:

$$\mathbb{P}\left(\max_{j \in S^c} |V_j| > \lambda_p\right) \leq \mathbb{P}\left(\max_{j \in S^c} |V_j| > \lambda_p \,\Big|\, T(\delta)^c\right) + \mathbb{P}(T(\delta)). \tag{48}$$

Conditioning on $X_S$ and $W$:

$$\mathbb{P}\left(\max_{j \in S^c} |V_j| > \lambda_p \,\Big|\, T(\delta)^c\right) = \mathbb{E}\left[\mathbb{P}\left(\max_{j \in S^c} |V_j| > \lambda_p \,\Big|\, X_S, W\right) \,\Big|\, T(\delta)^c\right]$$

Note that, conditionally on $(X_S, W)$, we have $V_j \overset{\text{i.i.d.}}{\sim} \mathcal{N}(0, M_p)$, for $j \in S^c$. Using the bound on expectation of Gaussian maxima (see Theorem 5.3.1 in (De Haan & Ferreira, 2006)):

$$\mathbb{E}\left[\max_{j \in S^c} |V_j| \,\Big|\, X_S, W\right] \leq \sqrt{2\log(2(p-s))M_p}.$$

Conditionally on $T(\delta)^c$, we have:

$$M_p \leq (1+\delta)\mathbb{E}[M_p].$$

Hence, conditionally on $T(\delta)^c$:

$$\frac{1}{\lambda_p}\mathbb{E}\left[\max_{j \in S^c} |V_j| \,\Big|\, X_S, W\right]$$

$$\leq \frac{1}{\lambda_p}\sqrt{2\log(2(p-s))(1+\delta)\mathbb{E}[M_p]}$$

$$= \frac{1}{\lambda_p}\sqrt{2\log(2(p-s))(1+\delta)\left(\frac{\lambda_p^2 s}{n-s-1} + \frac{(n-s)(n_1\sigma_1^2+n_2\sigma_2^2)}{n^3}\right)}$$

$$= \sqrt{2+2\delta}\sqrt{\frac{\log(2(p-s))s}{n-s-1} + \frac{\log(2(p-s))(n-s)(n_1\sigma_1^2+n_2\sigma_2^2)}{\lambda_p^2 n^3}}$$

$$= \sqrt{2+2\delta}$$

$$\times \sqrt{\frac{s\log 2}{n-s-1} + \frac{s\log(p-s)}{n-s-1} + \frac{n-s}{n}\left(\frac{(n_1\sigma_1^2+n_2\sigma_2^2)\log(p-s)}{\lambda_p^2 n^2} + \frac{(n_1\sigma_1^2+n_2\sigma_2^2)\log 2}{\lambda_p^2 n^2}\right)}.$$

Taking the limsup as $p \to +\infty$ and using conditions (27) and (28) we get, conditionally on $T(\delta)^c$:

$$\limsup_{p \to +\infty} \frac{1}{\lambda_p} \mathbb{E}\left[\max_{j \in S^c} |V_j| \,\middle|\, X_S, W\right]$$

$$\leq \limsup_{p \to +\infty} \sqrt{(1+\delta)\left(\frac{2s\log(p-s)}{(1+\varepsilon)(2s\log(p-s)+s+1)-s-1}\right)}$$

$$\leq \limsup_{p \to +\infty} \sqrt{(1+\delta)\left(\frac{2s\log(p-s)}{(1+\varepsilon)(2s\log(p-s)+s+1)-s-1}\right)}$$

$$\leq \sqrt{\frac{1+\delta}{1+\varepsilon}}.$$

Fix $\delta := \varepsilon/4$. By the above, we know that for large enough $n$, we have:

$$\mathbb{E}\left[\max_{j \in S^c} |V_j| \,\middle|\, X_S, W\right] \leq \lambda_p \sqrt{\frac{1+\varepsilon/2}{1+\varepsilon}}.$$

For simplicity of notation, set $\upsilon := 1 - \sqrt{\frac{1+\varepsilon/2}{1+\varepsilon}} > 0$. In addition, we by Lemma D.3 on concentration of Gaussian maxima, for all $\eta > 0$:

$$\mathbb{P}\left(\max_{j \in S^c} |V_j| > \eta + \mathbb{E}\left[\max_{j \in S^c} |V_j|\right] \,\middle|\, X_S, W\right) \leq \exp\left(-\frac{\eta^2}{2(1+\delta)\mathbb{E}[M_p]}\right).$$

Let $\eta := \upsilon \lambda_p$. Then we get, for $n$ large enough:

$$\mathbb{P}\left(\max_{j \in S^c} |V_j| > \lambda_p \,\middle|\, X_S, W\right) \leq \mathbb{P}\left(\max_{j \in S^c} |V_j| > \eta + \mathbb{E}\left[\max_{j \in S^c} |V_j|\right] \,\middle|\, X_S, W\right)$$

$$\leq \exp\left(-\frac{\eta^2}{2(1+\delta)\mathbb{E}[M_p]}\right)$$

$$= \exp\left(-\frac{\upsilon^2 \lambda_p^2}{2(1+\varepsilon/2)\left(\frac{\lambda_p^2 s}{n-s-1} + \frac{(n-s)(n_1\sigma_1^2 + n_2\sigma_2^2)}{n^3}\right)}\right)$$

$$= \exp\left(-\frac{\upsilon^2}{2(1+\varepsilon/2)\left(\frac{s}{n-s-1} + \frac{n-s}{n}\frac{n_1\sigma_1^2 + n_2\sigma_2^2}{\lambda_p^2 n^2}\right)}\right).$$

Substituting in the above, we get:

$$\mathbb{P}\left(\max_{j \in S^c} |V_j| > \lambda_p \,\middle|\, T(\delta)^c\right) = \mathbb{E}\left[\mathbb{P}\left(\max_{j \in S^c} |V_j| > \lambda_p \,\middle|\, X_S, W\right) \,\middle|\, T(\delta)^c\right]$$

$$\leq \mathbb{E}\left[\exp\left(-\frac{\upsilon^2}{2(1+\varepsilon/2)\left(\frac{s}{n-s-1} + \frac{n-s}{n}\frac{n_1\sigma_1^2 + n_2\sigma_2^2}{\lambda_p^2 n^2}\right)}\right) \,\middle|\, T(\delta)^c\right]$$

$$= \exp\left(-\frac{\upsilon^2}{2(1+\varepsilon/2)\left(\frac{s}{n-s-1} + \frac{n-s}{n}\frac{n_1\sigma_1^2 + n_2\sigma_2^2}{\lambda_p^2 n^2}\right)}\right)$$

$$\xrightarrow{p \to +\infty} 0,$$

since, by condition (28), we have:

$$0 < \frac{n_1\sigma_1^2 + n_2\sigma_2^2}{\lambda_p^2 n^2} \leq \frac{(n_1\sigma_1^2 + n_2\sigma_2^2)\log(p-s)}{\lambda_p^2 n^2} \xrightarrow{p \to +\infty} 0.$$

Taking the limit in (48) and using the fact that $\mathbb{P}(T(\delta)) \to 0$, we conclude:

$$\mathbb{P}\left(\max_{j \in S^c} |V_j| \leq \lambda_p\right) \xrightarrow{p \to +\infty} 1.$$

$\square$

### D.2.4 PROOF OF LEMMA D.1

*Proof of Lemma D.1.* Recall from (44) that:

$$V_j = \left\{ X_S \left( X_S^T X_S \right)^{-1} \lambda_p \vec{b} - \left[ X_S \left( X_S^T X_S \right)^{-1} X_S^T - I_n \right] \frac{\Sigma W}{n} \right\}^T X_j,$$

for all $j \in S^c$. Conditionally on $X_S$ and $W$, the first term of the RHS above is constant and $(X_j)_{j \in S^c} \overset{\text{i.i.d.}}{\sim} \mathcal{N}(0, I_n)$. Therefore:

$$V_j \mid X_S, W \sim \mathcal{N}\left( 0, A^T A \right),$$

where:

$$A = \left\{ X_S \left( X_S^T X_S \right)^{-1} \lambda_p \vec{b} - \left[ X_S \left( X_S^T X_S \right)^{-1} X_S^T - I_n \right] \frac{\Sigma W}{n} \right\} \in \mathbb{R}^n.$$

Expanding the expression of $A^T A$, we get:

$$A^T A = \lambda_p^2 \vec{b}^T \left( X_S^T X_S \right)^{-1} \vec{b} + \frac{1}{n^2} W^T \Sigma^T \left[ I_n - X_S \left( X_S^T X_S \right)^{-1} X_S^T \right]^T \Sigma W.$$

In addition, we have:

$$\text{Cov}\left( X_{j_1}, X_{j_2} \right) = \delta_{j_1 j_2} I_n,$$

therefore:

$$\text{Cov}\left( V_{j_1}, V_{j_2} \right) = \delta_{j_1 j_2} A^T A.$$

We conclude:

$$V \mid X_S, W \sim \mathcal{N}\left( 0, M_p I_{|S^c|} \right),$$

where $M_p := A^T A = \|A\|_2^2$. $\qquad\square$

### D.2.5 PROOF OF LEMMA D.2

*Proof of Lemma D.2.* Recall from Lemma D.1 that:

$$M_p = \lambda_p^2 \vec{b}^T \left( X_S^T X_S \right)^{-1} \vec{b} + \frac{1}{n^2} W^T \Sigma \left[ I_n - X_S \left( X_S^T X_S \right)^{-1} X_S^T \right] \Sigma W.$$

By expectation of inverse Wishart matrices, we have:

$$\mathbb{E}\left[ \lambda_p^2 \vec{b}^T \left( X_S^T X_S \right)^{-1} \vec{b} \right] = \frac{\lambda_p^2}{n - s - 1} \|b\|_2^2.$$

By Gram-Schmidt decomposition of $X_S$ (Meckes, 2019), we write:

$$X_S = QR \in \mathbb{R}^{n \times s}, \tag{49}$$

with $R_{ii} > 0$ and $Q^T Q = I_s$, where $R \in \mathbb{R}^{s \times s}$ is upper triangular (hence invertible) $Q \in \mathbb{R}^{n \times s}$ corresponds to $s$ columns of a $n \times n$ matrix of Haar distribution over the orthogonal group $O(n)$, which we define as follows:

$$U = [P \quad Q] \sim \text{Haar on } O(n).$$

Then we have:

$$\begin{aligned}
X_S \left( X_S^T X_S \right)^{-1} X_S^T &= QR \left( R^T Q^T Q R \right)^{-1} R^T Q^T \\
&= QR \left( R^T R \right)^{-1} R^T Q^T \\
&= QR R^{-1} \left( R^T \right)^{-1} R^T Q^T \\
&= QQ^T.
\end{aligned}$$

Therefore:

$$I_n - X_S \left( X_S^T X_S \right)^{-1} X_S^T = I_n - QQ^T = PP^T = [P \quad Q] \begin{bmatrix} I_{n-s} & 0_{(n-s)\times s} \\ 0_{s\times(n-s)} & 0_{s\times s} \end{bmatrix} \begin{bmatrix} P \\ Q \end{bmatrix} = UDU^T,$$

where:
$$D = \begin{bmatrix} I_{n-s} & 0_{(n-s)\times s} \\ 0_{s\times(n-s)} & 0_{s\times s} \end{bmatrix}.$$

Note that unlike $U$, $D$ is deterministic. Since $W \sim \mathcal{N}(0, I_n)$, we have:

$$\mathbb{E}\left[\frac{1}{n^2} W^T \Sigma \left[I_n - X_S \left(X_S^T X_S\right)^{-1} X_S^T\right] \Sigma W \mid X_S\right] = \frac{1}{n^2} \operatorname{tr}\left\{\Sigma \left[I_n - X_S \left(X_S^T X_S\right)^{-1} X_S^T\right] \Sigma\right\}$$

$$= \frac{1}{n^2} \operatorname{tr}\left(\Sigma^T U D U^T \Sigma\right).$$

Hence, by total expectation:

$$\mathbb{E}\left[\frac{1}{n^2} W^T \Sigma \left[I_n - X_S \left(X_S^T X_S\right)^{-1} X_S^T\right] \Sigma W\right] = \mathbb{E}_{X_S}\left[\frac{1}{n^2} \operatorname{tr}\left(\Sigma^T U D U^T \Sigma\right)\right]$$

$$= \frac{1}{n^2} \operatorname{tr}\left(\mathbb{E}_{X_S}\left[\Sigma^T U D U^T \Sigma\right]\right)$$

$$= \frac{1}{n^2} \operatorname{tr}\left(\Sigma^T \mathbb{E}_U\left[U D U^T\right] \Sigma\right).$$

By properties of the Haar distribution (see Example 1.8 of (Gu, 2013)), we have:

$$\mathbb{E}_{U\sim\text{Haar}(n)}\left[U^T D U\right] = \frac{\operatorname{tr}(D)}{n} I_n.$$

Substituting in the above, we get:

$$\mathbb{E}\left[\frac{1}{n^2} W^T \Sigma \left[I_n - X_S \left(X_S^T X_S\right)^{-1} X_S^T\right] \Sigma W\right] = \frac{1}{n^2} \operatorname{tr}\left(\Sigma \mathbb{E}_U\left[U D U^T\right] \Sigma\right)$$

$$= \frac{1}{n^2} \operatorname{tr}\left(\frac{\operatorname{tr}(D)}{n} \Sigma I_n \Sigma\right)$$

$$= \frac{\operatorname{tr}(D)}{n^3} \operatorname{tr}\left(\Sigma^T I_n \Sigma\right)$$

$$= \frac{n-s}{n^3} \operatorname{tr}\left(\Sigma^T \Sigma\right)$$

$$= \frac{(n-s)\left(n_1\sigma_1^2 + n_2\sigma_2^2\right)}{n^3}.$$

Hence, we conclude:

$$\mathbb{E}[M_p] = \frac{\lambda_p^2}{n-s-1} \|b\|_2^2 + \frac{(n-s)\left(n_1\sigma_1^2 + n_2\sigma_2^2\right)}{n^3}.$$

We now compute $\operatorname{Var}(M_p)$. For simplicity, we write $\Lambda = I_n - X_S \left(X_S^T X_S\right)^{-1} X_S^T = U D U^T$, so that:

$$M_p = \lambda_p^2 \vec{b}^T \left(X_S^T X_S\right)^{-1} \vec{b} + \frac{1}{n^2} W^T \Sigma \Lambda \Sigma W.$$

Therefore:

$$M_p^2 = \lambda_p^4 \left[\vec{b}^T \left(X_S^T X_S\right)^{-1} \vec{b}\right]^2 + \frac{1}{n^4} \left(W^T \Sigma \Lambda \Sigma W\right)^2 + 2\frac{\lambda_p^2}{n^2} \left[\vec{b}^T \left(X_S^T X_S\right)^{-1} \vec{b}\right] \left(W^T \Sigma \Lambda \Sigma W\right).$$

Let:

$$T_1 := \left[\vec{b}^T \left(X_S^T X_S\right)^{-1} \vec{b}\right]^2, \quad T_2 := \left(W^T \Sigma \Lambda \Sigma W\right)^2, \quad T_3 := \left[\vec{b}^T \left(X_S^T X_S\right)^{-1} \vec{b}\right]\left(W^T \Sigma \Lambda \Sigma W\right),$$

so that:

$$M_p^2 = \lambda_p^4 T_1 + \frac{1}{n^4} T_2 + 2\frac{\lambda_p^2}{n^2} T_3.$$

We start by computing $\mathbb{E}[T_1]$. Recall that $X_S^T X_S \sim \mathcal{W}_s(I_s, n)$. We use the following result for second moments of inverse Wishart matrices from (Siskind, 1972).

**Lemma D.4** (Second moment of inverse Wishart matrices, (Siskind, 1972)). *Let $a, b \in \mathbb{N}$ with $a > b + 3$. Let $t \in \mathbb{R}^s$ and $M \in R^{s \times s}$ and $A \sim \mathcal{W}_a(T, b)$. Then:*

$$(b - a)(b - a - 3)\,\mathbb{E}\left[A^{-1}tt^T A^{-1}\right] = T^{-1}tt^T T^{-1} + T^{-1}\left(t^T T^{-1}t\right)/(b - a - 1).$$

Setting $b := n$; $a := s$; $t := \vec{b}$; $T := I_s$ and $A := X_S^T X_S$ in Lemma D.4, we get:

$$(n - s)(n - s - 3)\,\mathbb{E}\left[\left(X_S^T X_S\right)^{-1}\vec{b}\vec{b}^T\left(X_S^T X_S\right)^{-1}\right] = \vec{b}\vec{b}^T + \left\|\vec{b}\right\|_2^2 I_s/(n - s - 1). \tag{50}$$

Multiplying by the LHS by $\vec{b}^T$ on the left and by $\vec{b}$ on right we get:

$$\mathbb{E}\left[T_1\right] = \frac{1}{(n - s)(n - s - 3)}\left(1 + \frac{1}{n - s - 1}\right)\left(\vec{b}^T\vec{b}\right)^2.$$

Hence:

$$\mathbb{E}\left[T_1\right] = \frac{1}{(n - s)(n - s - 3)}\left(1 + \frac{1}{n - s - 1}\right)\left\|\vec{b}\right\|_2^4. \tag{51}$$

Now let us compute $\mathbb{E}\left[T_2\right]$. Since $W \sim \mathcal{N}(0, I_n)$, we have by moments of the multivariate normal distribution (see Theorem 5.2a and Theorem 5.2b in (Rencher & Schaalje, 2008)):

$$\mathbb{E}\left[T_2 \mid X_S\right] = 2\,\mathrm{tr}\left[\left(\Sigma\Lambda\Sigma\right)^2\right] + \left[\mathrm{tr}\left(\Sigma\Lambda\Sigma\right)\right]^2.$$

**Lemma D.5.** *We have:*

$$\mathbb{E}\left\{\left[\mathrm{tr}\left(\Sigma\Lambda\Sigma\right)\right]^2\right\}$$
$$= \left(n_1\sigma_1^2 + n_2\sigma_2^2\right)^2\left\{\frac{(n + 1)(n - s)}{(n - 1)\,n\,(n + 2)}\right\}\left\{n - s - \frac{2}{n + 1}\right\}$$
$$+ \left(n_1\sigma_1^4 + n_2\sigma_2^4\right)\left\{\frac{(n - s)(n - s + 2)}{n\,(n + 2)} - \frac{(n + 1)(n - s)\left(n - s - \frac{2}{(n+1)}\right)}{(n - 1)\,n\,(n + 2)}\right\}.$$

*and:*

$$\mathbb{E}\left\{\mathrm{tr}\left[\left(\Sigma\Lambda\Sigma\right)^2\right]\right\} = \left(n_1\sigma_1^2 + n_2\sigma_2^2\right)^2\left\{\frac{s\,(n - s)}{(n - 1)\,n\,(n + 2)}\right\}$$
$$+ \left(n_1\sigma_1^4 + n_2\sigma_2^4\right)\left\{\frac{n - s}{n\,(n + 2)}\right\}\left\{n - s + 1 + \frac{n - s - 1}{n - 1}\right\}.$$

*Proof.* See appendix D.2.7. $\qquad\square$

Substituting in the expression of $\mathbb{E}\left[T_2 \mid X_S\right]$ and using total expectation, we get:

$$
\begin{aligned}
&\mathbb{E}\left[T_2\right] \\
&= \mathbb{E}\left[\mathbb{E}\left[T_2 \mid X_S\right]\right] \\
&= \mathbb{E}\left[2\operatorname{tr}\left[(\Sigma\Lambda\Sigma)^2\right] + \left[\operatorname{tr}(\Sigma\Lambda\Sigma)\right]^2\right] \\
&= 2\mathbb{E}\left[\operatorname{tr}\left[(\Sigma\Lambda\Sigma)^2\right]\right] + \mathbb{E}\left[\left[\operatorname{tr}(\Sigma\Lambda\Sigma)\right]^2\right] \\
&= \left(n_1\sigma_1^2 + n_2\sigma_2^2\right)^2 \left\{\frac{2s(n-s)}{(n-1)n(n+2)}\right\} \\
&\quad + \left(n_1\sigma_1^4 + n_2\sigma_2^4\right)\left\{\frac{2(n-s)}{n(n+2)}\right\}\left\{n-s+1+\frac{n-s-1}{n-1}\right\} \\
&\quad + \left(n_1\sigma_1^2 + n_2\sigma_2^2\right)^2\left\{\frac{(n+1)(n-s)}{(n-1)n(n+2)}\right\}\left\{n-s-\frac{2}{n+1}\right\} \\
&\quad + \left(n_1\sigma_1^4 + n_2\sigma_2^4\right)\left\{\frac{(n-s)(n-s+2)}{n(n+2)} - \frac{(n+1)(n-s)\left(n-s-\frac{2}{(n+1)}\right)}{(n-1)n(n+2)}\right\} \\
&= \left(n_1\sigma_1^2 + n_2\sigma_2^2\right)^2\left\{\frac{n-s}{(n-1)n(n+2)}\right\}\left\{2s+(n+1)\left(n-s-\frac{2}{n+1}\right)\right\} \\
&\quad + \left(n_1\sigma_1^4 + n_2\sigma_2^4\right)\left\{\frac{n-s}{n(n+2)}\right\} \\
&\qquad \times \left\{2n-2s+2+\frac{2(n-s-1)}{n-1}+n-s+2-\frac{(n+1)\left(n-s-\frac{2}{(n+1)}\right)}{n-1}\right\} \\
&= \left(n_1\sigma_1^2 + n_2\sigma_2^2\right)^2\left\{\frac{n-s}{(n-1)n(n+2)}\right\}\left\{2s+(n+1)\left(n-s-\frac{2}{n+1}\right)\right\} \\
&\quad + \left(n_1\sigma_1^4 + n_2\sigma_2^4\right)\left\{\frac{n-s}{n(n+2)}\right\} \\
&\qquad \times \left\{3n-3s+4+\frac{2(n-s-1)}{n-1}-\frac{(n+1)\left(n-s-\frac{2}{(n+1)}\right)}{n-1}\right\}.
\end{aligned}
$$

Therefore:

$$
\begin{aligned}
\mathbb{E}\left[T_2\right] &= \left(n_1\sigma_1^2 + n_2\sigma_2^2\right)^2\left\{\frac{n-s}{(n-1)n(n+2)}\right\}\left\{2s+(n+1)\left(n-s-\frac{2}{n+1}\right)\right\} \\
&\quad + \left(n_1\sigma_1^4 + n_2\sigma_2^4\right)\left\{\frac{n-s}{n(n+2)}\right\} \\
&\qquad \times \left\{3n-3s+4+\frac{2(n-s-1)}{n-1}-\frac{(n+1)\left(n-s-\frac{2}{(n+1)}\right)}{n-1}\right\}. \quad (52)
\end{aligned}
$$

Finally, we compute $\mathbb{E}\left[T_3\right]$. We have:

$$
\begin{aligned}
\mathbb{E}\left[T_3 \mid X_S\right] &= \mathbb{E}\left\{\left[\vec{b}^T\left(X_S^T X_S\right)^{-1}\vec{b}\right]\left(W^T\Sigma\Lambda\Sigma W\right) \mid X_S\right\} \\
&= \left[\vec{b}^T\left(X_S^T X_S\right)^{-1}\vec{b}\right]\mathbb{E}\left\{W^T\Sigma\Lambda\Sigma W \mid X_S\right\} \\
&= \left[\vec{b}^T\left(X_S^T X_S\right)^{-1}\vec{b}\right]\operatorname{tr}\left(\Sigma\Lambda\Sigma\right).
\end{aligned}
$$

Recall from (49) that:

$$
X_S = QR,
$$

where $Q \in \mathbb{R}^{n \times s}$ and $R \in \mathbb{R}^{s \times s}$ are independent, $R$ is upper triangular with $R_{ii} > 0$ and $Q^T Q = I_s$. Therefore, by total expectation:

$$
\begin{aligned}
\mathbb{E}[T_3] &= \mathbb{E}\Big[\mathbb{E}[T_3 \mid X_S]\Big] \\
&= \mathbb{E}\Big\{ \Big[\vec{b}^T \left(X_S^T X_S\right)^{-1} \vec{b}\Big] \operatorname{tr}(\Sigma \Lambda \Sigma) \Big\} \\
&= \mathbb{E}\Big\{ \Big[\vec{b}^T \left(X_S^T X_S\right)^{-1} \vec{b}\Big] \operatorname{tr}\Big[\Sigma \Big(I_n - X_S \left(X_S^T X_S\right)^{-1} X_S^T\Big) \Sigma\Big] \Big\} \\
&= \mathbb{E}\Big\{ \Big[\vec{b}^T \left(R^T Q^T Q R\right)^{-1} \vec{b}\Big] \operatorname{tr}\Big[\Sigma \Big(I_n - QR \left(R^T Q^T Q R\right)^{-1} R^T Q^T\Big) \Sigma\Big] \Big\} \\
&= \mathbb{E}\Big\{ \Big[\vec{b}^T \left(R^T R\right)^{-1} \vec{b}\Big] \operatorname{tr}\Big[\Sigma \Big(I_n - QR \left(R^T R\right)^{-1} R^T Q^T\Big) \Sigma\Big] \Big\} \\
&= \mathbb{E}\Big\{ \Big[\vec{b}^T \left(R^T R\right)^{-1} \vec{b}\Big] \operatorname{tr}\Big[\Sigma \Big(I_n - QR R^{-1} \left(R^T\right)^{-1} R^T Q^T\Big) \Sigma\Big] \Big\} \\
&= \mathbb{E}\Big\{ \Big[\vec{b}^T \left(R^T R\right)^{-1} \vec{b}\Big] \operatorname{tr}\Big[\Sigma \left(I_n - QQ^T\right) \Sigma\Big] \Big\} \\
&= \mathbb{E}\Big\{ \Big[\vec{b}^T \left(R^T R\right)^{-1} \vec{b}\Big] \Big\} \, \mathbb{E}\Big\{ \operatorname{tr}\Big[\Sigma \left(I_n - QQ^T\right) \Sigma\Big] \Big\} \\
&= \mathbb{E}\Big\{ \Big[\vec{b}^T \left(R^T R\right)^{-1} \vec{b}\Big] \Big\} \, \mathbb{E}\Big\{ \operatorname{tr}\Big[\Sigma \left(I_n - QQ^T\right) \Sigma\Big] \Big\} \\
&= \mathbb{E}\Big\{ \Big[\vec{b}^T \left(X_S^T X_S\right)^{-1} \vec{b}\Big] \Big\} \, \mathbb{E}\Big\{ \operatorname{tr}\left(\Sigma U D U^T \Sigma\right) \Big\} .
\end{aligned}
$$

By expectation of inverse Wishart matrices (Anderson et al., 1958), we have:

$$
\mathbb{E}\Big\{ \Big[\vec{b}^T \left(X_S^T X_S\right)^{-1} \vec{b}\Big] \Big\} = \frac{1}{n-s-1} \left\|\vec{b}\right\|_2^2 .
$$

Using this in the above expression of $\mathbb{E}[T_3]$ yields:

$$
\begin{aligned}
\mathbb{E}[T_3] &= \mathbb{E}\Big\{ \Big[\vec{b}^T \left(X_S^T X_S\right)^{-1} \vec{b}\Big] \Big\} \, \mathbb{E}\Big\{ \operatorname{tr}\left(\Sigma U D U^T \Sigma\right) \Big\} \\
&= \frac{1}{n-s-1} \left\|\vec{b}\right\|_2^2 \operatorname{tr}\Big\{ \mathbb{E}\big[\Sigma U D U^T \Sigma\big] \Big\} \\
&= \frac{1}{n-s-1} \left\|\vec{b}\right\|_2^2 \operatorname{tr}\Big\{ \Sigma \mathbb{E}[U D U^T] \Sigma \Big\} \\
&= \frac{1}{n-s-1} \left\|\vec{b}\right\|_2^2 \operatorname{tr}\Big\{ \Sigma \left(\frac{\operatorname{tr}(D)}{n} I_n\right) \Sigma \Big\} \\
&= \frac{\operatorname{tr}(D)}{n(n-s-1)} \left\|\vec{b}\right\|_2^2 \operatorname{tr}\left(\Sigma^2\right) \\
&= \frac{(n-s)\left(n_1 \sigma_1^2 + n_2 \sigma_2^2\right)}{n(n-s-1)} \left\|\vec{b}\right\|_2^2 .
\end{aligned}
$$

Therefore:

$$
\mathbb{E}[T_3] = \frac{(n-s)\left(n_1 \sigma_1^2 + n_2 \sigma_2^2\right)}{n(n-s-1)} \left\|\vec{b}\right\|_2^2 . \tag{53}
$$

Now we have:

$$
\begin{aligned}
&\mathbb{E}\left[M_p^2\right] - \left(\mathbb{E}[M_p]\right)^2 \\
&= \lambda_p^4 \mathbb{E}[T_1] + \frac{1}{n^4} \mathbb{E}[T_2] + \frac{2\lambda_p^2}{n^2} \mathbb{E}[T_3] - \left(\frac{\lambda_p^2 s}{n-s-1} + \frac{(n-s)\left(n_1 \sigma_1^2 + n_2 \sigma_2^2\right)}{n^3}\right)^2 \\
&= \lambda_p^4 \left(\mathbb{E}[T_1] - \frac{s^2}{(n-s-1)^2}\right) + \left(\frac{\mathbb{E}[T_2]}{n^4} - \frac{(n-s)^2\left(n_1 \sigma_1^2 + n_2 \sigma_2^2\right)^2}{n^6}\right) \\
&\quad + \frac{2\lambda_p^2}{n^3}\left(\mathbb{E}[T_3]\, n - \frac{s(n-s)\left(n_1 \sigma_1^2 + n_2 \sigma_2^2\right)}{n-s-1}\right) .
\end{aligned}
$$

Let:

$$H_1 := \lambda_p^4 \left( \mathbb{E}\left[T_1\right] - \frac{s^2}{(n-s-1)^2} \right),$$

$$H_2 := \frac{\mathbb{E}\left[T_2\right]}{n^4} - \frac{(n-s)^2 \left(n_1 \sigma_1^2 + n_2 \sigma_2^2\right)^2}{n^6},$$

and:

$$H_3 := \frac{2\lambda_p^2}{n^3} \left\{ \mathbb{E}\left[T_3\right] n - \frac{s\left(n-s\right)\left(n_1 \sigma_1^2 + n_2 \sigma_2^2\right)}{n-s-1} \right\}.$$

Then we have, using (51):

$$
\begin{aligned}
\frac{H_1}{\left(\mathbb{E}\left[M_p\right]\right)^2} &= \frac{\lambda_p^4 \left( \mathbb{E}\left[T_1\right] - \frac{s^2}{(n-s-1)^2} \right)}{\left( \frac{\lambda_p^2 s}{n-s-1} + \frac{(n-s)\left(n_1 \sigma_1^2 + n_2 \sigma_2^2\right)}{n^3} \right)^2} \\
&\leq \frac{\lambda_p^4 s^2 \left( \frac{1}{(n-s)(n-s-3)} \left( 1 + \frac{1}{n-s-1} \right) - \frac{1}{(n-s-1)^2} \right)}{\frac{\lambda_p^4 s^2}{(n-s-1)^2}} \\
&= (n-s-1)^2 \left( \frac{1}{(n-s)(n-s-3)} \left( 1 + \frac{1}{n-s-1} \right) - \frac{1}{(n-s-1)^2} \right) \\
&= \frac{(n-s-1)^2}{(n-s)(n-s-3)} \left( 1 + \frac{1}{n-s-1} \right) - 1 \\
&= o\left(1\right).
\end{aligned}
$$

Similarly, using (52):

$$
\begin{aligned}
&H_2 \\
&= \frac{\mathbb{E}\left[T_2\right]}{n^4} - \frac{(n-s)^2 \left(n_1 \sigma_1^2 + n_2 \sigma_2^2\right)^2}{n^6} \\
&= \left(n_1 \sigma_1^2 + n_2 \sigma_2^2\right)^2 \left\{ \frac{2s\left(n-s\right)}{(n-1)\,n^5\,(n+2)} + \frac{(n-s)(n+1)}{(n-1)\,n^5\,(n+2)} \left( n-s-\frac{2}{n+1} \right) - \frac{(n-s)^2}{n^6} \right\} \\
&\quad + \left(n_1 \sigma_1^4 + n_2 \sigma_2^4\right) \left\{ \frac{n-s}{n^5\,(n+2)} \right\} \left\{ 3n - 3s + 4 + \frac{2\left(n-s-1\right)}{n-1} - \frac{(n+1)\left(n-s-\frac{2}{(n+1)}\right)}{n-1} \right\}.
\end{aligned}
$$

We have:

$$\left(\mathbb{E}\left[M_p\right]\right)^2 \geq \frac{(n-s)^2 \left(n_1 \sigma_1^2 + n_2 \sigma_2^2\right)^2}{n^6}.$$

Hence:

$$\frac{H_2}{\left(\mathbb{E}\left[M_p\right]\right)^2}$$

$$\leq \left\{\frac{n^6}{(n-s)^2}\right\}\left\{\frac{2s(n-s)}{(n-1)n^5(n+2)} + \frac{(n-s)(n+1)}{(n-1)n^5(n+2)}\left(n-s-\frac{2}{n+1}\right) - \frac{(n-s)^2}{n^6}\right\}$$

$$+ \frac{n_1\sigma_1^4 + n_2\sigma_2^4}{(n_1\sigma_1^2 + n_2\sigma_2^2)^2}\left\{\frac{n^6}{(n-s)^2}\right\}\left\{\frac{(n-s)}{n^5(n+2)}\right\}$$

$$\times \left\{3n - 3s + 4 + \frac{2(n-s-1)}{n-1} - \frac{(n+1)\left(n-s-\frac{2}{(n+1)}\right)}{n-1}\right\}$$

$$= \left\{\frac{2sn}{(n-1)(n-s)(n+2)} + \frac{n(n+1)}{(n-s)(n-1)(n+2)}\left(n-s-\frac{2}{n+1}\right) - 1\right\}$$

$$+ \frac{n_1\sigma_1^4 + n_2\sigma_2^4}{(n_1\sigma_1^2 + n_2\sigma_2^2)^2}\left\{\frac{n}{(n+2)(n-s)}\right\}$$

$$\times \left\{3n - 3s + 4 + \frac{2(n-s-1)}{n-1} - \frac{(n+1)\left(n-s-\frac{2}{(n+1)}\right)}{n-1}\right\}$$

$$= \left\{\frac{2sn}{(n-1)(n-s)(n+2)} + \frac{n(n+1)}{(n-s)(n-1)(n+2)}\left(n-s-\frac{2}{n+1}\right) - 1\right\}$$

$$+ \frac{n_1\sigma_1^4 + n_2\sigma_2^4}{(n_1\sigma_1^2 + n_2\sigma_2^2)^2}$$

$$\times \left\{\frac{3n}{(n+2)} + \frac{4n}{(n+2)(n-s)} + \frac{2(n-s-1)n}{(n-s)(n-1)(n+2)} - \frac{n(n+1)\left(n-s-\frac{2}{(n+1)}\right)}{(n-s)(n-1)(n+2)}\right\}$$

$$= \left\{\frac{2sn}{(n-1)(n-s)(n+2)} + \frac{n(n+1)}{(n-s)(n-1)(n+2)}\left(n-s-\frac{2}{n+1}\right) - 1\right\}$$

$$+ \left\{\frac{n_1\sigma_1^4}{(n_1\sigma_1^2 + n_2\sigma_2^2)^2} + \frac{n_2\sigma_2^4}{(n_1\sigma_1^2 + n_2\sigma_2^2)^2}\right\}$$

$$\times \left\{\frac{3n}{(n+2)} + \frac{4n}{(n+2)(n-s)} + \frac{2(n-s-1)n}{(n-s)(n-1)(n+2)} - \frac{n(n+1)\left(n-s-\frac{2}{(n+1)}\right)}{(n-s)(n-1)(n+2)}\right\}$$

$$\leq \left\{\frac{2sn}{(n-1)(n-s)(n+2)} + \frac{n(n+1)}{(n-s)(n-1)(n+2)}\left(n-s-\frac{2}{n+1}\right) - 1\right\}$$

$$+ \left\{\frac{1}{n_1} + \frac{1}{n_2}\right\}$$

$$\times \left\{\frac{3n}{(n+2)} + \frac{4n}{(n+2)(n-s)} + \frac{2(n-s-1)n}{(n-s)(n-1)(n+2)} - \frac{n(n+1)\left(n-s-\frac{2}{(n+1)}\right)}{(n-s)(n-1)(n+2)}\right\}$$

$$= o(1).$$

For $H_3$, using (53):

$$
\begin{aligned}
H_3 &:= \frac{2\lambda_p^2}{n^3}\left\{\mathbb{E}\left[T_3\right]n - \frac{s\left(n-s\right)\left(n_1\sigma_1^2 + n_2\sigma_2^2\right)}{n-s-1}\right\} \\
&= \frac{2\lambda_p^2}{n^3}\left\{\frac{n\left(n-s\right)s\left(n_1\sigma_1^2 + n_2\sigma_2^2\right)}{n\left(n-s-1\right)} - \frac{s\left(n-s\right)\left(n_1\sigma_1^2 + n_2\sigma_2^2\right)}{n-s-1}\right\} \\
&= 0.
\end{aligned}
$$

Therefore, we conclude:

$$
\frac{\mathbb{E}\left[M_p^2\right] - \left(\mathbb{E}\left[M_p\right]\right)^2}{\left(\mathbb{E}\left[M_p\right]\right)^2} = \frac{H_1}{\left(\mathbb{E}\left[M_p\right]\right)^2} + \frac{H_2}{\left(\mathbb{E}\left[M_p\right]\right)^2} + \frac{H_3}{\left(\mathbb{E}\left[M_p\right]\right)^2} = o\left(1\right).
$$

$\square$

### D.2.6 PROOF OF LEMMA D.3

*Proof of Lemma D.3.* Define:

$$
\begin{aligned}
f\colon \mathbb{R}^k &\longrightarrow \mathbb{R} \\
w &\longmapsto \max_{i\in[k]} w_i.
\end{aligned}
$$

Note that for any $u, v \in \mathbb{R}^k$:

$$
\begin{aligned}
\left|f\left(u\right) - f\left(v\right)\right| &= \left|\max_{i\in[k]} u_i - \max_{i\in[k]} v_i\right| \\
&\leq \max_{i\in[k]}\left|u_i - v_i\right| \\
&\leq \sqrt{\sum_{i\in[k]}\left(u_i - v_i\right)^2} \\
&= \left\|u - v\right\|_2.
\end{aligned}
$$

Therefore $f$ is 1-Lipschitz. Assume $\tau^2 = 1$. By Gaussian concentration of measure for Lipschitz functions (Ledoux, 2001; Massart, 2007), we have for all $t \geq 0$:

$$
\begin{cases}
\mathbb{P}\left(\max_{i\in[k]} N_i - \mathbb{E}\left[\max_{i\in[k]} N_i\right] > t\right) \leq \exp\left(-t^2/2\right), \\
\mathbb{P}\left(\max_{i\in[k]} N_i - \mathbb{E}\left[\max_{i\in[k]} N_i\right] < -t\right) \leq \exp\left(-t^2/2\right).
\end{cases}
$$

The result for general $\tau^2$ follows by substituting $t := \eta/\tau$. $\square$

### D.2.7 PROOF OF LEMMA D.5

*Proof of Lemma D.5.* We have, by Einstein notation:

$$
\left[\mathrm{tr}\left(\Sigma\Lambda\Sigma\right)\right]^2
$$

$$
= \left(\sum_{a=1}^{n}\left[\Sigma U D U^T \Sigma\right]_{aa}\right)^2
$$

$$
= \left(\sum_{a=1}^{n}\sum_{b=1}^{n}\sum_{c=1}^{n}\sum_{d=1}^{n}\sum_{e=1}^{n}\Sigma_{ab}U_{bc}D_{cd}\left[U^T\right]_{de}\Sigma_{ea}\right)^2
$$

$$
= \left(\sum_{a=1}^{n}\sum_{c=1}^{n}\Sigma_{aa}U_{ac}D_{cc}U_{ac}\Sigma_{aa}\right)^2
$$

$$
= \left(\sum_{a=1}^{n}\sum_{c=1}^{n}D_{aa}U_{ac}^2\Sigma_{cc}^2\right)^2
$$

$$
= \sum_{a=1}^{n}\sum_{c=1}^{n}D_{cc}^2 U_{ac}^4 \Sigma_{aa}^4 + \sum_{a=1}^{n}\sum_{c\neq d\in[n]}D_{cc}D_{dd}U_{ac}^2 U_{ad}^2 \Sigma_{aa}^4
$$

$$
+ \sum_{a\neq b\in[n]}\sum_{c=1}^{n}D_{cc}^2 U_{ac}^2 U_{bc}^2 \Sigma_{aa}^2\Sigma_{bb}^2 + \sum_{a\neq b\in[n]}\sum_{c\neq d\in[n]}D_{cc}D_{dd}U_{ac}^2 U_{bd}^2 \Sigma_{aa}^2\Sigma_{bb}^2.
$$

Therefore:

$$
\mathbb{E}\left\{\left[\mathrm{tr}\left(\Sigma\Lambda\Sigma\right)\right]^2\right\} = \sum_{a=1}^{n}\sum_{c=1}^{n}D_{cc}^2\Sigma_{aa}^4 \mathbb{E}\left[U_{ac}^4\right]
$$

$$
+ \sum_{a=1}^{n}\sum_{c\neq d\in[n]}D_{cc}D_{dd}\Sigma_{aa}^4 \mathbb{E}\left[U_{ac}^2 U_{ad}^2\right]
$$

$$
+ \sum_{a\neq b\in[n]}\sum_{c=1}^{n}D_{cc}^2\Sigma_{aa}^2\Sigma_{bb}^2 \mathbb{E}\left[U_{ac}^2 U_{bc}^2\right]
$$

$$
+ \sum_{a\neq b\in[n]}\sum_{c\neq d\in[n]}D_{cc}D_{dd}\Sigma_{aa}^2\Sigma_{bb}^2 \mathbb{E}\left[U_{ac}^2 U_{bd}^2\right].
$$

We use the following result from (Meckes, 2019) on fourth-moments of Haar$(n)$ matrices:

**Lemma D.6** (Fourth-moments of Haar$(n)$ matrices, Lemma 2.22 in (Meckes, 2019))**.**
*Let $U \sim Haar\left(n\right)$. Then for all $i, j, r, s, \alpha, \beta, \lambda, \mu \in [n]$ we have:*

$$
\mathbb{E}\left[U_{ij}U_{rs}U_{\alpha\beta}U_{\lambda\mu}\right]
$$

$$
= -\frac{1}{(n-1)\,n\,(n+2)}\Big[\delta_{ir}\delta_{\alpha\lambda}\delta_{j\beta}\delta_{s\mu} + \delta_{ir}\delta_{\alpha\lambda}\delta_{j\mu}\delta_{s\beta} + \delta_{i\alpha}\delta_{r\lambda}\delta_{js}\delta_{\beta\mu}
$$

$$
+ \delta_{i\alpha}\delta_{r\lambda}\delta_{j\mu}\delta_{\beta s} + \delta_{i\lambda}\delta_{r\alpha}\delta_{js}\delta_{\beta\mu} + \delta_{i\lambda}\delta_{r\alpha}\delta_{j\beta}\delta_{s\mu}\Big]
$$

$$
+ \frac{n+1}{(n-1)\,n\,(n+2)}\Big[\delta_{ir}\delta_{\alpha\lambda}\delta_{js}\delta_{\beta\mu} + \delta_{i\alpha}\delta_{r\lambda}\delta_{j\beta}\delta_{s\mu} + \delta_{i\lambda}\delta_{r\alpha}\delta_{j\mu}\delta_{s\beta}\Big].
$$

Substituting: $i, r, \alpha, \lambda := a$; and $j, s, \beta, \mu := c$, we get:

$$\mathbb{E}\left[U_{ac}^4\right]$$
$$= -\frac{1}{(n-1)\,n\,(n+2)}\Big[\delta_{aa}\delta_{aa}\delta_{cc}\delta_{cc} + \delta_{aa}\delta_{aa}\delta_{cc}\delta_{cc} + \delta_{aa}\delta_{aa}\delta_{cc}\delta_{cc}$$
$$+ \delta_{aa}\delta_{aa}\delta_{cc}\delta_{cc} + \delta_{aa}\delta_{aa}\delta_{cc}\delta_{cc} + \delta_{aa}\delta_{aa}\delta_{cc}\delta_{cc}\Big]$$
$$+ \frac{n+1}{(n-1)\,n\,(n+2)}\Big[\delta_{aa}\delta_{aa}\delta_{cc}\delta_{cc} + \delta_{aa}\delta_{aa}\delta_{cc}\delta_{cc} + \delta_{aa}\delta_{aa}\delta_{cc}\delta_{cc}\Big]$$
$$= -\frac{6}{(n-1)\,n\,(n+2)} + \frac{3\,(n+1)}{(n-1)\,n\,(n+2)}$$
$$= \frac{3}{n\,(n+2)}.$$

Thus:

$$\mathbb{E}\left[U_{ac}^4\right] = \frac{3}{n\,(n+2)}. \tag{54}$$

Now substituting $i, r, \alpha, \lambda := a$; $j, s := c$; and $\beta, \mu := d$, we get:

$$\mathbb{E}\left[U_{ac}^2 U_{ad}^2\right]$$
$$= -\frac{1}{(n-1)\,n\,(n+2)}\Big[\delta_{aa}\delta_{aa}\delta_{cd}\delta_{cd} + \delta_{aa}\delta_{aa}\delta_{cd}\delta_{cd} + \delta_{aa}\delta_{aa}\delta_{cc}\delta_{dd}$$
$$+ \delta_{aa}\delta_{aa}\delta_{cd}\delta_{dc} + \delta_{aa}\delta_{aa}\delta_{cc}\delta_{dd} + \delta_{aa}\delta_{aa}\delta_{cd}\delta_{cd}\Big]$$
$$+ \frac{n+1}{(n-1)\,n\,(n+2)}\Big[\delta_{aa}\delta_{aa}\delta_{cc}\delta_{dd} + \delta_{aa}\delta_{aa}\delta_{cd}\delta_{cd} + \delta_{aa}\delta_{aa}\delta_{cd}\delta_{cd}\Big]$$
$$= -\frac{2}{(n-1)\,n\,(n+2)} + \frac{n+1}{(n-1)\,n\,(n+2)}$$
$$= \frac{1}{n\,(n+2)}.$$

Thus:

$$\mathbb{E}\left[U_{ac}^2 U_{ad}^2\right] = \frac{1}{n\,(n+2)}. \tag{55}$$

Now substituting $i, r := a$; $j, s, \beta, \mu := c$; and $\alpha, \lambda := b$, we get:

$$\mathbb{E}\left[U_{ac}^2 U_{bc}^2\right]$$
$$= -\frac{1}{(n-1)\,n\,(n+2)}\Big[\delta_{aa}\delta_{bb}\delta_{cc}\delta_{cc} + \delta_{aa}\delta_{bb}\delta_{cc}\delta_{cc} + \delta_{ab}\delta_{ab}\delta_{cc}\delta_{cc}$$
$$+ \delta_{ab}\delta_{ab}\delta_{cc}\delta_{cc} + \delta_{ab}\delta_{ab}\delta_{cc}\delta_{cc} + \delta_{ab}\delta_{ab}\delta_{cc}\delta_{cc}\Big]$$
$$+ \frac{n+1}{(n-1)\,n\,(n+2)}\Big[\delta_{aa}\delta_{bb}\delta_{cc}\delta_{cc} + \delta_{ab}\delta_{ab}\delta_{cc}\delta_{cc} + \delta_{ab}\delta_{ab}\delta_{cc}\delta_{cc}\Big]$$
$$+ \frac{n+1}{(n-1)\,n\,(n+2)}\Big[\delta_{aa}\delta_{aa}\delta_{cc}\delta_{dd} + \delta_{aa}\delta_{aa}\delta_{cd}\delta_{cd} + \delta_{aa}\delta_{aa}\delta_{cd}\delta_{cd}\Big]$$
$$= -\frac{2}{(n-1)\,n\,(n+2)} + \frac{n+1}{(n-1)\,n\,(n+2)}$$
$$= \frac{1}{n\,(n+2)}.$$

Thus:

$$\mathbb{E}\left[U_{ac}^2 U_{bc}^2\right] = \frac{1}{n\,(n+2)}. \tag{56}$$

Now substituting $i, r := a$; $j, s := c$; $\alpha, \lambda := b$; and $\beta, \mu := d$, we get:

$$
\begin{aligned}
&\mathbb{E}\left[U_{ac}^2 U_{bd}^2\right] \\
&= -\frac{1}{(n-1)\,n\,(n+2)}\Big[\delta_{aa}\delta_{bb}\delta_{cd}\delta_{cd} + \delta_{aa}\delta_{bb}\delta_{cd}\delta_{cd} + \delta_{ab}\delta_{ab}\delta_{cc}\delta_{dd} \\
&\qquad\qquad\qquad\qquad + \delta_{ab}\delta_{ab}\delta_{cd}\delta_{dc} + \delta_{ab}\delta_{ab}\delta_{cc}\delta_{dd} + \delta_{ab}\delta_{ab}\delta_{cd}\delta_{cd}\Big] \\
&\quad + \frac{n+1}{(n-1)\,n\,(n+2)}\Big[\delta_{aa}\delta_{bb}\delta_{cc}\delta_{dd} + \delta_{ab}\delta_{ab}\delta_{cd}\delta_{cd} + \delta_{ab}\delta_{ab}\delta_{cd}\delta_{cd}\Big] \\
&= \frac{n+1}{(n-1)\,n\,(n+2)}.
\end{aligned}
$$

Thus:

$$
\mathbb{E}\left[U_{ac}^2 U_{bd}^2\right] = \frac{n+1}{(n-1)\,n\,(n+2)}. \tag{57}
$$

Substituting (54), (55), (56), and (57) in the expression of $\mathbb{E}\left\{\left[\mathrm{tr}\left(\Sigma\Lambda\Sigma\right)\right]^2\right\}$ above, we get:

$$\mathbb{E}\left\{\left[\mathrm{tr}\left(\Sigma\Lambda\Sigma\right)\right]^2\right\}$$

$$= \sum_{a=1}^{n}\sum_{c=1}^{n} D_{cc}^2 \Sigma_{aa}^4 \mathbb{E}\left[U_{ac}^4\right] + \sum_{a=1}^{n}\sum_{c\neq d\in[n]} D_{cc}D_{dd}\Sigma_{aa}^4 \mathbb{E}\left[U_{ac}^2 U_{ad}^2\right]$$

$$+ \sum_{a\neq b\in[n]}\sum_{c=1}^{n} D_{cc}^2 \Sigma_{aa}^2\Sigma_{bb}^2 \mathbb{E}\left[U_{ac}^2 U_{bc}^2\right] + \sum_{a\neq b\in[n]}\sum_{c\neq d\in[n]} D_{cc}D_{dd}\Sigma_{aa}^2\Sigma_{bb}^2 \mathbb{E}\left[U_{ac}^2 U_{bd}^2\right]$$

$$= \frac{3}{n\left(n+2\right)}\left(\sum_{a=1}^{n}\Sigma_{aa}^4\right)\left(\sum_{c=1}^{n}D_{cc}^2\right) + \frac{1}{n\left(n+2\right)}\left(\sum_{a=1}^{n}\Sigma_{aa}^4\right)\left(\sum_{c\neq d\in[n]}D_{cc}D_{dd}\right)$$

$$+ \frac{1}{n\left(n+2\right)}\left(\sum_{a\neq b\in[n]}\Sigma_{aa}^2\Sigma_{bb}^2\right)\left(\sum_{c=1}^{n}D_{cc}^2\right)$$

$$+ \frac{n+1}{\left(n-1\right)n\left(n+2\right)}\left(\sum_{a\neq b\in[n]}\Sigma_{aa}^2\Sigma_{bb}^2\right)\left(\sum_{c\neq d\in[n]}D_{cc}D_{dd}\right)$$

$$= \frac{1}{n\left(n+2\right)}\left(\sum_{a=1}^{n}\Sigma_{aa}^4\right)\left\{3\left(\sum_{c=1}^{n}D_{cc}^2\right) + \left(\sum_{c\neq d\in[n]}D_{cc}D_{dd}\right)\right\}$$

$$+ \frac{n+1}{\left(n-1\right)n\left(n+2\right)}\left(\sum_{a\neq b\in[n]}\Sigma_{aa}^2\Sigma_{bb}^2\right)\left\{\frac{n-1}{n+1}\left(\sum_{c=1}^{n}D_{cc}^2\right) + \left(\sum_{c\neq d\in[n]}D_{cc}D_{dd}\right)\right\}$$

$$= \frac{1}{n\left(n+2\right)}\left(\sum_{a=1}^{n}\Sigma_{aa}^4\right)\left\{2\left(\sum_{c=1}^{n}D_{cc}^2\right) + \left(\sum_{c=1}^{n}D_{cc}\right)^2\right\}$$

$$+ \frac{n+1}{\left(n-1\right)n\left(n+2\right)}\left\{\left(\sum_{a=1}^{n}\Sigma_{aa}^2\right)^2 - \sum_{a=1}^{n}\Sigma_{aa}^4\right\}\left\{-\frac{2}{n+1}\left(\sum_{c=1}^{n}D_{cc}^2\right) + \left(\sum_{c=1}^{n}D_{cc}\right)^2\right\}$$

$$= \frac{\mathrm{tr}\left(\Sigma^4\right)}{n\left(n+2\right)}\left\{2\,\mathrm{tr}\left(D^2\right) + \mathrm{tr}\left(D\right)^2\right\}$$

$$+ \frac{n+1}{\left(n-1\right)n\left(n+2\right)}\left\{\mathrm{tr}\left(\Sigma^2\right)^2 - \mathrm{tr}\left(\Sigma^4\right)\right\}\left\{-\frac{2\,\mathrm{tr}\left(D^2\right)}{n+1} + \mathrm{tr}\left(D\right)^2\right\}$$

$$= \frac{n_1\sigma_1^4 + n_2\sigma_2^4}{n\left(n+2\right)}\left\{2\left(n-s\right) + \left(n-s\right)^2\right\}$$

$$+ \frac{n+1}{\left(n-1\right)n\left(n+2\right)}\left\{\left(n_1\sigma_1^2 + n_2\sigma_2^2\right)^2 - \left(n_1\sigma_1^4 + n_2\sigma_2^4\right)\right\}\left\{-\frac{2\left(n-s\right)}{n+1} + \left(n-s\right)^2\right\}$$

$$= \left(n_1\sigma_1^2 + n_2\sigma_2^2\right)^2\left\{\frac{\left(n+1\right)\left(n-s\right)}{\left(n-1\right)n\left(n+2\right)}\right\}\left\{n - s - \frac{2}{n+1}\right\}$$

$$+ \left(n_1\sigma_1^4 + n_2\sigma_2^4\right)\left\{\frac{\left(n-s\right)\left(n-s+2\right)}{n\left(n+2\right)} - \frac{\left(n+1\right)\left(n-s\right)\left(n-s-\frac{2}{\left(n+1\right)}\right)}{\left(n-1\right)n\left(n+2\right)}\right\}.$$

Now we have, by Einstein notation:

$$\operatorname{tr}\left[(\Sigma\Lambda\Sigma)^2\right]$$

$$= \operatorname{tr}\left[\Sigma U D U^T \Sigma^2 U D U^T \Sigma\right]$$

$$= \sum_{a=1}^n \left[\Sigma U D U^T \Sigma^2 U D U^T \Sigma\right]_{aa}$$

$$= \sum_{a=1}^n \sum_{b=1}^n \sum_{c=1}^n \sum_{d=1}^n \sum_{e=1}^n \sum_{f=1}^n \sum_{g=1}^n \sum_{h=1}^n \sum_{j=1}^n \Sigma_{ab} U_{bc} D_{cd} \left[U^T\right]_{de} \left[\Sigma^2\right]_{ef} U_{fg} D_{gh} \left[U^T\right]_{hj} \Sigma_{ja}$$

$$= \sum_{a=1}^n \sum_{c=1}^n \sum_{e=1}^n \sum_{h=1}^n \Sigma_{aa} U_{ac} D_{cc} U_{ec} \Sigma_{ee}^2 U_{eh} D_{hh} U_{ah} \Sigma_{aa}$$

$$= \sum_{a=1}^n \sum_{c=1}^n \sum_{e=1}^n \sum_{h=1}^n D_{cc} D_{hh} \Sigma_{aa}^2 \Sigma_{ee}^2 U_{ac} U_{ec} U_{eh} U_{ah}$$

$$= \sum_{a=1}^n \sum_{c=1}^n D_{cc}^2 \Sigma_{aa}^4 U_{ac}^4 + \sum_{a\neq e\in[n]} \sum_{c=1}^n D_{cc}^2 \Sigma_{aa}^2 \Sigma_{ee}^2 U_{ac}^2 U_{ec}^2 + \sum_{a=1}^n \sum_{c\neq h\in[n]} D_{cc} D_{hh} \Sigma_{aa}^4 U_{ac}^2 U_{ah}^2$$

$$+ \sum_{a\neq e\in[n]} \sum_{c\neq h\in[n]} D_{cc} D_{hh} \Sigma_{aa}^2 \Sigma_{ee}^2 U_{ac} U_{ec} U_{eh} U_{ah}.$$

Therefore:

$$\mathbb{E}\left\{\operatorname{tr}\left[(\Sigma\Lambda\Sigma)^2\right]\right\} = \sum_{a=1}^n \sum_{c=1}^n D_{cc}^2 \Sigma_{aa}^4 \mathbb{E}\left[U_{ac}^4\right]$$

$$+ \sum_{a\neq e\in[n]} \sum_{c=1}^n D_{cc}^2 \Sigma_{aa}^2 \Sigma_{ee}^2 \mathbb{E}\left[U_{ac}^2 U_{ec}^2\right]$$

$$+ \sum_{a=1}^n \sum_{c\neq h\in[n]} D_{cc} D_{hh} \Sigma_{aa}^4 \mathbb{E}\left[U_{ac}^2 U_{ah}^2\right]$$

$$+ \sum_{a\neq e\in[n]} \sum_{c\neq h\in[n]} D_{cc} D_{hh} \Sigma_{aa}^2 \Sigma_{ee}^2 \mathbb{E}\left[U_{ac} U_{ec} U_{eh} U_{ah}\right].$$

Recall that:

$$\mathbb{E}\left[U_{ac}^4\right] = \frac{3}{n(n+2)},$$

and:

$$\mathbb{E}\left[U_{ac}^2 U_{ec}^2\right] = \mathbb{E}\left[U_{ac}^2 U_{ah}^2\right] = \frac{1}{n(n+2)}.$$

Now substituting $i, \lambda := a$; $j, s := c$; $r, \alpha := e$; and $\beta, \mu := h$ in Lemma D.6, we get:

$$\mathbb{E}\left[U_{ac} U_{ec} U_{eh} U_{ah}\right]$$

$$= -\frac{1}{(n-1)n(n+2)}\Big[\delta_{ae}\delta_{ea}\delta_{ch}\delta_{ch} + \delta_{ae}\delta_{ea}\delta_{ch}\delta_{ch} + \delta_{ae}\delta_{ea}\delta_{cc}\delta_{hh}$$

$$+ \delta_{ae}\delta_{ea}\delta_{ch}\delta_{hc} + \delta_{aa}\delta_{ee}\delta_{cc}\delta_{hh} + \delta_{aa}\delta_{ee}\delta_{ch}\delta_{ch}\Big]$$

$$+ \frac{n+1}{(n-1)n(n+2)}\Big[\delta_{ae}\delta_{ea}\delta_{cc}\delta_{hh} + \delta_{ae}\delta_{ea}\delta_{ch}\delta_{ch} + \delta_{aa}\delta_{ee}\delta_{ch}\delta_{ch}\Big]$$

$$= -\frac{1}{(n-1)n(n+2)}.$$

Therefore:

$$\mathbb{E}\left[U_{ac} U_{ec} U_{eh} U_{ah}\right] = -\frac{1}{(n-1)n(n+2)}. \tag{58}$$

Substituting (54), (56), (55), and (58) in the expression of $\mathbb{E}\left\{ \mathrm{tr}\left[ (\Sigma\Lambda\Sigma)^2 \right] \right\}$ above, we get:

$$
\mathbb{E}\left\{ \mathrm{tr}\left[ (\Sigma\Lambda\Sigma)^2 \right] \right\}
$$

$$
= \frac{3}{n(n+2)} \sum_{a=1}^{n} \sum_{c=1}^{n} D_{cc}^2 \Sigma_{aa}^4 + \frac{1}{n(n+2)} \sum_{a\neq e\in[n]} \sum_{c=1}^{n} D_{cc}^2 \Sigma_{aa}^2 \Sigma_{ee}^2
$$

$$
+ \frac{1}{n(n+2)} \sum_{a=1}^{n} \sum_{c\neq h\in[n]} D_{cc} D_{hh} \Sigma_{aa}^4
$$

$$
- \frac{1}{(n-1)n(n+2)} \sum_{a\neq e\in[n]} \sum_{c\neq h\in[n]} D_{cc} D_{hh} \Sigma_{aa}^2 \Sigma_{ee}^2
$$

$$
= \frac{3}{n(n+2)} \left( \sum_{a=1}^{n} \Sigma_{aa}^4 \right) \left( \sum_{c=1}^{n} D_{cc}^2 \right) + \frac{1}{n(n+2)} \left( \sum_{a\neq e\in[n]} \Sigma_{aa}^2 \Sigma_{ee}^2 \right) \left( \sum_{c=1}^{n} D_{cc}^2 \right)
$$

$$
+ \frac{1}{n(n+2)} \left( \sum_{a=1}^{n} \Sigma_{aa}^4 \right) \left( \sum_{c\neq h\in[n]} D_{cc} D_{hh} \right)
$$

$$
- \frac{1}{(n-1)n(n+2)} \left( \sum_{a\neq e\in[n]} \Sigma_{aa}^2 \Sigma_{ee}^2 \right) \left( \sum_{c\neq h\in[n]} D_{cc} D_{hh} \right)
$$

$$
= \frac{1}{n(n+2)} \left( \sum_{c=1}^{n} D_{cc}^2 \right) \left\{ 3 \sum_{a=1}^{n} \Sigma_{aa}^4 + \sum_{a\neq e\in[n]} \Sigma_{aa}^2 \Sigma_{ee}^2 \right\}
$$

$$
+ \frac{1}{n(n+2)} \left( \sum_{a=1}^{n} \Sigma_{aa}^4 \right) \left\{ \left( \sum_{c=1}^{n} D_{cc} \right)^2 - \sum_{c=1}^{n} D_{cc}^2 \right\}
$$

$$
- \frac{1}{(n-1)n(n+2)} \left\{ \left( \sum_{a=1}^{n} \Sigma_{aa}^2 \right)^2 - \sum_{a=1}^{n} \Sigma_{aa}^4 \right\} \left\{ \left( \sum_{c=1}^{n} D_{cc} \right)^2 - \sum_{c=1}^{n} D_{cc}^2 \right\}
$$

$$
= \frac{1}{n(n+2)} \left( \sum_{c=1}^{n} D_{cc}^2 \right) \left\{ 2 \sum_{a=1}^{n} \Sigma_{aa}^4 + \left( \sum_{a=1}^{n} \Sigma_{aa}^2 \right)^2 \right\}
$$

$$
+ \frac{1}{n(n+2)} \left( \sum_{a=1}^{n} \Sigma_{aa}^4 \right) \left\{ \left( \sum_{c=1}^{n} D_{cc} \right)^2 - \sum_{c=1}^{n} D_{cc}^2 \right\}
$$

$$
- \frac{1}{(n-1)n(n+2)} \left\{ \left( \sum_{a=1}^{n} \Sigma_{aa}^2 \right)^2 - \sum_{a=1}^{n} \Sigma_{aa}^4 \right\} \left\{ \left( \sum_{c=1}^{n} D_{cc} \right)^2 - \sum_{c=1}^{n} D_{cc}^2 \right\}.
$$

Thus:

$$
\mathbb{E}\left\{\operatorname{tr}\left[\left(\Sigma\Lambda\Sigma\right)^2\right]\right\}
$$

$$
= \frac{\operatorname{tr}\left(D^2\right)}{n\left(n+2\right)}\left\{2\operatorname{tr}\left(\Sigma^4\right) + \left(\operatorname{tr}\Sigma^2\right)^2\right\} + \frac{\operatorname{tr}\left(\Sigma^4\right)}{n\left(n+2\right)}\left\{\operatorname{tr}\left(D\right)^2 - \operatorname{tr}\left(D^2\right)\right\}
$$

$$
- \frac{1}{\left(n-1\right)n\left(n+2\right)}\left\{\operatorname{tr}\left(\Sigma^2\right)^2 - \operatorname{tr}\left(\Sigma^4\right)\right\}\left\{\operatorname{tr}\left(D\right)^2 - \operatorname{tr}\left(D^2\right)\right\}
$$

$$
= \frac{n-s}{n\left(n+2\right)}\left\{2\left(n_1\sigma_1^4 + n_2\sigma_2^4\right) + \left(n_1\sigma_1^2 + n_2\sigma_2^2\right)^2\right\} + \frac{n_1\sigma_1^4 + n_2\sigma_2^4}{n\left(n+2\right)}\left\{\left(n-s\right)^2 - \left(n-s\right)\right\}
$$

$$
- \frac{1}{\left(n-1\right)n\left(n+2\right)}\left\{\left(n_1\sigma_1^2 + n_2\sigma_2^2\right)^2 - \left(n_1\sigma_1^4 + n_2\sigma_2^4\right)\right\}\left\{\left(n-s\right)^2 - \left(n-s\right)\right\}
$$

$$
= \left(n_1\sigma_1^2 + n_2\sigma_2^2\right)^2\left\{\frac{n-s}{n\left(n+2\right)}\right\}\left\{1 - \frac{n-s-1}{n-1}\right\}
$$

$$
+ \left(n_1\sigma_1^4 + n_2\sigma_2^4\right)\left\{\frac{2\left(n-s\right)}{n\left(n+2\right)} + \frac{\left(n-s\right)\left(n-s-1\right)}{n\left(n+2\right)} - \frac{\left(n-s\right)\left(n-s-1\right)}{\left(n-1\right)n\left(n+2\right)}\right\}
$$

$$
= \left(n_1\sigma_1^2 + n_2\sigma_2^2\right)^2\left\{\frac{s\left(n-s\right)}{\left(n-1\right)n\left(n+2\right)}\right\}
$$

$$
+ \left(n_1\sigma_1^4 + n_2\sigma_2^4\right)\left\{\frac{n-s}{n\left(n+2\right)}\right\}\left\{n-s+1 + \frac{n-s-1}{n-1}\right\}.
$$

$\square$

## D.3    PROOF OF PROPOSITION D.3

*Proof of Proposition D.3.* Recall that:

$$
U_i := e_i^T\left(\frac{1}{n}X_S^T X_S\right)^{-1}\left[\frac{1}{n}X_S^T\Sigma W - \lambda_p\vec{b}\right].
$$

Note that conditionally on $X_S$, $U_i$ is Gaussian for each $i \in S$ and:

$$
Y_i := \mathbb{E}\left[U_i \mid X_S\right] = -\lambda_p e_i^T\left(\frac{1}{n}X_S^T X_S\right)^{-1}\vec{b},
$$

$$
Y_i' := \operatorname{Var}\left[U_i \mid X_S\right] = \frac{1}{n^2}e_i^T\left(\frac{1}{n}X_S^T X_S\right)^{-1}X_S^T\Sigma^2 X_S\left(\frac{1}{n}X_S^T X_S\right)^{-1}e_i.
$$

**Lemma D.7.**

*(a) The random variables $Y_i$ and $Y_i'$ have means:*

$$
\mathbb{E}\left[Y_i\right] = \frac{-\lambda_p n}{n-s-1}e_i^T\vec{b}, \quad \text{and} \quad \mathbb{E}\left[Y_i'\right] = \frac{n_1\sigma_1^2 + n_2\sigma_2^2}{n\left(n-s-1\right)}.
$$

*(b) Moreover, each pair $(Y_i, Y_i')$ is concentrated such that:*

$$
\mathbb{P}\left(|Y_i| \geq \frac{n\lambda_p\sqrt{s}}{n-s-1}, \text{ or } |Y_i'| \geq 2\mathbb{E}\left[Y_i'\right]\right) \leq K\left(\frac{1}{n_1} + \frac{1}{n_2}\right),
$$

*where $K$ is a universal constant.*

*Proof.* See appendix D.3.1. $\square$

Now define the event:

$$
T := \bigcup_{i=1}^{s}\left\{|Y_i| \geq \frac{n\lambda_p\sqrt{s}}{n-s-1} \text{ or } |Y_i'| \geq 2\mathbb{E}\left[Y_i'\right]\right\}.
$$

By union bound and statement ($b$) of Lemma D.7, we have:

$$\mathbb{P}\left(T\right) \leq \sum_{i \in S} \mathbb{P}\left(|Y_i| \geq \frac{n\lambda_p\sqrt{s}}{n-s-1}, \text{ or } |Y_i'| \geq 2\mathbb{E}\left[Y_i'\right]\right)$$

$$\leq sK\left(\frac{1}{n_1} + \frac{1}{n_2}\right)$$

$$= K\left(\frac{s}{n_1} + \frac{s}{n_2}\right),$$

which converges to 0 as $p \to +\infty$. Conditionning on $T$, we have by total probability:

$$\mathbb{P}\left(\max_{i \in S} U_i \geq \rho\right) \leq \mathbb{P}\left(\max_{i \in S} U_i \geq \rho \mid T^c\right) + \mathbb{P}\left(T\right)$$

$$\leq \mathbb{P}\left(\max_{i \in S} U_i \geq \rho \mid T^c\right) + K\left(\frac{s}{n_1} + \frac{s}{n_2}\right).$$

In addition:

$$\mathbb{P}\left(\max_{i \in S} U_i \geq \rho \mid T^c\right) = \mathbb{E}\left[\mathbb{1}\left\{\max_{i \in S} U_i \geq \rho\right\} \mid T^c\right]$$

$$= \mathbb{E}\left[\mathbb{E}\left[\mathbb{1}\left\{\max_{i \in S} U_i \geq \rho\right\} \mid T^c, X_S\right]\right]$$

$$= \mathbb{E}\left[\mathbb{P}\left(\max_{i \in S} U_i \geq \rho \mid T^c, X_S\right)\right].$$

Now we have:

$$\mathbb{P}\left(\max_{i \in S} U_i \geq \rho \mid T^c, X_S\right) \leq \mathbb{P}_{N_i \overset{\text{ind}}{\sim} \mathcal{N}\left(Y_i, Y_i'\right)}\left(\max_{i \in S} N_i \geq \rho \mid T^c\right)$$

$$\leq \mathbb{P}_{N_i \overset{\text{i.i.d.}}{\sim} \mathcal{N}\left(\frac{n\lambda_p\sqrt{s}}{n-s-1}, \frac{2\left(n_1\sigma_1^2 + n_2\sigma_2^2\right)}{n(n-s-1)}\right)}\left(\max_{i \in S} N_i \geq \rho\right),$$

where:

- The first inequality holds because the maximum of independent Gaussians has a heavier positive tail than the maximum of correlated ones (under the same distributions).

- The second inequality holds because a Gaussian with a larger mean and variance has a heavier positive tail than one with a smaller mean and variance (therefore each of the $N_i$s would have a heavier tail if its mean and variance were equal to their respective upper bounds).

For simplicity, we drop the long subscript and assume $N_i \overset{\text{i.i.d.}}{\sim} \mathcal{N}\left(\frac{n\lambda_p\sqrt{s}}{n-s-1}, \frac{2\left(n_1\sigma_1^2 + n_2\sigma_2^2\right)}{n(n-s-1)}\right)$. Using Markov's inequality in the above, we have:

$$\mathbb{P}\left(\max_{i \in S} U_i \geq \rho \mid T^c, X_S\right) \leq \mathbb{P}\left(\max_{i \in S} N_i \geq \rho\right) \leq \frac{\mathbb{E}\left[\max_{i \in S} N_i\right]}{\rho}. \tag{59}$$

Using the formula for expectation of Gaussian (see Theorem 5.3.1 in (De Haan & Ferreira, 2006)) maxima, we have:

$$\mathbb{E}\left[\max_{i \in S} N_i\right] \leq \frac{n\lambda_p\sqrt{s}}{n-s-1} + \sqrt{2\log\left(s\right)\frac{2\left(n_1\sigma_1^2 + n_2\sigma_2^2\right)}{n\left(n-s-1\right)}}$$

$$= \frac{n\lambda_p\sqrt{s}}{n-s-1} + 2\sqrt{\frac{\left(n_1\sigma_1^2 + n_2\sigma_2^2\right)\log\left(s\right)}{n\left(n-s-1\right)}}.$$

Therefore, we have:

$$
\mathbb{P}\left(\max_{i\in S} U_i \geq \rho\right) \leq \mathbb{P}\left(\max_{i\in S} U_i \geq \rho \mid T^c\right) + K\left(\frac{s}{n_1} + \frac{s}{n_2}\right)
$$

$$
\leq \mathbb{E}\left[\mathbb{P}\left(\max_{i\in S} U_i \geq \rho \mid T^c, X_S\right)\right] + K\left(\frac{s}{n_1} + \frac{s}{n_2}\right)
$$

$$
\overset{(59)}{\leq} \frac{1}{\rho}\left(\frac{n\lambda_p\sqrt{s}}{n-s-1} + 2\sqrt{\frac{(n_1\sigma_1^2 + n_2\sigma_2^2)\log(s)}{n(n-s-1)}}\right) + K\left(\frac{s}{n_1} + \frac{s}{n_2}\right).
$$

Hence we have:

$$
\mathbb{P}\left(\max_{i\in S} U_i \geq \rho\right) \leq \frac{1}{\rho}\left(\lambda_p\sqrt{s} + \sqrt{\frac{(n_1\sigma_1^2 + n_2\sigma_2^2)\log(s)}{n^2}}\right)(1 + o_p(1)) + o_p(1), \tag{60}
$$

which converges to 0 as $p \to +\infty$ under condition (28). Using a similar argument, we establish the same bound for $\{-U_i\}_{i\in S}$, that:

$$
\mathbb{P}\left(\max_{i\in S}\{-U_i\} \geq \rho\right) \leq \frac{1}{\rho}\left(\lambda_p\sqrt{s} + \sqrt{\frac{(n_1\sigma_1^2 + n_2\sigma_2^2)\log(s)}{n^2}}\right)(1 + o_p(1)) + o_p(1). \tag{61}
$$

Bringing together (60) and (61) and using a union bound, we conclude:

$$
\mathbb{P}\left(\max_{i\in S}|U_i| < \rho\right) \overset{p\to+\infty}{\longrightarrow} 1.
$$

$\square$

### D.3.1 PROOF OF LEMMA D.7

*Proof of Lemma D.7.*

**Mean of $Y_i$.** Recall that:

$$
Y_i = -\lambda_p e_i^T\left(\frac{1}{n}X_S^T X_S\right)^{-1}\vec{b} = -\lambda_p n e_i^T\left(X_S^T X_S\right)^{-1}\vec{b}
$$

Note that $X_S^T X_S \sim \mathcal{W}_s(I_s, n)$. Using properties of the Wishart distribution (see Lemma 7.7.1 of (Anderson et al., 1958)), we have:

$$
\mathbb{E}\left[\left(X_S^T X_S\right)^{-1}\right] = \left(\frac{1}{n-s-1}\right)I_s.
$$

Therefore, we get:

$$
\mathbb{E}[Y_i] = \frac{-\lambda_p n}{n-s-1}e_i^T\vec{b}. \tag{62}
$$

**Mean of $Y_i'$.** Recall that:

$$
Y_i' = \frac{1}{n^2}e_i^T\left(\frac{1}{n}X_S^T X_S\right)^{-1}X_S^T\Sigma^2 X_S\left(\frac{1}{n}X_S^T X_S\right)^{-1}e_i
$$

$$
= e_i^T\left(X_S^T X_S\right)^{-1}X_S^T\Sigma^2 X_S\left(X_S^T X_S\right)^{-1}e_i.
$$

Now recall from (49) in the proof of Lemma D.2 the matrices $Q \in \mathbb{R}^{n\times s}, R \in \mathbb{R}^{s\times s}, U \in \mathbb{R}^{n\times n}$ such that:

$$
X_S = QR, \quad U = [P \quad Q],
$$

where $Q^T Q = I_s$, $R$ is upper triangular and $U \sim \text{Haar}(n)$. We have:

$$
Y_i' = e_i^T\left(X_S^T X_S\right)^{-1}X_S^T\Sigma^2 X_S\left(X_S^T X_S\right)^{-1}e_i
$$

$$
= e_i^T\left(R^T R\right)^{-1}R^T Q^T\Sigma^2 QR\left(R^T R\right)^{-1}e_i
$$

$$
= e_i^T R^{-1}Q^T\Sigma^2 Q\left(R^T\right)^{-1}e_i.
$$

Note that:
$$U^T\Sigma^2 U = \begin{bmatrix} P^T \\ Q^T \end{bmatrix} \Sigma^2 [P \quad Q] = \begin{bmatrix} P^T\Sigma^2 \\ Q^T\Sigma^2 \end{bmatrix} [P \quad Q] = \begin{bmatrix} P^T\Sigma^2 P & P^T\Sigma^2 Q \\ Q^T\Sigma^2 P & Q^T\Sigma^2 Q \end{bmatrix}.$$

Therefore:
$$\mathbb{E}\left[U^T\Sigma^2 U\right] = \begin{bmatrix} \mathbb{E}\left[P^T\Sigma^2 P\right] & \mathbb{E}\left[P^T\Sigma^2 Q\right] \\ \mathbb{E}\left[Q^T\Sigma^2 P\right] & \mathbb{E}\left[Q^T\Sigma^2 Q\right] \end{bmatrix}.$$

On the other hand, we know by the properties of the Haar distribution (see Example 1.8 of (Gu, 2013)) that:
$$\mathbb{E}\left[U^T\Sigma^2 U\right] = \frac{\operatorname{tr}\left(\Sigma^2\right)}{n} I_n = \frac{n_1\sigma_1^2 + n_2\sigma_2^2}{n} I_n.$$

Hence:
$$\mathbb{E}\left[Q^T\Sigma^2 Q\right] = \frac{n_1\sigma_1^2 + n_2\sigma_2^2}{n} I_s.$$

Therefore:
$$\begin{aligned}
\mathbb{E}\left[Y_i'\right] &= \mathbb{E}_{Q,R}\left[e_i^T R^{-1} Q^T\Sigma^2 Q \left(R^T\right)^{-1} e_i\right] \\
&= \mathbb{E}\left[\mathbb{E}\left[e_i^T R^{-1} Q^T\Sigma^2 Q \left(R^T\right)^{-1} e_i \mid R\right]\right] \\
&= \mathbb{E}\left[e_i^T R^{-1}\mathbb{E}\left[Q^T\Sigma^2 Q \mid R\right]\left(R^T\right)^{-1} e_i\right] \\
&= \mathbb{E}\left[e_i^T R^{-1}\mathbb{E}\left[Q^T\Sigma^2 Q\right]\left(R^T\right)^{-1} e_i\right] \\
&= \mathbb{E}\left[e_i^T R^{-1}\left(\frac{n_1\sigma_1^2 + n_2\sigma_2^2}{n} I_s\right)\left(R^T\right)^{-1} e_i\right] \\
&= \left(\frac{n_1\sigma_1^2 + n_2\sigma_2^2}{n}\right)\mathbb{E}\left[e_i^T \left(R^T R\right)^{-1} e_i\right] \\
&= \left(\frac{n_1\sigma_1^2 + n_2\sigma_2^2}{n}\right)\mathbb{E}\left[e_i^T \left(R^T R\right)^{-1} e_i\right] \\
&= \left(\frac{n_1\sigma_1^2 + n_2\sigma_2^2}{n}\right)\mathbb{E}\left[e_i^T \left(X_S^T X_S\right)^{-1} e_i\right] \\
&= \frac{n_1\sigma_1^2 + n_2\sigma_2^2}{n\left(n - s - 1\right)} e_i^T e_i \\
&= \frac{n_1\sigma_1^2 + n_2\sigma_2^2}{n\left(n - s - 1\right)}.
\end{aligned}$$

**Concentration of $Y_i$.** Recall that:
$$Y_i = -\lambda_p n e_i^T \left(X_S^T X_S\right)^{-1} \vec{b}.$$

Thus:
$$Y_i^2 = Y_i Y_i^T = \lambda_p^2 n^2 e_i^T \left(X_S^T X_S\right)^{-1} \vec{b}\vec{b}^T \left(X_S^T X_S\right)^{-1} e_i.$$

Taking the expectation and recalling (50), we have:
$$\begin{aligned}
\mathbb{E}\left[Y_i^2\right] &= \lambda_p^2 n^2 e_i^T\mathbb{E}\left[\left(X_S^T X_S\right)^{-1} \vec{b}\vec{b}^T \left(X_S^T X_S\right)^{-1}\right] e_i \\
&\overset{(50)}{=} \frac{\lambda_p^2 n^2}{(n - s - 3)(n - s)} e_i^T\left[\vec{b}\vec{b}^T + \left\|\vec{b}\right\|_2^2 I_s / (n - s - 1)\right] e_i \\
&= \frac{\lambda_p^2 n^2}{(n - s - 3)(n - s)}\left[\vec{b}_i^2 + \left\|\vec{b}\right\|_2^2 / (n - s - 1)\right] \\
&= \frac{\lambda_p^2 n^2}{(n - s - 3)(n - s)}\left[1 + \frac{s}{n - s - 1}\right],
\end{aligned}$$

where the last equality above holds because $\vec{b} = \text{sign}\left(\beta^\star\right)$ and $i \in S = \text{Supp}\left(\beta^\star\right)$. Thus:

$$\mathbb{E}\left[Y_i^2\right] = \frac{\lambda_p^2 n^2 \left(n - 1\right)}{\left(n - s - 3\right)\left(n - s - 1\right)\left(n - s\right)}.$$

Recalling (62), we get:

$$\begin{aligned}
\text{Var}\left[Y_i\right] &= \frac{\lambda_p^2 n^2 \left(n - 1\right)}{\left(n - s - 3\right)\left(n - s - 1\right)\left(n - s\right)} - \left(\frac{-\lambda_p n}{n - s - 1} e_i^T \vec{b}\right)^2 \\
&= \frac{\lambda_p^2 n^2 \left(n - 1\right)}{\left(n - s - 3\right)\left(n - s - 1\right)\left(n - s\right)} - \frac{\lambda_p^2 n^2 \vec{b}_i^2}{\left(n - s - 1\right)^2} \\
&= \frac{\lambda_p^2 n^2}{\left(n - s - 1\right)} \left[\frac{\left(n - 1\right)}{\left(n - s - 3\right)\left(n - s\right)} - \frac{1}{n - s - 1}\right].
\end{aligned}$$

Therefore:

$$\begin{aligned}
\frac{\text{Var}\left[Y_i\right]}{s\left(\mathbb{E}\left[Y_i\right]\right)^2} &= \frac{\left(n - s - 1\right)^2}{s\lambda_p^2 n^2} \times \frac{\lambda_p^2 n^2}{\left(n - s - 1\right)} \left[\frac{\left(n - 1\right)}{\left(n - s - 3\right)\left(n - s\right)} - \frac{1}{n - s - 1}\right] \\
&= \frac{1}{s} \left(\frac{\left(n - 1\right)\left(n - s - 1\right)}{\left(n - s - 3\right)\left(n - s\right)} - 1\right) \\
&= \frac{n^2 - n - ns + s - n + 1 - n^2 + ns + ns - s^2 + 3n - 3s}{s\left(n - s - 3\right)\left(n - s\right)} \\
&= \frac{1 + ns - s^2 + n - 2s}{s\left(n - s - 3\right)\left(n - s\right)} \\
&= \Theta\left(\frac{1}{n}\right).
\end{aligned}$$

Now using inclusion and Chebyshev's inequality, we have:

$$\begin{aligned}
\mathbb{P}\left(|Y_i| \geq \frac{n\lambda_p \sqrt{s}}{n - s - 1}\right) &\overset{(62)}{\leq} \mathbb{P}\left(\left|Y_i - \mathbb{E}\left[Y_i\right]\right| \geq \left|\mathbb{E}\left[Y_i\right]\right|\left(\sqrt{s} - 1\right)\right) \\
&\leq \frac{\text{Var}\left[Y_i\right]}{\left(\sqrt{s} - 1\right)^2 \left(\mathbb{E}\left[Y_i\right]\right)^2} \\
&= \Theta\left(\frac{\text{Var}\left[Y_i\right]}{s\left(\mathbb{E}\left[Y_i\right]\right)^2}\right) \\
&= \Theta\left(\frac{1}{n}\right).
\end{aligned}$$

In particular, the above implies:

$$\mathbb{P}\left(|Y_i| \geq \frac{n\lambda_p \sqrt{s}}{n - s - 1}\right) = \mathcal{O}\left(\frac{1}{n_1} + \frac{1}{n_2}\right).$$

Therefore, the exists a universal constant $K_1 > 0$ such that:

$$\mathbb{P}\left(|Y_i| \geq \frac{n\lambda_p \sqrt{s}}{n - s - 1}\right) \leq K_1 \left(\frac{1}{n_1} + \frac{1}{n_2}\right). \tag{63}$$

**Concentration of $Y_i'$.** We have:

$$\begin{aligned}
\mathbb{E}\left[\left(Y_i'\right)^2 \mid R\right] &= \mathbb{E}\left[\left(e_i^T R^{-1} Q^T \Sigma^2 Q \left(R^T\right)^{-1} e_i\right)^2 \mid R\right] \\
&= \mathbb{E}\left[\left(\left[R^{-1} Q^T \Sigma^2 Q \left(R^T\right)^{-1}\right]_{ii}\right)^2 \mid R\right]
\end{aligned}$$

We have, by Einstein notation:

$$
\begin{aligned}
\left[ R^{-1} Q^T \Sigma^2 Q \left( R^T \right)^{-1} \right]_{ii} &= \sum_{a=1}^{s} \sum_{b=1}^{n} \sum_{c=1}^{n} \sum_{e=1}^{s} \left[ R^{-1} \right]_{ia} \left[ Q^T \right]_{ab} \left[ \Sigma^2 \right]_{bc} Q_{ce} \left[ \left( R^T \right)^{-1} \right]_{ei} \\
&= \sum_{a=1}^{s} \sum_{b=1}^{n} \sum_{c=1}^{n} \sum_{e=1}^{s} \left[ R^{-1} \right]_{ia} \left[ Q^T \right]_{ab} \left[ \Sigma^2 \right]_{bc} Q_{ce} \left[ \left( R^{-1} \right)^T \right]_{ei} \\
&= \sum_{a=1}^{s} \sum_{b=1}^{n} \sum_{c=1}^{n} \sum_{e=1}^{s} \left[ R^{-1} \right]_{ia} \left[ Q^T \right]_{ab} \left[ \Sigma^2 \right]_{bc} Q_{ce} \left[ R^{-1} \right]_{ie} \\
&= \sum_{a=1}^{s} \sum_{c=1}^{n} \sum_{e=1}^{s} \left[ R^{-1} \right]_{ia} Q_{ca} \Sigma_{cc}^2 Q_{ce} \left[ R^{-1} \right]_{ie} \\
&= \sum_{c=1}^{n} \sum_{a=1}^{s} \sum_{e=1}^{s} \Sigma_{cc}^2 \left[ R^{-1} \right]_{ia} \left[ R^{-1} \right]_{ie} Q_{ca} Q_{ce}.
\end{aligned}
$$

Therefore:

$$
\begin{aligned}
&\left( \left[ R^{-1} Q^T \Sigma^2 Q \left( R^T \right)^{-1} \right]_{ii} \right)^2 \\
&= \left( \sum_{c=1}^{n} \sum_{a=1}^{s} \sum_{e=1}^{s} \Sigma_{cc}^2 \left[ R^{-1} \right]_{ia} \left[ R^{-1} \right]_{ie} Q_{ca} Q_{ce} \right)^2 \\
&= \sum_{c=1}^{n} \sum_{a=1}^{s} \sum_{e=1}^{s} \left( \Sigma_{cc}^2 \left[ R^{-1} \right]_{ia} \left[ R^{-1} \right]_{ie} Q_{ca} Q_{ce} \right)^2 \\
&\quad + \sum_{c=1}^{n} \sum_{a=1}^{s} \sum_{e \neq f \in [s]} \left( \Sigma_{cc}^2 \left[ R^{-1} \right]_{ia} \left[ R^{-1} \right]_{ie} Q_{ca} Q_{ce} \right) \left( \Sigma_{cc}^2 \left[ R^{-1} \right]_{ia} \left[ R^{-1} \right]_{if} Q_{ca} Q_{cf} \right) \\
&\quad + \sum_{c=1}^{n} \sum_{a \neq b \in [s]} \sum_{e=1}^{s} \left( \Sigma_{cc}^2 \left[ R^{-1} \right]_{ia} \left[ R^{-1} \right]_{ie} Q_{ca} Q_{ce} \right) \left( \Sigma_{cc}^2 \left[ R^{-1} \right]_{ib} \left[ R^{-1} \right]_{ie} Q_{cb} Q_{ce} \right) \\
&\quad + \sum_{c=1}^{n} \sum_{a \neq b \in [s]} \sum_{e \neq f \in [s]} \left( \Sigma_{cc}^2 \left[ R^{-1} \right]_{ia} \left[ R^{-1} \right]_{ie} Q_{ca} Q_{ce} \right) \left( \Sigma_{cc}^2 \left[ R^{-1} \right]_{ib} \left[ R^{-1} \right]_{if} Q_{cb} Q_{cf} \right) \\
&\quad + \sum_{c \neq d \in [n]} \sum_{a=1}^{s} \sum_{e=1}^{s} \left( \Sigma_{cc}^2 \left[ R^{-1} \right]_{ia} \left[ R^{-1} \right]_{ie} Q_{ca} Q_{ce} \right) \left( \Sigma_{dd}^2 \left[ R^{-1} \right]_{ia} \left[ R^{-1} \right]_{ie} Q_{da} Q_{de} \right) \\
&\quad + \sum_{c \neq d \in [n]} \sum_{a=1}^{s} \sum_{e \neq f \in [s]} \left( \Sigma_{cc}^2 \left[ R^{-1} \right]_{ia} \left[ R^{-1} \right]_{ie} Q_{ca} Q_{ce} \right) \left( \Sigma_{dd}^2 \left[ R^{-1} \right]_{ia} \left[ R^{-1} \right]_{if} Q_{da} Q_{df} \right) \\
&\quad + \sum_{c \neq d \in [n]} \sum_{a \neq b \in [s]} \sum_{e=1}^{s} \left( \Sigma_{cc}^2 \left[ R^{-1} \right]_{ia} \left[ R^{-1} \right]_{ie} Q_{ca} Q_{ce} \right) \left( \Sigma_{dd}^2 \left[ R^{-1} \right]_{ib} \left[ R^{-1} \right]_{ie} Q_{db} Q_{de} \right) \\
&\quad + \sum_{c \neq d \in [n]} \sum_{a \neq b \in [s]} \sum_{e \neq f \in [s]} \left( \Sigma_{cc}^2 \left[ R^{-1} \right]_{ia} \left[ R^{-1} \right]_{ie} Q_{ca} Q_{ce} \right) \left( \Sigma_{dd}^2 \left[ R^{-1} \right]_{ib} \left[ R^{-1} \right]_{if} Q_{db} Q_{df} \right)
\end{aligned}
$$

$$
\begin{aligned}
= & \sum_{c=1}^{n} \sum_{a=1}^{s} \sum_{e=1}^{s} \Sigma_{cc}^4 \left[R^{-1}\right]_{ia}^2 \left[R^{-1}\right]_{ie}^2 Q_{ca}^2 Q_{ce}^2 \\
& + \sum_{c=1}^{n} \sum_{a=1}^{s} \sum_{e \neq f \in [s]} \Sigma_{cc}^4 \left[R^{-1}\right]_{ia}^2 \left[R^{-1}\right]_{ie} \left[R^{-1}\right]_{if} Q_{ca}^2 Q_{ce} Q_{cf} \\
& + \sum_{c=1}^{n} \sum_{a \neq b \in [s]} \sum_{e=1}^{s} \Sigma_{cc}^4 \left[R^{-1}\right]_{ia} \left[R^{-1}\right]_{ib} \left[R^{-1}\right]_{ie}^2 Q_{ca} Q_{cb} Q_{ce}^2 \\
& + \sum_{c=1}^{n} \sum_{a \neq b \in [s]} \sum_{e \neq f \in [s]} \Sigma_{cc}^4 \left[R^{-1}\right]_{ia} \left[R^{-1}\right]_{ib} \left[R^{-1}\right]_{ie} \left[R^{-1}\right]_{if} Q_{ca} Q_{cb} Q_{ce} Q_{cf} \\
& + \sum_{c \neq d \in [n]} \sum_{a=1}^{s} \sum_{e=1}^{s} \Sigma_{cc}^2 \Sigma_{dd}^2 \left[R^{-1}\right]_{ia}^2 \left[R^{-1}\right]_{ie}^2 Q_{ca} Q_{ce} Q_{da} Q_{de} \\
& + \sum_{c \neq d \in [n]} \sum_{a=1}^{s} \sum_{e \neq f \in [s]} \Sigma_{cc}^2 \Sigma_{dd}^2 \left[R^{-1}\right]_{ia}^2 \left[R^{-1}\right]_{ie} \left[R^{-1}\right]_{if} Q_{ca} Q_{ce} Q_{da} Q_{df} \\
& + \sum_{c \neq d \in [n]} \sum_{a \neq b \in [s]} \sum_{e=1}^{s} \Sigma_{cc}^2 \Sigma_{dd}^2 \left[R^{-1}\right]_{ia} \left[R^{-1}\right]_{ib} \left[R^{-1}\right]_{ie}^2 Q_{ca} Q_{ce} Q_{db} Q_{de} \\
& + \sum_{c \neq d \in [n]} \sum_{a \neq b \in [s]} \sum_{e \neq f \in [s]} \Sigma_{cc}^2 \Sigma_{dd}^2 \left[R^{-1}\right]_{ia} \left[R^{-1}\right]_{ib} \left[R^{-1}\right]_{ie} \left[R^{-1}\right]_{if} Q_{ca} Q_{ce} Q_{db} Q_{df}
\end{aligned}
$$

$$
\begin{aligned}
= & \sum_{c=1}^{n} \sum_{a=1}^{s} \sum_{e=1}^{s} \Sigma_{cc}^4 \left[R^{-1}\right]_{ia}^2 \left[R^{-1}\right]_{ie}^2 Q_{ca}^2 Q_{ce}^2 \\
& + 2 \sum_{c=1}^{n} \sum_{a=1}^{s} \sum_{e \neq f \in [s]} \Sigma_{cc}^4 \left[R^{-1}\right]_{ia}^2 \left[R^{-1}\right]_{ie} \left[R^{-1}\right]_{if} Q_{ca}^2 Q_{ce} Q_{cf} \\
& + \sum_{c=1}^{n} \sum_{a \neq b \in [s]} \sum_{e \neq f \in [s]} \Sigma_{cc}^4 \left[R^{-1}\right]_{ia} \left[R^{-1}\right]_{ib} \left[R^{-1}\right]_{ie} \left[R^{-1}\right]_{if} Q_{ca} Q_{cb} Q_{ce} Q_{cf} \\
& + \sum_{c \neq d \in [n]} \sum_{a=1}^{s} \sum_{e=1}^{s} \Sigma_{cc}^2 \Sigma_{dd}^2 \left[R^{-1}\right]_{ia}^2 \left[R^{-1}\right]_{ie}^2 Q_{ca} Q_{ce} Q_{da} Q_{de} \\
& + 2 \sum_{c \neq d \in [n]} \sum_{a=1}^{s} \sum_{e \neq f \in [s]} \Sigma_{cc}^2 \Sigma_{dd}^2 \left[R^{-1}\right]_{ia}^2 \left[R^{-1}\right]_{ie} \left[R^{-1}\right]_{if} Q_{ca} Q_{ce} Q_{da} Q_{df} \\
& + \sum_{c \neq d \in [n]} \sum_{a \neq b \in [s]} \sum_{e \neq f \in [s]} \Sigma_{cc}^2 \Sigma_{dd}^2 \left[R^{-1}\right]_{ia} \left[R^{-1}\right]_{ib} \left[R^{-1}\right]_{ie} \left[R^{-1}\right]_{if} Q_{ca} Q_{ce} Q_{db} Q_{df}.
\end{aligned}
$$

Taking the expectation of the above conditionally on $R$ and by independence of $Q$ and $R$, we get:

$$\mathbb{E}\left[\left(Y_i'\right)^2 \mid R\right]$$

$$= \mathbb{E}\left[\left(\left[R^{-1}Q^T\Sigma^2 Q\left(R^T\right)^{-1}\right]_{ii}\right)^2 \mid R\right]$$

$$= \sum_{c=1}^{n}\sum_{a=1}^{s}\sum_{e=1}^{s}\Sigma_{cc}^4\left[R^{-1}\right]_{ia}^2\left[R^{-1}\right]_{ie}^2\mathbb{E}\left[Q_{ca}^2 Q_{ce}^2\right]$$

$$+ 2\sum_{c=1}^{n}\sum_{a=1}^{s}\sum_{e\neq f\in[s]}\Sigma_{cc}^4\left[R^{-1}\right]_{ia}^2\left[R^{-1}\right]_{ie}\left[R^{-1}\right]_{if}\mathbb{E}\left[Q_{ca}^2 Q_{ce}Q_{cf}\right]$$

$$+ \sum_{c=1}^{n}\sum_{a\neq b\in[s]}\sum_{e\neq f\in[s]}\Sigma_{cc}^4\left[R^{-1}\right]_{ia}\left[R^{-1}\right]_{ib}\left[R^{-1}\right]_{ie}\left[R^{-1}\right]_{if}\mathbb{E}\left[Q_{ca}Q_{cb}Q_{ce}Q_{cf}\right]$$

$$+ \sum_{c\neq d\in[n]}\sum_{a=1}^{s}\sum_{e=1}^{s}\Sigma_{cc}^2\Sigma_{dd}^2\left[R^{-1}\right]_{ia}^2\left[R^{-1}\right]_{ie}^2\mathbb{E}\left[Q_{ca}Q_{ce}Q_{da}Q_{de}\right]$$

$$+ 2\sum_{c\neq d\in[n]}\sum_{a=1}^{s}\sum_{e\neq f\in[s]}\Sigma_{cc}^2\Sigma_{dd}^2\left[R^{-1}\right]_{ia}^2\left[R^{-1}\right]_{ie}\left[R^{-1}\right]_{if}\mathbb{E}\left[Q_{ca}Q_{ce}Q_{da}Q_{df}\right]$$

$$+ \sum_{c\neq d\in[n]}\sum_{a\neq b\in[s]}\sum_{e\neq f\in[s]}\Sigma_{cc}^2\Sigma_{dd}^2\left[R^{-1}\right]_{ia}\left[R^{-1}\right]_{ib}\left[R^{-1}\right]_{ie}\left[R^{-1}\right]_{if}\mathbb{E}\left[Q_{ca}Q_{ce}Q_{db}Q_{df}\right],$$

Now recall from Lemma D.6 above (by Meckes (2019)) that for all $i,j,r,s,\alpha,\beta,\lambda,\mu\in[n]$ we have:

$$\mathbb{E}\left[U_{ij}U_{rs}U_{\alpha\beta}U_{\lambda\mu}\right]$$

$$= -\frac{1}{(n-1)n(n+2)}\Big[\delta_{ir}\delta_{\alpha\lambda}\delta_{j\beta}\delta_{s\mu} + \delta_{ir}\delta_{\alpha\lambda}\delta_{j\mu}\delta_{s\beta} + \delta_{i\alpha}\delta_{r\lambda}\delta_{js}\delta_{\beta\mu}$$

$$+ \delta_{i\alpha}\delta_{r\lambda}\delta_{j\mu}\delta_{\beta s} + \delta_{i\lambda}\delta_{r\alpha}\delta_{js}\delta_{\beta\mu} + \delta_{i\lambda}\delta_{r\alpha}\delta_{j\beta}\delta_{s\mu}\Big]$$

$$+ \frac{n+1}{(n-1)n(n+2)}\Big[\delta_{ir}\delta_{\alpha\lambda}\delta_{js}\delta_{\beta\mu} + \delta_{i\alpha}\delta_{r\lambda}\delta_{j\beta}\delta_{s\mu} + \delta_{i\lambda}\delta_{r\alpha}\delta_{j\mu}\delta_{s\beta}\Big].$$

Also, recall from (49) that:

$$U = \begin{bmatrix} P & Q \end{bmatrix},$$

hence for any $\alpha\in[n],\beta\in[s]$:

$$Q_{\alpha\beta} = U_{\alpha(n-s+\beta)}. \tag{64}$$

Since the above fourth-order formula only depends on indices through Kronecker deltas, it holds that for any $i,r,\alpha,\lambda\in[n]$, $j,s,\alpha,\beta,\mu\in[s]$:

$$\mathbb{E}\left[Q_{ij}Q_{rs}Q_{\alpha\beta}Q_{\lambda\mu}\right]$$

$$= -\frac{1}{(n-1)n(n+2)}\Big[\delta_{ir}\delta_{\alpha\lambda}\delta_{j\beta}\delta_{s\mu} + \delta_{ir}\delta_{\alpha\lambda}\delta_{j\mu}\delta_{s\beta} + \delta_{i\alpha}\delta_{r\lambda}\delta_{js}\delta_{\beta\mu}$$

$$+ \delta_{i\alpha}\delta_{r\lambda}\delta_{j\mu}\delta_{\beta s} + \delta_{i\lambda}\delta_{r\alpha}\delta_{js}\delta_{\beta\mu} + \delta_{i\lambda}\delta_{r\alpha}\delta_{j\beta}\delta_{s\mu}\Big]$$

$$+ \frac{n+1}{(n-1)n(n+2)}\Big[\delta_{ir}\delta_{\alpha\lambda}\delta_{js}\delta_{\beta\mu} + \delta_{i\alpha}\delta_{r\lambda}\delta_{j\beta}\delta_{s\mu} + \delta_{i\lambda}\delta_{r\alpha}\delta_{j\mu}\delta_{s\beta}.\Big]$$

In addition, note that the expression above is equal to zero when one of $\{j, s, \alpha, \mu\}$ is different than all the others. This observation simplifies the expression of $\mathbb{E}\left[(Y_i')^2 \mid R\right]$ above as follows:

$$
\begin{aligned}
& \mathbb{E}\left[(Y_i')^2 \mid R\right] \\
&= \sum_{c=1}^{n}\sum_{a=1}^{s}\sum_{e=1}^{s} \Sigma_{cc}^4 \left[R^{-1}\right]_{ia}^2 \left[R^{-1}\right]_{ie}^2 \mathbb{E}\left[Q_{ca}^2 Q_{ce}^2\right] \\
&\quad + \sum_{c=1}^{n}\sum_{a\neq b\in[s]}\sum_{e\neq f\in[s]} \Sigma_{cc}^4 \left[R^{-1}\right]_{ia}\left[R^{-1}\right]_{ib}\left[R^{-1}\right]_{ie}\left[R^{-1}\right]_{if} \mathbb{E}\left[Q_{ca}Q_{cb}Q_{ce}Q_{cf}\right] \\
&\quad + \sum_{c\neq d\in[n]}\sum_{a=1}^{s}\sum_{e=1}^{s} \Sigma_{cc}^2\Sigma_{dd}^2 \left[R^{-1}\right]_{ia}^2 \left[R^{-1}\right]_{ie}^2 \mathbb{E}\left[Q_{ca}Q_{ce}Q_{da}Q_{de}\right] \\
&\quad + \sum_{c\neq d\in[n]}\sum_{a\neq b\in[s]}\sum_{e\neq f\in[s]} \Sigma_{cc}^2\Sigma_{dd}^2 \left[R^{-1}\right]_{ia}\left[R^{-1}\right]_{ib}\left[R^{-1}\right]_{ie}\left[R^{-1}\right]_{if} \mathbb{E}\left[Q_{ca}Q_{ce}Q_{db}Q_{df}\right].
\end{aligned}
$$

Thus:

$$
\begin{aligned}
& \mathbb{E}\left[(Y_i')^2 \mid R\right] \\
&= \sum_{c=1}^{n}\sum_{a=1}^{s} \Sigma_{cc}^4 \left[R^{-1}\right]_{ia}^4 \mathbb{E}\left[Q_{ca}^4\right] \\
&\quad + \sum_{c=1}^{n}\sum_{a\neq e\in[s]} \Sigma_{cc}^4 \left[R^{-1}\right]_{ia}^2 \left[R^{-1}\right]_{ie}^2 \mathbb{E}\left[Q_{ca}^2 Q_{ce}^2\right] \\
&\quad + \sum_{c=1}^{n}\sum_{a\neq b\in[s]}\sum_{e\neq f\in[s]} \Sigma_{cc}^4 \left[R^{-1}\right]_{ia}\left[R^{-1}\right]_{ib}\left[R^{-1}\right]_{ie}\left[R^{-1}\right]_{if} \\
&\hspace{6cm} \times \mathbb{E}\left[Q_{ca}Q_{cb}Q_{ce}Q_{cf}\right] \mathbb{1}\left\{(a,b)=(e,f)\right\} \\
&\quad + \sum_{c=1}^{n}\sum_{a\neq b\in[s]}\sum_{e\neq f\in[s]} \Sigma_{cc}^4 \left[R^{-1}\right]_{ia}\left[R^{-1}\right]_{ib}\left[R^{-1}\right]_{ie}\left[R^{-1}\right]_{if} \\
&\hspace{6cm} \times \mathbb{E}\left[Q_{ca}Q_{cb}Q_{ce}Q_{cf}\right] \mathbb{1}\left\{(a,b)=(f,e)\right\} \\
&\quad + \sum_{c\neq d\in[n]}\sum_{a=1}^{s} \Sigma_{cc}^2\Sigma_{dd}^2 \left[R^{-1}\right]_{ia}^4 \mathbb{E}\left[Q_{ca}^2 Q_{da}^2\right] \\
&\quad + \sum_{c\neq d\in[n]}\sum_{a\neq e\in[s]} \Sigma_{cc}^2\Sigma_{dd}^2 \left[R^{-1}\right]_{ia}^2 \left[R^{-1}\right]_{ie}^2 \mathbb{E}\left[Q_{ca}Q_{ce}Q_{da}Q_{de}\right] \\
&\quad + \sum_{c\neq d\in[n]}\sum_{a\neq b\in[s]}\sum_{e\neq f\in[s]} \Sigma_{cc}^2\Sigma_{dd}^2 \left[R^{-1}\right]_{ia}\left[R^{-1}\right]_{ib}\left[R^{-1}\right]_{ie}\left[R^{-1}\right]_{if} \\
&\hspace{6cm} \times \mathbb{E}\left[Q_{ca}Q_{ce}Q_{db}Q_{df}\right] \mathbb{1}\left\{(a,b)=(e,f)\right\} \\
&\quad + \sum_{c\neq d\in[n]}\sum_{a\neq b\in[s]}\sum_{e\neq f\in[s]} \Sigma_{cc}^2\Sigma_{dd}^2 \left[R^{-1}\right]_{ia}\left[R^{-1}\right]_{ib}\left[R^{-1}\right]_{ie}\left[R^{-1}\right]_{if} \\
&\hspace{6cm} \times \mathbb{E}\left[Q_{ca}Q_{ce}Q_{db}Q_{df}\right] \mathbb{1}\left\{(a,b)=(f,e)\right\}.
\end{aligned}
$$

Thus:

$$
\begin{aligned}
\mathbb{E}\left[\left(Y_i'\right)^2 \mid R\right] = & \sum_{c=1}^{n}\sum_{a=1}^{s} \Sigma_{cc}^4 \left[R^{-1}\right]_{ia}^4 \mathbb{E}\left[Q_{ca}^4\right] \\
& + \sum_{c=1}^{n}\sum_{a\neq e\in[s]} \Sigma_{cc}^4 \left[R^{-1}\right]_{ia}^2 \left[R^{-1}\right]_{ie}^2 \mathbb{E}\left[Q_{ca}^2 Q_{ce}^2\right] \\
& + \sum_{c=1}^{n}\sum_{a\neq b\in[s]} \Sigma_{cc}^4 \left[R^{-1}\right]_{ia}^2 \left[R^{-1}\right]_{ib}^2 \mathbb{E}\left[Q_{ca}^2 Q_{cb}^2\right] \\
& + \sum_{c=1}^{n}\sum_{a\neq b\in[s]} \Sigma_{cc}^4 \left[R^{-1}\right]_{ia}^2 \left[R^{-1}\right]_{ib}^2 \mathbb{E}\left[Q_{ca}^2 Q_{cb}^2\right] \\
& + \sum_{c\neq d\in[n]}\sum_{a=1}^{s} \Sigma_{cc}^2\Sigma_{dd}^2 \left[R^{-1}\right]_{ia}^4 \mathbb{E}\left[Q_{ca}^2 Q_{da}^2\right] \\
& + \sum_{c\neq d\in[n]}\sum_{a\neq e\in[s]} \Sigma_{cc}^2\Sigma_{dd}^2 \left[R^{-1}\right]_{ia}^2 \left[R^{-1}\right]_{ie}^2 \mathbb{E}\left[Q_{ca}Q_{ce}Q_{da}Q_{de}\right] \\
& + \sum_{c\neq d\in[n]}\sum_{a\neq b\in[s]} \Sigma_{cc}^2\Sigma_{dd}^2 \left[R^{-1}\right]_{ia}^2 \left[R^{-1}\right]_{ib}^2 \mathbb{E}\left[Q_{ca}^2 Q_{db}^2\right] \\
& + \sum_{c\neq d\in[n]}\sum_{a\neq b\in[s]} \Sigma_{cc}^2\Sigma_{dd}^2 \left[R^{-1}\right]_{ia}^2 \left[R^{-1}\right]_{ib}^2 \mathbb{E}\left[Q_{ca}Q_{cb}Q_{db}Q_{da}\right].
\end{aligned}
$$

Hence:

$$
\begin{aligned}
\mathbb{E}\left[\left(Y_i'\right)^2 \mid R\right] = & \sum_{c=1}^{n}\sum_{a=1}^{s} \Sigma_{cc}^4 \left[R^{-1}\right]_{ia}^4 \mathbb{E}\left[Q_{ca}^4\right] \\
& + 3\sum_{c=1}^{n}\sum_{a\neq b\in[s]} \Sigma_{cc}^4 \left[R^{-1}\right]_{ia}^2 \left[R^{-1}\right]_{ib}^2 \mathbb{E}\left[Q_{ca}^2 Q_{cb}^2\right] \\
& + \sum_{c\neq d\in[n]}\sum_{a=1}^{s} \Sigma_{cc}^2\Sigma_{dd}^2 \left[R^{-1}\right]_{ia}^4 \mathbb{E}\left[Q_{ca}^2 Q_{da}^2\right] \\
& + \sum_{c\neq d\in[n]}\sum_{a\neq b\in[s]} \Sigma_{cc}^2\Sigma_{dd}^2 \left[R^{-1}\right]_{ia}^2 \left[R^{-1}\right]_{ib}^2 \mathbb{E}\left[Q_{ca}^2 Q_{db}^2\right] \\
& + 2\sum_{c\neq d\in[n]}\sum_{a\neq b\in[s]} \Sigma_{cc}^2\Sigma_{dd}^2 \left[R^{-1}\right]_{ia}^2 \left[R^{-1}\right]_{ib}^2 \mathbb{E}\left[Q_{ca}Q_{cb}Q_{db}Q_{da}\right].
\end{aligned}
$$

Now recalling (54), (55), (56), (57), (58), and using (64) we have:

$$
\begin{aligned}
\mathbb{E}\left[Q_{ca}^4\right] &= \frac{3}{n(n+2)} \\
\mathbb{E}\left[Q_{ca}^2 Q_{cb}^2\right] &= \mathbb{E}\left[Q_{ca}^2 Q_{da}^2\right] = \frac{1}{n(n+2)} \\
\mathbb{E}\left[Q_{ca}^2 Q_{db}^2\right] &= \frac{n+1}{(n-1)n(n+2)} \\
\mathbb{E}\left[Q_{ca}Q_{cb}Q_{db}Q_{da}\right] &= -\frac{1}{(n-1)n(n+2)}.
\end{aligned}
$$

Therefore:

$$
\mathbb{E}\left[\left(Y_i'\right)^2 \mid R\right] = \sum_{c=1}^{n}\sum_{a=1}^{s} \Sigma_{cc}^4 \left[R^{-1}\right]_{ia}^4 \mathbb{E}\left[Q_{ca}^4\right]
$$

$$
+ 3\sum_{c=1}^{n}\sum_{a\neq b\in[s]} \Sigma_{cc}^4 \left[R^{-1}\right]_{ia}^2 \left[R^{-1}\right]_{ib}^2 \mathbb{E}\left[Q_{ca}^2 Q_{cb}^2\right]
$$

$$
+ \sum_{c\neq d\in[n]}\sum_{a=1}^{s} \Sigma_{cc}^2\Sigma_{dd}^2 \left[R^{-1}\right]_{ia}^4 \mathbb{E}\left[Q_{ca}^2 Q_{da}^2\right]
$$

$$
+ \sum_{c\neq d\in[n]}\sum_{a\neq b\in[s]} \Sigma_{cc}^2\Sigma_{dd}^2 \left[R^{-1}\right]_{ia}^2 \left[R^{-1}\right]_{ib}^2 \mathbb{E}\left[Q_{ca}^2 Q_{db}^2\right]
$$

$$
+ 2\sum_{c\neq d\in[n]}\sum_{a\neq b\in[s]} \Sigma_{cc}^2\Sigma_{dd}^2 \left[R^{-1}\right]_{ia}^2 \left[R^{-1}\right]_{ib}^2 \mathbb{E}\left[Q_{ca} Q_{cb} Q_{db} Q_{da}\right]
$$

Thus:

$$
\mathbb{E}\left[\left(Y_i'\right)^2 \mid R\right] = \frac{3}{n(n+2)}\sum_{c=1}^{n}\sum_{a=1}^{s}\Sigma_{cc}^4\left[R^{-1}\right]_{ia}^4
$$

$$
+ \frac{3}{n(n+2)}\sum_{c=1}^{n}\sum_{a\neq b\in[s]}\Sigma_{cc}^4\left[R^{-1}\right]_{ia}^2\left[R^{-1}\right]_{ib}^2
$$

$$
+ \frac{1}{n(n+2)}\sum_{c\neq d\in[n]}\sum_{a=1}^{s}\Sigma_{cc}^2\Sigma_{dd}^2\left[R^{-1}\right]_{ia}^4
$$

$$
+ \frac{n+1}{(n-1)n(n+2)}\sum_{c\neq d\in[n]}\sum_{a\neq b\in[s]}\Sigma_{cc}^2\Sigma_{dd}^2\left[R^{-1}\right]_{ia}^2\left[R^{-1}\right]_{ib}^2
$$

$$
- \frac{2}{(n-1)n(n+2)}\sum_{c\neq d\in[n]}\sum_{a\neq b\in[s]}\Sigma_{cc}^2\Sigma_{dd}^2\left[R^{-1}\right]_{ia}^2\left[R^{-1}\right]_{ib}^2
$$

Thus:

$$
\mathbb{E}\left[\left(Y_i'\right)^2 \mid R\right]
$$

$$
= \frac{3}{n(n+2)}\left(\sum_{c=1}^{n}\Sigma_{cc}^4\right)\left\{\sum_{a=1}^{s}\left[R^{-1}\right]_{ia}^4 + \sum_{a\neq b\in[s]}\left[R^{-1}\right]_{ia}^2\left[R^{-1}\right]_{ib}^2\right\}
$$

$$
+ \frac{1}{n(n+2)}\left(\sum_{c\neq d\in[n]}\Sigma_{cc}^2\Sigma_{dd}^2\right)\left\{\sum_{a=1}^{s}\left[R^{-1}\right]_{ia}^4 + \sum_{a\neq b\in[s]}\left[R^{-1}\right]_{ia}^2\left[R^{-1}\right]_{ib}^2\right\}
$$

$$
= \frac{1}{n(n+2)}\left\{3\sum_{c=1}^{n}\Sigma_{cc}^4 + \sum_{c\neq d\in[n]}\Sigma_{cc}^2\Sigma_{dd}^2\right\}\left\{\sum_{a=1}^{s}\left[R^{-1}\right]_{ia}^4 + \sum_{a\neq b\in[s]}\left[R^{-1}\right]_{ia}^2\left[R^{-1}\right]_{ib}^2\right\}
$$

$$
= \frac{1}{n(n+2)}\left\{3\sum_{c=1}^{n}\Sigma_{cc}^4 + \left[\left(\sum_{c=1}^{n}\Sigma_{cc}^2\right)^2 - \sum_{c=1}^{n}\Sigma_{cc}^4\right]\right\}\left(\sum_{a=1}^{s}\left[R^{-1}\right]_{ia}^2\right)^2
$$

$$
= \frac{1}{n(n+2)}\left\{2\operatorname{tr}\left(\Sigma^4\right) + \operatorname{tr}\left(\Sigma^2\right)^2\right\}\left(\sum_{a=1}^{s}\left[R^{-1}\right]_{ia}^2\right)^2 .
$$

Now note that:

$$
\begin{aligned}
e_i^T \left( X_S^T X_S \right)^{-1} e_i &= \left[ \left( R^T R \right)^{-1} \right]_{ii} \\
&= \left[ R^{-1} \left( R^{-1} \right)^T \right]_{ii} \\
&= \sum_{a=1}^s \left[ R^{-1} \right]_{ia} \left[ \left( R^{-1} \right)^T \right]_{ai} \\
&= \sum_{a=1}^s \left[ R^{-1} \right]_{ia} \left[ R^{-1} \right]_{ia} \\
&= \sum_{a=1}^s \left[ R^{-1} \right]_{ia}^2 .
\end{aligned}
$$

On the other hand, recall that $X_S^T X_S \sim \mathcal{W}_s \left( I_s, n \right)$. Setting $b := n$; $a := s$; $t := e_i$; $T := I_s$ and $A := X_S^T X_S$ in Lemma D.4, we get:

$$
\begin{aligned}
\mathbb{E} \left[ \left( X_S^T X_S \right)^{-1} e_i e_i^T \left( X_S^T X_S \right)^{-1} \right] &= \frac{1}{(n-s)(n-s-3)} \left( e_i e_i^T + \left( e_i^T e_i \right) I_s / (n-s-1) \right) \\
&= \frac{1}{(n-s)(n-s-3)} \left( e_i e_i^T + \frac{1}{n-s-1} I_s \right) .
\end{aligned}
$$

Therefore:

$$
\begin{aligned}
\mathbb{E} \left[ \left( e_i^T \left( X_S^T X_S \right)^{-1} e_i \right)^2 \right] &= e_i^T \mathbb{E} \left[ \left( X_S^T X_S \right)^{-1} e_i e_i^T \left( X_S^T X_S \right)^{-1} \right] e_i \\
&= \left[ \mathbb{E} \left[ \left( X_S^T X_S \right)^{-1} e_i e_i^T \left( X_S^T X_S \right)^{-1} \right] \right]_{ii} \\
&= \left[ \frac{1}{(n-s)(n-s-3)} \left( e_i e_i^T + \frac{1}{n-s-1} I_s \right) \right]_{ii} .
\end{aligned}
$$

Hence:

$$
\mathbb{E} \left[ \left( \sum_{a=1}^s \left[ R^{-1} \right]_{ia}^2 \right)^2 \right] = \mathbb{E} \left[ \left( e_i^T \left( X_S^T X_S \right)^{-1} e_i \right)^2 \right] = \frac{1}{(n-s)(n-s-3)} \left( 1 + \frac{1}{n-s-1} \right) .
$$

Hence, we get:

$$
\begin{aligned}
& \mathbb{E} \left[ \left( Y_i' \right)^2 \right] \\
&= \mathbb{E} \left[ \mathbb{E} \left[ \left( Y_i' \right)^2 \mid R \right] \right] \\
&= \mathbb{E} \left[ \frac{1}{n(n+2)} \left\{ 2 \operatorname{tr} \left( \Sigma^4 \right) + \operatorname{tr} \left( \Sigma^2 \right)^2 \right\} \left( \sum_{a=1}^s \left[ R^{-1} \right]_{ia}^2 \right)^2 \right] \\
&= \frac{1}{n(n+2)} \left\{ 2 \operatorname{tr} \left( \Sigma^4 \right) + \operatorname{tr} \left( \Sigma^2 \right)^2 \right\} \mathbb{E} \left[ \left( \sum_{a=1}^s \left[ R^{-1} \right]_{ia}^2 \right)^2 \right] \\
&= \frac{1}{(n-s)(n-s-3) n(n+2)} \left( 1 + \frac{1}{n-s-1} \right) \left\{ 2 \operatorname{tr} \left( \Sigma^4 \right) + \operatorname{tr} \left( \Sigma^2 \right)^2 \right\} \\
&= \frac{2 \left( n_1 \sigma_1^4 + n_2 \sigma_2^4 \right) + \left( n_1 \sigma_1^2 + n_2 \sigma_2^2 \right)^2}{(n-s)(n-s-3) n(n+2)} \left( 1 + \frac{1}{n-s-1} \right) .
\end{aligned}
$$

Therefore:

$$
\begin{aligned}
\frac{\mathrm{Var}\left(Y_i'\right)}{\left(\mathbb{E}\left[Y_i'\right]\right)^2} &= \frac{\mathbb{E}\left[\left(Y_i'\right)^2\right] - \left(\mathbb{E}\left[Y_i'\right]\right)^2}{\left(\mathbb{E}\left[Y_i'\right]\right)^2} \\
&= \frac{\frac{2\left(n_1\sigma_1^4 + n_2\sigma_2^4\right) + \left(n_1\sigma_1^2 + n_2\sigma_2^2\right)^2}{(n-s)(n-s-3)n(n+2)}\left(1 + \frac{1}{n-s-1}\right) - \left(\frac{n_1\sigma_1^2 + n_2\sigma_2^2}{n(n-s-1)}\right)^2}{\left(\frac{n_1\sigma_1^2 + n_2\sigma_2^2}{n(n-s-1)}\right)^2} \\
&= \frac{\frac{2\left(n_1\sigma_1^4 + n_2\sigma_2^4\right) + \left(n_1\sigma_1^2 + n_2\sigma_2^2\right)^2}{(n-s)(n-s-3)n(n+2)}\left(1 + \frac{1}{n-s-1}\right) - \left(\frac{n_1\sigma_1^2 + n_2\sigma_2^2}{n(n-s-1)}\right)^2}{\left(\frac{n_1\sigma_1^2 + n_2\sigma_2^2}{n(n-s-1)}\right)^2} \\
&= \frac{n^2\left(n-s-1\right)^2}{(n-s)\left(n-s-3\right)n\left(n+2\right)}\left(\frac{2\left(n_1\sigma_1^4 + n_2\sigma_2^4\right) + \left(n_1\sigma_1^2 + n_2\sigma_2^2\right)^2}{\left(n_1\sigma_1^2 + n_2\sigma_2^2\right)^2}\right) \\
&\quad \times \left(1 + \frac{1}{n-s-1}\right) - 1 \\
&= \frac{n\left(n-s-1\right)}{\left(n-s-3\right)\left(n+2\right)}\left(\frac{2\left(n_1\sigma_1^4 + n_2\sigma_2^4\right)}{\left(n_1\sigma_1^2 + n_2\sigma_2^2\right)^2} + 1\right) - 1 \\
&= \frac{n\left(n-s-1\right)}{\left(n-s-3\right)\left(n+2\right)}\left(\frac{2n_1\sigma_1^4 + 2n_2\sigma_2^4}{n_1^2\sigma_1^4 + n_2^2\sigma_2^4 + 2n_1n_2\sigma_1^2\sigma_2^2} + 1\right) - 1 \\
&\leq \frac{n\left(n-s-1\right)}{\left(n-s-3\right)\left(n+2\right)}\left(\frac{2}{n_1} + \frac{2}{n_2} + 1\right) - 1 \\
&= \frac{n\left(n-s-1\right)}{\left(n-s-3\right)\left(n+2\right)}\left(\frac{2}{n_1} + \frac{2}{n_2}\right) + \left\{\frac{n\left(n-s-1\right)}{\left(n-s-3\right)\left(n+2\right)} - 1\right\} \\
&= \frac{n\left(n-s-1\right)}{\left(n-s-3\right)\left(n+2\right)}\left(\frac{2}{n_1} + \frac{2}{n_2}\right) \\
&\quad + \frac{n^2 - ns - n - n^2 + ns + 3n - 2n + 2s + 6}{\left(n-s-3\right)\left(n+2\right)} \\
&= \frac{n\left(n-s-1\right)}{\left(n-s-3\right)\left(n+2\right)}\left(\frac{2}{n_1} + \frac{2}{n_2}\right) + \frac{2\left(s+3\right)}{\left(n-s-3\right)\left(n+2\right)}.
\end{aligned}
$$

Hence there exists a universal constant $K_2$ such that:

$$
\frac{\mathrm{Var}\left(Y_i'\right)}{\left(\mathbb{E}\left[Y_i'\right]\right)^2} \leq K_2\left(\frac{1}{n_1} + \frac{1}{n_2}\right).
$$

By Chebyshev's inequality, we conclude:

$$
\mathbb{P}\left(Y_i' \geq 2\mathbb{E}\left[Y_i'\right]\right) \leq K_2\left(\frac{1}{n_1} + \frac{1}{n_2}\right). \tag{65}
$$

Bringing together (63) and (65) and using a union bound, we conclude that there exists a universal constant $K > 0$ such that:

$$
\mathbb{P}\left(|Y_i| \geq \frac{n\lambda_p\sqrt{s}}{n-s-1}, \ \text{ or } \ |Y_i'| \geq 2\mathbb{E}\left[Y_i'\right]\right) \leq K\left(\frac{1}{n_1} + \frac{1}{n_2}\right). \tag{66}
$$

$\square$

## E    PROOF OF PROPOSITION 4.1

*Proof of Proposition 4.1.* First, assume there exists $\left(\lambda_p\right)_{p\geq 1} \to 0$ such that (28) holds. By the first part of (28), we have:

$$
\frac{\sigma_{\mathrm{avg}}^2 \log\left(p-s\right)}{n} = o\left(\lambda_p^2\right). \tag{67}
$$

Using (67) and $(\lambda_p)_{p \geq 1} \to 0$, we get:

$$\frac{\sigma_{\text{avg}}^2 \log (p - s)}{n} \longrightarrow 0. \tag{68}$$

In addition, by the second part of (28), we have:

$$\lambda_p^2 = o\left(\frac{\rho^2}{s}\right). \tag{69}$$

Using (67) and (69), we get:

$$\frac{\sigma_{\text{avg}}^2 \log (p - s)\, s}{n \rho^2} \longrightarrow 0. \tag{70}$$

Taking the sum of (68) and (70), we obtain:

$$\frac{\sigma_{\text{avg}}^2 \log (p - s) \left(1 + s/\rho^2\right)}{n} \longrightarrow 0.$$

Hence, we conclude:

$$\sigma_{\text{avg}}^2 = o\left(\frac{n}{(1 + s/\rho^2) \log (p - s)}\right).$$

Second, assume (30) holds and let:

$$\lambda_p := \left(\frac{\sigma_{\text{avg}}^2 \log (p - s)}{n \left(1 + s/\rho^2\right)}\right)^{1/4}.$$

Then we have:

$$\lambda_p^2 = \sqrt{\frac{\sigma_{\text{avg}}^2 \log (p - s)}{(1 + s/\rho^2)\, n}}.$$

By (30), we have:

$$\lambda_p^2 = o\left(\sqrt{\frac{n \log (p - s)}{(1 + s/\rho^2)^2 \log (p - s)\, n}}\right)$$

$$= o\left(\frac{1}{1 + s/\rho^2}\right),$$

thus:

$$\lambda_p^2 \left(1 + s/\rho^2\right) \longrightarrow 0.$$

Therefore

$$\lambda_p \longrightarrow 0 \quad \text{and} \quad \frac{\lambda_p \sqrt{s}}{\rho} \longrightarrow 0. \tag{71}$$

In addition, we have by definition of $\lambda_p$:

$$\frac{n \lambda_p^2}{\sigma_{\text{avg}}^2 \log (p - s)} = \sqrt{\frac{\sigma_{\text{avg}}^2 \log (p - s)\, n^2}{\sigma_{\text{avg}}^4 \log (p - s)^2 (1 + s/\rho^2)\, n}}$$

$$= \sqrt{\frac{n}{\sigma_{\text{avg}}^2 (1 + s/\rho^2) \log (p - s)}}.$$

Therefore we have, by (30):

$$\frac{n \lambda_p^2}{\sigma_{\text{avg}}^2 \log (p - s)} \longrightarrow +\infty. \tag{72}$$

Finally, by (30) it holds that:

$$\frac{\sigma_{\text{avg}}^2 \log s}{n \rho^2} = o\left(\frac{n \log s}{n \rho^2 (1 + s/\rho^2) \log (p - s)}\right)$$

$$= o\left(\frac{1}{\rho^2 + s}\right) = o\left(\frac{1}{s}\right).$$

Therefore:

$$\frac{1}{\rho}\sqrt{\frac{\sigma_{\text{avg}}^2 \log s}{n}} \longrightarrow 0. \tag{73}$$

Bringing together (71), (72) and (73), we conclude that $\lambda_p \longrightarrow 0$ and (28) holds. $\qquad \square$

