# OpenReview forum: "Price of Quality: Sufficient Conditions for Sparse Recovery using Mixed-Quality Data"
_ICLR.cc/2026/Conference — ICLR 2026 Poster_

### Official Review · Reviewer_6vxY · 2025-10-18

**Soundness:** 3
**Presentation:** 3
**Contribution:** 3
**Rating:** 8
**Confidence:** 3

**Summary:**

Sparse recovery has typically been discussed under the assumption that noise variance is constant (homogeneous noise). However, real data often contains a mixture of a small number of "high-quality (low-noise)" observations (e.g., expert measurements) and a large number of "low-quality (high-noise)" observations (e.g., crowdsourcing or LLM labels).
This paper theoretically analyzes the support recovery problem for a sparse signal $\beta$ under mixed-quality data and defines two bounds:
i) Information-theoretic bounds: What sample size (n₁, n₂) is theoretically necessary for recovery?
ii) Algorithmic bounds: To what extent can LASSO recover the signal?

**Strengths:**

Originality: This paper  newly introduced the viewpoint that the Kronecker product structure is equivalent to "measuring a multidimensional signal at multiple stages." Based on this idea, it developed a signal recovery algorithm that operates with low computational complexity.
Quality: The developed algorithm reduces the run time for the recovery by $O(10^2)--O(10^3)$. In addition, theoretical guarantees are presented mathematically. The performance is also validated numerically. In summary, this paper is of high quality.
Clarity: The writing is well structured and the results are clear.
Significance: The results obtained in this paper are expected to bring significant advances to the problem of compressed sensing with Kronecker structures.

**Weaknesses:**

For problems where compressed sensing with a Kronecker structure is useful, the sparsity structure must be independent for each dimension. In the introduction, this paper mentions that there are examples of this in radar imaging and wireless communications, but does not provide specific examples. It is unclear how useful it is for practical applications.

**Questions:**

I understand that this paper is theoretical and not intended the practical usefulness. But, I would like to ask about its practical relevance. I could not imagine example problems for which the setup assumed in this paper are realistic. Can you raise up some?

---

> ### Comment · Reviewer_6vxY · 2025-11-20
> **Corrected my report.**
>
> Sorry. I realized that I had put incorrect comments for "Strengths" and "Weaknesses" parts in the report.
> I now corrected them. I keep my evaluation as it was.

---

> ### Author Response · Authors · 2025-11-20
>
> &#8202;
>
> We thank the reviewer for their thorough evaluation. Below, we address their questions and concerns to the best of our ability.
>
>    &#8202;
>
> ---
>
>    &#8202;
>
> > - **Weaknesses:** Regarding Lasso estimation, only agnostic situations are analyzed.
>
>    &#8202;
>
> - We thank the reviewer for noting this point. As discussed in Remark 4.2, extending the LASSO recovery analysis to the informed setting introduces substantial technical challenges. In particular, rescaling by $\Sigma^{-1}$ makes the rows of the design matrix heteroscedastic, making $X_S^\top \Sigma^{-2} X_S$ no longer Wishart and the rotational invariance (and associated Haar properties) used in the proof of Theorem 3 no longer apply. For this reason, we restricted our analysis to the agnostic case and leave the informed algorithmic threshold as an interesting direction for future work.
>
>    &#8202;
>
> ---
>
>    &#8202;
>
> > - **Questions:** I understand that this paper is theoretical and not intended the practical usefulness. But, I would like to ask about its practical relevance. I could not imagine example problems for which the setup assumed in this paper are realistic. Can you raise up some?
>
>    &#8202;
>
> - We thank the reviewer for this question. Our primary motivation comes from sparse linear regression in machine learning, where heterogeneous measurement noise naturally arises in large-scale data acquisition. More broadly, we believe classical compressive sensing applications such as the single-pixel camera, MRI, and radar (as discussed in Foucart and Rauhut’s introductory chapter) can also admit heterogeneous noise levels when implemented in practice. In these settings, assuming different variances across measurements yields exactly the mixed-quality observation model studied in our paper.
>
>    &#8202;

---

### Official Review · Reviewer_QshA · 2025-10-29

**Soundness:** 3
**Presentation:** 3
**Contribution:** 3
**Rating:** 8
**Confidence:** 3

**Summary:**

In this paper, the authors study sparse recovery from noisy linear observations of mixed quality. To be more precise, they seek to recover an $s$-sparse ground-truth $\beta_* \in \mathbb R^p$ from $n$ observations

$Y = X\beta_* + Z \in \mathbb R^n,$

where the measurement process is modelled by a Gaussian matrix $X$, and the entries of $Z$ model random noise and are zero-mean Gaussians. For $n_1+n_2=n$, the first $n_1$ observations $Y_1,…,Y_{n_1}$ are of high quality (modelled by noise variance $\sigma_1^2$) whereas the remaining $n_2$ observations $Y_{n_1+1},…,Y_n$ are of low quality (modelled by noise variance $\sigma_2^2 > \sigma_1^2$). The authors consider the agnostic setting in which the recovery method has no information on the data quality, and the informed setting in which the recovery method knows which noise level applies to which observation.

Theorem 1 provides sufficient conditions on $n_1,n_2,n$ for asymptotic support recovery in the agnostic setting. The result shows that asymptotically a high-quality observation is never worth more than two low-quality observations.

Theorem 2 is the counterpart of Theorem 1 in the informed setting and shows that depending on the ratio between $\sigma_1$ and $\sigma_2$ a high-quality observation can be worth more than any number of low-quality observations.

Finally, Theorem 3 analyzes sufficient and necessary conditions on $n$ to achieve support recovery algorithmically.

**Strengths:**

+ Rigorous results
+ Study well-motivated via mixed quality data in learning problems

**Weaknesses:**

- Algorithmic recovery is not analyzed in the informed setting

**Questions:**

Since the results are interesting and the derivation appears to be rigorous, I do recommend this paper to be accepted. Due to the limited reviewing time, I have only loosely screened parts of the supplementary material, so I cannot confirm correctness of the presented results. They appear to be reasonable though.

While the paper is well-written and easy to read, the notation can be improved at certain points and there are some points that should be corrected:

- X,Y,Z are all upper-case although X is a matrix and Y,Z are vectors. In contrast $\beta$ is lower case. This is confusing.
- l. 126: „… a linear combination of ??? has the …“
- l. 199: Maybe I missed it, but I think Z_1,Z_2 are used without being defined
- l. 235: Can you comment on the fact that your required size of n_* is smaller for s=p than for s=0.5p? Clearly, there is less ambiguity in the possible supports, nevertheless, this is rather counter-intuitive and should be properly discussed in the paper.
- l. 251: „…Chernoff bound to the LHS…“ -> Shouldn’t this be RHS?
- Eq (11): $\sigma_4^2$ -> $\sigma_2^4$
- Remark 3.2, second bullet point: It would make sense to mention that the proposed estimator is motivated by the MLE discussed afterwards in Section 3.2
- Eq (20): This should be argmin and not min
- l. 604: $\hat S$ is only an element of the argmin not necessarily the only solution.
- l. 725: If I'm not mistaken, you apply Markov’s inequality here, and not a Chernoff bound. This would need to be adapted all over the document.

---

> ### Author Response · Authors · 2025-11-20
>
> &#8202;
>
> We thank the reviewer for their thorough evaluation. Below, we address their questions and concerns to the best of our ability.
>
>    &#8202;
>
> ---
>
>    &#8202;
>
> > - **Weaknesses:** Algorithmic recovery is not analyzed in the informed setting.
>
>    &#8202;
>
> - We thank the reviewer for noting this point. As discussed in Remark 4.2, extending the LASSO recovery analysis to the informed setting introduces substantial technical challenges. In particular, rescaling by $\Sigma^{-1}$ makes the rows of the design matrix heteroscedastic, making $X_S^\top \Sigma^{-2} X_S$ no longer Wishart and the rotational invariance (and associated Haar properties) used in the proof of Theorem 3 no longer apply. For this reason, we restricted our analysis to the agnostic case and leave the informed algorithmic threshold as an interesting direction for future work.
>
>    &#8202;
>
> ---
>
>    &#8202;
>
> > - **Questions:** $X,Y,Z$ are all upper-case although $X$ is a matrix and $Y,Z$ are vectors. In contrast $\beta$ is lower case. This is confusing.
>
>    &#8202;
>
> - We thank the reviewer for the comment. In our notation, $X$, $Y$, and $Z$ are uppercase because they denote random entities, whereas $\beta$ is deterministic. This choice is consistent with conventions in prior sparse-recovery literature (e.g., Aeron et al., 2010; Reeves et al., 2019; Wang et al., 2010). We will add a remark clarifying this point.
>
>    &#8202;
>
> ---
>
>    &#8202;
>
> > - **Questions:** l. 126: "... a linear combination of ??? has the ...".
>
>    &#8202;
>
> - We thank the reviewer. The intended text is "the condition has the form ...". We will correct this in the final version.
>
>    &#8202;
>
> ---
>
>    &#8202;
>
> > - **Questions:** l. 199: Maybe I missed it, but I think $Z_1,Z_2$ are used without being defined.
>
>    &#8202;
>
> - We thank the reviewer for pointing this out. $Z^1$ and $Z^2$ denote the subvectors of $Z$ corresponding to the high- and low-quality observations. In the final version, we will explicitly define $Z^1 = (Z_1,\dots,Z_{n_1})^T$ and $Z^2 = (Z_{n_1+1},\dots,Z_n)^T$.
>
>    &#8202;
>
> ---
>
>    &#8202;
>
> > - **Questions:** l. 235: Can you comment on the fact that your required size of $n^\star$ is smaller for $s=p$ than for $s=0.5 p$? Clearly, there is less ambiguity in the possible supports, nevertheless, this is rather counter-intuitive and should be properly discussed in the paper.
>
>    &#8202;
>
> - We thank the reviewer for raising this point. In our setup, the decoder knows that $\beta^\star$ is exactly $s$-sparse. When $s=p$, the support is fully determined and no samples are needed. As $s$ decreases, the cardinality of the set of possible supports, $\binom{p}{s}$ increases, peaking at $s = 0.5 p$, making the recovery problem information-theoretically harder. We will clarify this intuition when we define $n^\star$.
>
>    &#8202;
>
> ---
>
>    &#8202;
>
> > - **Questions:** l. 251: "...Chernoff bound to the LHS..." $\rightarrow$ Shouldn’t this be RHS?
>
>    &#8202;
>
> - We thank the reviewer. Indeed, the text should refer to the right-hand side. We will correct this in the final version.
>
>    &#8202;
>
> ---
>
>    &#8202;
>
> > - **Questions:** Eq (11): $\sigma_4^2 \rightarrow \sigma_2^4$
>
>    &#8202;
>
> - We thank the reviewer for pointing this out. We will correct it in the final version of the paper.
>
>    &#8202;
>
> ---
>
>    &#8202;
>
> > - **Questions:** Remark 3.2, second bullet point: It would make sense to mention that the proposed estimator is motivated by the MLE discussed afterwards in Section 3.2.
>
>    &#8202;
>
> - We thank the reviewer for their helpful suggestion. Indeed, the proposed estimator in Remark 3.2 is motivated by the structure of the maximum likelihood estimator introduced in Section 3.2. We will make this connection explicit in the final version to improve readability.
>
>    &#8202;
>
> ---
>
>    &#8202;
>
> > - **Questions:** Eq (20): This should be argmin and not min
>
>    &#8202;
>
> - We thank the reviewer for pointing this out. Equation (20) should indeed use $\text{argmin}$ rather than $\min$. We will correct this in the final version.
>
>    &#8202;
>
> ---
>
>    &#8202;
>
> > - **Questions:** l. 604: $\hat{S}$ is only an element of the argmin not necessarily the only solution.
>
>    &#8202;
>
> - We thank the reviewer for this remark. We will correct the notation in the final version to make it clear that $\hat S$ denotes an element of the argmin set. We note that in our Gaussian design and Gaussian noise model, the losses associated with different supports are distinct with probability one, hence the minimizer is almost surely unique.
>
>    &#8202;
>
> ---
>
>    &#8202;
>
> > - **Questions:** l. 725: If I'm not mistaken, you apply Markov’s inequality here, and not a Chernoff bound. This would need to be adapted all over the document.
>
>    &#8202;
>
> - We thank the reviewer for noting this. While in that specific line we indeed apply Markov's inequality; the overall argument corresponds to the standard Chernoff-bound derivation. We will revise this in the final version to make the distinction clear.

---

> ### Comment · Reviewer_QshA · 2025-11-21
>
> If the authors implement the promised changes, they will address my comments. I thus keep my score.

---

### Official Review · Reviewer_zQsn · 2025-10-31

**Soundness:** 4
**Presentation:** 3
**Contribution:** 3
**Rating:** 8
**Confidence:** 4

**Summary:**

The paper studies the problem of sparse signal recovery in the mixed-quality data setting: where a subset of the data is observed with low noise, and the complement observed with different (higher, w.l.o.g) noise. They study the information theoretic thresholds of the problem, and demonstrate that the maximum likelihood estimator in the gaussian noise setting has significantly different behaviour depending on whether one is agnostic or not to the noise variances. They also analyze the LASSO and prove that it is robust under this model.

**Strengths:**

The paper is original, they study a timely problem and do so with an original look at a new variable called the price of quality. I must mention that such problems were already considered in the heterogeneous statistics literature, but they were moreso concerned about inferring the heterogeneities themselves, rather than quantifying downstream implications of standard models on mixed quality data. In this regard, I find this work novel.

The work is clear, and is of high quality as the authors investigate various aspects of the problem thoroughly.

The work is significant, especially in the era of AI, where models can often be trained using a mix of real and synthetic data. In particular, the authors clarify their findings and present them as important and non-trivial rules of thumb for machine learning, such as that in the agnostic setting, where one high-quality sample is never worth more than two low-quality samples.

**Weaknesses:**

I found no major weaknesses in the paper. Following were minor comments/questions:

- The information-theoretic and algorithmic thresholds in (2) are mainly for Gaussian noise right? Or have these been proven for additive noise more broadly? Please clarify this, and maybe state earlier on in the work that you consider additive independent Gaussian noise.
- Typos: First paragraph after 1.2.1, “In the first part of [our] work ([S]ection 3) …”. In the next paragraph: “, the condition requires that a linear combination of [?] has the form …”
- A bit more clarification as to how \sigma_1, \sigma_2 scale w.r.t other parameters, and remind this throughout the paper.
- Your results in Theorem 1 are mostly relevant for additive (sub)-Gaussian noise, please clarify this and that your sufficient conditions (and therefore findings) are not universal over general additive noise.
- End of page 7, you mention “OGP” but do not explain/state the full name or what it is. It would be good to get more background on that.
- Could you comment on how this relates to existing statistical work on heteroscedastic regression, and whether you expect improvements for algorithms which first estimate the heterogeneities, then proceed to weight samples accordingly?

**Questions:**

Questions asked in the “Weaknesses” section

---

> ### Author Response · Authors · 2025-11-20
>
> &#8202;
>
> We thank the reviewer for their thorough evaluation. Below, we address their questions and concerns to the best of our ability.
>
>    &#8202;
>
> ---
>
>    &#8202;
>
> > - **Weaknesses:** The information-theoretic and algorithmic thresholds in (2) are mainly for Gaussian noise right? Or have these been proven for additive noise more broadly? Please clarify this, and maybe state earlier on in the work that you consider additive independent Gaussian noise.
>
>    &#8202;
>
> - We thank the reviewer for highlighting this point. All our results are derived under additive independent Gaussian noise. We will explicitly state this assumption earlier in the introduction to avoid ambiguity.
>
>    &#8202;
>
> ---
>
>    &#8202;
>
> > - **Weaknesses:** Typos: First paragraph after 1.2.1, "In the first part of [our] work ([S]ection 3) …". In the next paragraph: ", the condition requires that a linear combination of [?] has the form ..."
>
>    &#8202;
>
> - We thank the reviewer for catching these typos. We will correct both in the final version as follows:
>
>    - "In the first part of our work (Section 3) ...",
>
>    - "the condition has the form ...".
>
>    &#8202;
>
> ---
>
>    &#8202;
>
> > - **Weaknesses:** A bit more clarification as to how $\sigma_1$, $\sigma_2$ scale w.r.t other parameters, and remind this throughout the paper.
>
>    &#8202;
>
> - We thank the reviewer for requesting clarification. Our analysis does not impose a unique scaling of $\sigma_1$ and $\sigma_2$ with respect to $p$ and $s$. Different SNR regimes (lines 203-206) correspond to different scalings of $\sigma_1$ and $\sigma_2$, each of which is meaningful in applications. In the final version, we will make sure to restate this explicitly in the introduction and recall it when introducing the different regimes.
>
>    &#8202;
>
> ---
>
>    &#8202;
>
> > - **Weaknesses:** Your results in Theorem 1 are mostly relevant for additive (sub)-Gaussian noise, please clarify this and that your sufficient conditions (and therefore findings) are not universal over general additive noise.
>
>    &#8202;
>
> - We thank the reviewer for raising this point. Indeed, our results are stated for additive independent Gaussian noise. The same first moment arguments extend to sub-Gaussian noise with equivalent variance proxy. However, our sufficient conditions are not universal for arbitrary additive noise distributions. We will clarify this point explicitly in the final version of the paper.
>
>    &#8202;
>
> ---
>
>    &#8202;
>
> > - **Weaknesses:** End of page 7, you mention "OGP" but do not explain/state the full name or what it is. It would be good to get more background on that.
>
>    &#8202;
>
> - We thank the reviewer for this remark. The abbreviation "OGP" refers to the Overlap Gap Property, which we stated earlier in the introduction (line 59). However, since the acronym is not used frequently throughout the text, the reader may lose track of this definition. In the final version, we will make sure to restate the full name and include a brief background for clarity.
>
>    &#8202;
>
> ---
>
>    &#8202;
>
> > - **Weaknesses:** Could you comment on how this relates to existing statistical work on heteroscedastic regression, and whether you expect improvements for algorithms which first estimate the heterogeneities, then proceed to weight samples accordingly?
>
>    &#8202;
>
> - We thank the reviewer for this insightful question. This point is closely related to the discussion in Remark 3.2. Incorporating an explicit variance–estimation step would introduce an additional layer of complexity, since one must control both the accuracy of the estimated variances and their effect on the subsequent weighted procedure. Although beyond the scope of this paper, this is a promising direction for future work. We expect such methods to yield improvement, especially when the two SNRs scale differently. Regarding prior work, classical approaches to heteroscedastic regression (see e.g. Buja et al., 2019) assume either known variances or estimate them first and then perform weighted fitting. To the best of our knowledge, these works focus on parameter estimation under general variance structures, rather than sparse support recovery under a mixed-quality model. In the final version, we will make sure to further highlight the idea of adding a heterogeneity-estimation step in Remark 3.2. In addition, we will include a discussion of the literature on heteroscedastic regression and the way it relates to our work.

---

### Official Review · Reviewer_dyNa · 2025-10-31

**Soundness:** 3
**Presentation:** 3
**Contribution:** 2
**Rating:** 6
**Confidence:** 4

**Summary:**

The paper studies the problem of support recovery from noisy observations in a linear regression setting with Gaussian design. The design is assumed to be centered and isotropic, while the noise is Gaussian with zero mean and a diagonal covariance matrix that takes two possible values: $\sigma_1$ and $\sigma_2$.

The paper focuses on two types of results: first, sufficient conditions for the maximum likelihood estimator (computationally inefficient) to consistently recover the support, considering both the ill-specified and well-specified noise covariance matrix cases; second, sufficient conditions ensuring that the Lasso estimator consistently identifies both the support and the sign of the regression vector.

**Strengths:**

- The problem of recovering the support of an unknown vector is a fundamental challenge in statistics, frequently regarded as a prototypical case for mathematical results on variable selection. Exploring the information-theoretic and algorithmic limitations of this problem holds, in my opinion, significant interest for the ICLR audience.

 - The mathematical proofs, as far as I could check, appear to be correct.

 - The proof of Theorem 3 overcomes some non-trivial technical challenges.

**Weaknesses:**

- I suspect that **Theorem 1** is not sharp, for two main reasons:
  - When $n_2 = 0$ (i.e., no low-quality data is present), the theorem's condition requires $n_1$ to exceed a quantity involving $\sigma_2$. Intuitively, $\sigma_2$ should not appear in this condition under these circumstances.
  - As noted in lines 265–269, if $\sigma_2 = o(s)$, the condition becomes independent of $\sigma_1$, which seems counterintuitive.

- The conclusion at the top of page 6, *"one unit of high-quality data is worth at most 2 units of low-quality data,"* does not seem justified by **Theorem 1**. The theorem provides only a **sufficient condition** for recovery, but it does not rule out the possibility that the maximum likelihood estimator could detect the support with a smaller value of $n_1$.

- The conditions under which the results are proved, although similar to those required in many papers on this topic, are quite restrictive.

**Questions:**

- Since in Thm 3 it is required that $s=o(p)$, can we replace $n_{Alg}$ by its asymptotic equivalent $2s\log (p)$?

- Can we generalize Theorem 2 to arbitrary matrices $\Sigma$, just by replacing the LHS of (15) by $\sum_{i=1}^n \log(1+\frac{\delta s}{2\lambda_i(\Sigma)})$?

 - What would happen if, in **Theorems 1 and 2**, we replaced the condition that nonzero elements are equal to one with the condition that they are either $1$ or $-1$? Would sign recovery still hold under the same conditions?

---

> ### Author Response · Authors · 2025-11-19
>
> &#8202;
>
> We thank the reviewer for their thorough evaluation. Below, we address their questions and concerns to the best of our ability. Due to  the character limit of a single official comment, we address their last two questions (on the generalization of Theorem 2 to arbitrary $\Sigma$ and on the information-theoretic conditions for signed support recovery) in two separate comments.
>
>    &#8202;
>
> ---
>
>    &#8202;
>
> > - **Weaknesses:** I suspect that Theorem 1 is not sharp, for two main reasons:
> >   - When $n_2 = 0$ (i.e., no low-quality data is present), the theorem's condition requires $n_1$ to exceed a quantity involving $\sigma_2$. Intuitively, $\sigma_2$ should not appear in this condition under these circumstances.
> >   - As noted in lines 265–269, if $\sigma_2 = o(s)$, the condition becomes independent of $\sigma_1$, which seems counterintuitive.
>
>    &#8202;
>
> - We thank the reviewer for this observation. Indeed, Theorem 1 provides a sufficient condition rather than a sharp information-theoretic threshold. As discussed in Remark 3.2, optimizing the Chernoff bound in the heterogeneous-noise case requires solving a cubic equation (eq. (33)), and we expect that using the exact root would yield a tight bound, as in the homogeneous setting (Wang et al., 2010; Gamarnik and Zadik, 2022). To keep the sufficient condition in a tractable closed form, we opted for a relaxation. In the final version, we will expand the discussion around this point and explicitly comment on the role of eq. (33).
>
>    &#8202;
>
> ---
>
>    &#8202;
>
> > - **Weaknesses:**  The conclusion at the top of page 6: "one unit of high-quality data is worth at most 2 units of low-quality data", does not seem justified by Theorem $1$. The theorem provides only a sufficient condition for recovery, but it does not rule out the possibility that the maximum likelihood estimator could detect the support with a smaller value of $n_1$.
>
>    &#8202;
>
> -   We thank the reviewer for pointing this out. Indeed, the phrase "is worth at most" suggests a *necessary* condition, whereas Theorem 1 only provides a *sufficient* one. Our intended meaning was: if $(n_1,n_2)$ satisfies condition (9), then replacing one high-quality sample with $\gamma$ low-quality samples still satisfies the same sufficient condition. We agree that the original phrasing may be misleading, and in the final version we will use the following more precise formulation: "under our sufficient condition, one high-quality sample can be replaced by up to two low-quality samples."
>
>    &#8202;
>
> ---
>
>    &#8202;
>
> > - **Weaknesses:** The conditions under which the results are proved, although similar to those required in many papers on this topic, are quite restrictive.
>
>    &#8202;
>
> - We thank the reviewer for this comment. The assumptions we impose are standard in the sparse-recovery literature and align with those used in prior work on information-theoretic and algorithmic thresholds (e.g., Wainwright, 2009; Reeves et al., 2019; Gamarnik and Zadik, 2022). They allow us to isolate the effect of heterogeneous noise while retaining the standard structure of the recovery problem. We will clarify this in the final version.
>
>    &#8202;
>
> ---
>
>    &#8202;
>
> > - **Questions:** Since in Theorem $3$ it is required that $s = o(p)$, can we replace $n_{\text{ALG}}$ by its asymptotic equivalent $2 s \log{p}$?
>
>    &#8202;
>
> - We thank the reviewer for raising this question. Yes, since $n_{\text{ALG}} = 2 s \log(p)(1 + o(1))$, then "$ \exists \text{ } \varepsilon_1 > 0$ such that $n > (1 + \varepsilon_1) n_{\text{ALG}}$" is equivalent to "$ \exists \text{ } \varepsilon_2 > 0$ such that $n > (1 + \varepsilon_2) 2 s \log{p}$" for large enough $n, p, s$. The same holds for the necessary condition.

---

> > ### Author Response · Authors · 2025-11-20
> > **Generalization of Theorem 2 to arbitrary $\Sigma$**
> >
> > &#8202;
> >
> > In this comment, we address the reviewer's question on the generalization of Theorem 2 to arbitrary $\Sigma$.
> >
> >    &#8202;
> >
> > ---
> >
> >    &#8202;
> >
> > > - **Questions:** Can we generalize Theorem 2 to arbitrary matrices $\Sigma$, just by replacing the LHS of (15) by $\sum_{i=1}^{n} \log(1 + \frac{\delta s}{2 \lambda_i(\Sigma)})$?
> >
> >    &#8202;
> >
> > -  We thank the reviewer for this very interesting question. Indeed, we can generalize Theorem 2 to arbitrary matrix $\Sigma$, just by replacing the LHS of (15) by $\sum_{i=1}^{n} \log(1 + \frac{\delta s}{2 \sigma_i(\Sigma)^2})$, as long as $\Sigma$ is invertible. The proof of Theorem 2 can be generalized to this setting as follows.
> >
> >    &#8202;
> >
> >    The key generalization is of Proposition C.1. The rest of the proof of Theorem 2 remains the same. The proof of Proposition C.1 (starting line 942) can be generalized as follows. The expansion of $\Delta(S)$ (lines 945-955) remains as in line 949:
> >    &#8202;
> >
> >    $\hspace{5cm} \Delta(S) =||\Sigma^{-1} X (1_{S^\star} - 1_{S})||_2^2 + 2 <\Sigma^{-1} Z,\Sigma^{-1} X (1_{S^\star} - 1_{S})>,$
> >
> >    &#8202;
> >
> >     without expanding on the $\sigma_1^2$ and $\sigma_2^2$ blocks (lines 951-955). Then the Chernoff bound (lines 977 - 996) can be generalized to:
> >    &#8202;
> >
> >    $\hspace{5cm} \mathbb{P}(\Delta(S) \leq 0) = \mathbb{P}(- \Delta(S) \geq 0)$
> >
> >    $\hspace{7.4cm} \leq \inf_{\theta \geq 0} \mathbb{E}[e^{-\theta \Delta(S)}]$
> >
> >    $\hspace{7.4cm} = \inf_{\theta \geq 0} \mathbb{E}[e^{-\theta ||\Sigma^{-1} X (1_{S^\star} - 1_{S})||_2^2 - 2 \theta <\Sigma^{-1} Z,\Sigma^{-1} X (1_{S^\star} - 1_{S})>}]$
> >
> >    $\hspace{7.4cm} = \inf_{\theta \geq 0} \mathbb{E} [ \mathbb{E} [ e^{-\theta ||\Sigma^{-1} X (1_{S^\star} - 1_{S})||_2^2 - 2 \theta <\Sigma^{-1} Z,\Sigma^{-1} X (1_{S^\star} - 1_{S})>} | X ] ]$
> >
> >    $\hspace{7.4cm} = \inf_{\theta \geq 0} \mathbb{E} [ e^{-\theta ||\Sigma^{-1} X (1_{S^\star} - 1_{S})||_2^2} \mathbb{E} [ e^{- 2 \theta <W,\Sigma^{-1} X (1_{S^\star} - 1_{S})>} | X ] ]$
> >
> >    $\hspace{7.4cm} = \inf_{\theta \geq 0} \mathbb{E} [ e^{-\theta ||\Sigma^{-1} X (1_{S^\star} - 1_{S})||_2^2} M_W (- 2 \theta \Sigma^{-1} X(\beta^\star - \beta) | X)]$
> >
> >    $\hspace{7.4cm} = \inf_{\theta \geq 0} \mathbb{E} [ e^{-\theta ||\Sigma^{-1} X (1_{S^\star} - 1_{S})||_2^2} e^{\frac{1}{2} ||- 2 \theta \Sigma^{-1} X(\beta^\star - \beta)||_2^2} ]$
> >
> >    $\hspace{7.4cm} = \inf_{\theta \geq 0} \mathbb{E} [ e^{(-\theta + 2 \theta^2) ||\Sigma^{-1} X (1_{S^\star} - 1_{S})||_2^2} ]$
> >
> >    $\hspace{7.4cm} = \mathbb{E} [ e^{- ||\Sigma^{-1} X (1_{S^\star} - 1_{S})||_2^2 / 8} ]$
> >
> >    $\hspace{7.4cm} = \mathbb{E} [ e^{- (1_{S^\star} - 1_{S})^T X^T (\Sigma \Sigma^T)^{-1} X (1_{S^\star} - 1_{S}) / 8} ]$
> >
> >    &#8202;
> >
> >     Note that $\Sigma \Sigma^T$ is symmetric positive definite and therefore diagonalizable. We write $\Sigma \Sigma^T = Q^{-1} \Lambda^2 Q$ with $\Lambda = \text{diag}([\sigma_i(\Sigma)]_{i=1}^{n})$ and $Q \in \mathcal{O}(n)$. Therefore, the above writes:
> >
> >    &#8202;
> >
> >    $\hspace{5cm} \mathbb{P}(\Delta(S) \leq 0) \leq  \mathbb{E} [ e^{- (1_{S^\star} - 1_{S})^T X^T Q^T \Lambda^{-2} Q X (1_{S^\star} - 1_{S}) / 8} ]$
> >
> >    $\hspace{7.4cm} \leq \mathbb{E} [ e^{- (1_{S^\star} - 1_{S})^T X^T Q^T (\Lambda^{-1})^T \Lambda^{-1} Q X (1_{S^\star} - 1_{S}) / 8} ]$
> >
> >    $\hspace{7.4cm} \leq \mathbb{E} [ e^{- || \Lambda^{-1} Q X (1_{S^\star} - 1_{S}) ||_2^2 / 8} ]$.
> >
> >    &#8202;
> >
> >    Since $Q$ is orthogonal and $X$ is Gaussian, we have $QX \stackrel{d}{=} X$, therefore:
> >
> >    &#8202;
> >
> >    $\hspace{5cm} \mathbb{P}(\Delta(S) \leq 0)  \leq \mathbb{E} [ e^{- || \Lambda^{-1} X (1_{S^\star} - 1_{S}) ||_2^2 / 8} ]$.
> >
> >    &#8202;
> >
> >    This brings us back to the diagonal-$\Sigma$ setting, therefore:
> >
> >    &#8202;
> >
> >    $\hspace{5cm} \mathbb{P}(\Delta(S) \leq 0)  \leq \mathbb{E} [ e^{- || \Lambda^{-1} X (1_{S^\star} - 1_{S}) ||_2^2 / 8} ]$
> >
> >    $\hspace{7.4cm} = \prod_{i=1}^{n} \mathbb{E} [ e^{- ( \sum_{j \in U} X_{ij} - \sum_{j \in V} X_{ij} )^2 / (8 \sigma_i (\Sigma)^2 )} ]$
> >
> >    $\hspace{7.4cm} \leq \prod_{i=1}^{n} ( 1 + \frac{\delta s}{2 \sigma_i (\Sigma)^2} )^{-1/2}$.
> >
> >    &#8202;
> >
> >    Therefore the generalization of Proposition C.1, and hence of Theorem 2, follow.
> >
> >    &#8202;
> >
> >    We expect that, similarly, condition (9) in Theorem (1) can be generalized to arbitrary invertible $\Sigma$ as follows:
> >
> >    &#8202;
> >
> >    $\hspace{5cm} \sum_{i=1}^{n} \log ( 1 + \frac{\delta (2 \sigma_{max}(\Sigma)^2 - \sigma_{i}(\Sigma)^2 ) s}{2 \sigma_{max}(\Sigma)^4} ) \geq (1 + \varepsilon) n^\star$.
> >
> >    &#8202;
> >
> >     We will update the manuscript to include these extensions and their complete proofs in the final version of the paper. Again, we thank the reviewer for raising this interesting question.

---

> ### Author Response · Authors · 2025-11-20
> **Information-theoretic conditions for signed support recovery**
>
> &#8202;
>
> In this comment, we address the reviewer's question on information-theoretic conditions for signed support recovery.
>
>    &#8202;
>
> ---
>
>    &#8202;
>
> > - **Questions:** What would happen if, in Theorems $1$ and $2$, we replaced the condition that nonzero elements are equal to one with the condition that they are either $+1$ or $-1$? Would sign recovery still hold under the same conditions?
>
>    &#8202;
>
> -  We thank the reviewer for raising up this interesting point. Yes, signed recovery would follow. The threshold $n^\star$ remains the same in the sub-linear sparsity regime ($s = o(p)$) and increases in the linear sparsity regime ($s = \alpha p$). For this, we extend the search space of $\hat{\beta}$ (see eq. (8) for Theorem 1 and eq. (14) for Theorem 2) to the set of signed supports, i.e.:
>
>    &#8202;
>
>    $\hspace{6cm} \mathcal{B}_{p,s} \coloneqq \Big\\{\beta \in \\{-1,0,1\\}^p \hspace{0.1cm} \colon \hspace{0.1cm} ||\beta||_0 = s \Big\\}.$
>
>    &#8202;
>
>    Then the large deviation bound on $\mathbb{P}(\Delta(S) \leq 0)$ (see Proposition A.1 in the proof of Theorem 1 and Proposition C.1 in the proof of Theorem 2) extends to this setting as follows. Let $W = \text{Supp}(\beta^\star) \cap  \text{Supp}(\beta)$. We have:
>
>    &#8202;
>
>    $\hspace{0.5cm} <X_i,\beta^\star - \beta>
>     = \sum_{j \in U} \beta^\star_j X_{ij} - \sum_{j \in V} \beta_j X_{ij} + \sum_{j \in W} (\beta^\star_j - \beta_j) X_{ij} \sim \mathcal{N} \Big( 0, |U \cup V| +  \sum_{j \in W} (\beta^\star_j - \beta_j)^2 \Big).$
>
>    &#8202;
>
>    Since the variance above is larger than $|U \cup V|$, the MGF evaluations (see line 822 in Proposition A.1 and line 1054 in Proposition C.1) can only get smaller, hence the upper bound on $\mathbb{P}(\Delta(S) \leq 0)$ still holds. The only difference would occur in the size of the search space. In fact, the LHS of the current condition, i.e.:
>
>    &#8202;
>
>    $\hspace{8cm} n^\star \simeq 2 \log{\binom{p}{s}},$
>
>    &#8202;
>
>    corresponds to twice the $\log$ of the size of the search space:
>
>    &#8202;
>
>    $\hspace{8cm}|\mathcal{S}_{p,s}| = \binom{p}{s}.$
>
>    &#8202;
>
>    For the new setting where $\beta^\star \in \mathcal{B}_{p,s}$ defined above, this becomes:
>
>    &#8202;
>
>    $\hspace{3cm} n^\star_{\text{signed, s-lin}} = 2 \log|\mathcal{B}_{p,s}| = 2 \log \Big(2^s \binom{p}{s} \Big) \simeq 2 s (\log{2} + \log(p/s)) \simeq 2 s \log(p/s)$,
>
>    &#8202;
>
>    (unchanged) when $s = o(p)$, and:
>
>    &#8202;
>
>    $\hspace{3cm} n^\star_{\text{signed, lin}} = 2 \log|\mathcal{B}_{p,s}| = 2 \log\Big(2^s \binom{p}{s}\Big) \simeq 2 s \log{2} + 2 p h (\alpha) = 2 (\alpha \log{2} + h(\alpha)) p,$
>
>    &#8202;
>
>    when $s = \alpha p$. We will provide the complete derivation in the final version. We thank the reviewer for raising this interesting point, which we will incorporate in a clear and fully rigorous form.

---

### Meta-Review · Area_Chair_qbnu · 2026-01-18

**Summary:**

The paper studies sparse recovery from mixed-quality data, introducing the “Price of Quality” to quantify trade-offs between high- and low-quality samples.

In general, the reviewers recommended acceptance for this paper, recognizing the paper’s solid and significant contributions.. Here is a summary of the main concerns from reviewers:

- Reviewer dyNa noted that some sufficient conditions (e.g., “one high-quality sample is worth at most two low-quality samples”) are not guaranteed to be tight.

- Reviewers dyNa and zQsn mentioned that assumptions on isotropic Gaussians and equal non-zero coefficients may limit the broadness of the paper.

- Reviewer QshA pointed out that the algorithmic analysis focuses on the agnostic setting but does not cover the informed setting.

**Reviewer Concerns:**

The authors have adequately addressed the main concerns raised by the reviewers in their rebuttal. Although they did not fully resolve Reviewer dyNa’s concern regarding the sharpness of Theorem 1, they provided relevant literature and committed to adding a discussion in the final version.

**Reviewer Scores:**

Reviewer dyNa would likely have maintained a score of 6, as their main concerns regarding the non-sharpness of Theorem 1 and restrictive assumptions were acknowledged but not fully resolved.

The remaining three reviewers would likely have maintained their original scores, as they recommended acceptance and their questions were addressed.

---

### Decision · Program_Chairs · 2026-01-26

Accept (Poster)